# Implementation and assessment of a model including mixotrophs and the carbonate cycle (Eco3M_MIX-CarbOx v1.0) in a highly dynamic Mediterranean coastal environment (Bay of Marseille, France) (Part. II): Towards a better representation of total alkalinity when modelling the carbonate system and air-sea CO₂ fluxes

Lucille Barré[1], Frédéric Diaz[1,†], Thibaut Wagener[1], Camille Mazoyer[1], Christophe Yohia[2] and Christel Pinazo[1]

[1]Aix Marseille Univ., Université de Toulon, CNRS, IRD, MIO, UM 110, 13288, Marseille, France
[2]Aix Marseille Univ., Université de Toulon, CNRS, IRD, OSU Institut Pythéas, 13288, Marseille France
[†]Deceased

*Correspondence to*: Lucille Barré (lucille.barre@mio.osupytheas.fr), Thibaut Wagener (thibaut.wagener@mio.osupytheas.fr)

**Abstract**

The Bay of Marseille (BoM), located in the north-western Mediterranean Sea, is affected by various hydrodynamic processes (e.g., Rhône River intrusion and upwelling events) that result in a highly complex local carbonate system. In any complex environment, the use of models is advantageous since it allows to identify the different environmental forcings, thereby facilitating a better understanding. By combining approaches from two biogeochemical ocean models and improving the formulation of total alkalinity, we develop a more realistic representation of the carbonate system variables at high temporal resolution which enables us study air-sea $CO_2$ fluxes and seawater $pCO_2$ variations more reliably. We apply this new formulation to two particular scenarios, typical for the BoM: (i) summer upwelling and (ii) Rhône River intrusion events. In both scenarios, our model was able to correctly reproduce the observed patterns of $pCO_2$ variability. Summer upwelling events are typically associated with $pCO_2$ decrease that mainly results from decreasing near-surface temperatures. Furthermore, Rhône River intrusion events are typically associated with $pCO_2$ decrease, although in this case the $pCO_2$ decrease results from a decrease in salinity and an overall increase in total alkalinity. While we were able to correctly represent the daily range of air-sea $CO_2$ fluxes, the present configuration of Eco3M_MIX-CarbOx does not allow to correctly reproduce the annual cycle of air-sea $CO_2$ fluxes observed in the area. This pattern directly impacts our estimates of the overall yearly air-sea $CO_2$ flux, as even if the model clearly identifies the bay as a $CO_2$ sink, its magnitude was underestimated which may be an indication of the limitations inherent in dimensionless models for representing air-sea $CO_2$ fluxes.

Keywords: Carbonate system, Bay of Marseille, Total alkalinity, Air-sea $CO_2$ fluxes, Modelling, Acidification

# 1 Introduction

Since the industrial revolution, atmospheric $CO_2$ concentrations have constantly increased (Mauna Loa Observatory: https://gml.noaa.gov/ccgg/trends/). By absorbing large amounts of $CO_2$, the global ocean acts as an important sink of anthropogenic $CO_2$. Recent estimates suggest that this absorption corresponds to roughly 25 % of annual emissions (Friedlingstein et al., 2022). During this absorption process, $CO_2$ undergoes a series of acid-base reactions that eventually lead to the formation of carbonate ions ($CO_3^{2-}$). Initially, dissolved $CO_2$ reacts with water to form carbonic acid ($H_2CO_3$) which then, dissociates into bicarbonate ($HCO_3^-$) and hydronium ($H^+$) ions. In turn, $HCO_3^-$ dissociates into $CO_3^{2-}$ and $H^+$ ions. Increased uptake of atmospheric $CO_2$ modifies this acid-base reaction chain, thus affecting the associated species concentrations, particularly of $H^+$ ions which increase significantly resulting in a decrease in seawater pH. This phenomenon, known as ocean acidification (OA), is ubiquitous as confirmed through global observations (Feely et al., 2009; Dore et al., 2009; Gonzales-Dávila et al., 2010; Bates et al., 2012). The increased uptake of atmospheric $CO_2$ not only results in lower pH but also modifies the overall carbonate equilibrium which is slowly shifting toward higher $HCO_3^-$ and $H_2CO_3$ concentrations and lower $CO_3^{2-}$ concentrations, which makes it more difficult for marine calcifiers to form their calcium carbonate shells (Orr et al., 2005).

Coastal oceans (depth < 200 m, Gattuso et al., 1998) accounts for over 10 % (0.18 to 0.45 PgC per year, Laruelle et al., 2010; 2014) of the total oceanic $CO_2$ uptake (Thomas et al., 2004) and are therefore particularly impacted by OA, generally exhibiting more pronounced localized decreases in pH (e.g., Kapsenberg et al., 2017; Luchetta et al., 2010). Nonetheless, coastal environments are highly complex mainly due to their high spatial and temporal variability, which makes their response to changes difficult to predict (Carstensen et al., 2018). Their proximity to the land means they are particularly exposed to anthropogenic pressures (run off and riverine input of anthropogenic nutrients and other chemical products, and organic matter rejects). Moreover, they are affected by strong physical forcings (e.g., tides, salinity gradients, wind induced currents) and account for about 30 % of all oceanic primary production which typically results in rich and diverse ecosystems (Gattuso et al., 1998).

The Mediterranean Sea is comparatively small and semi-enclosed; it receives nutrients through several pathways including Saharan dust depositions (Guerzoni et al., 1997) and numerous riverine inputs (e.g., Hopkins, 1992; Salat et al., 2002; Pujo-Pay et al., 2006). Considering that the Mediterranean Sea is mostly oligotrophic (Morel & Andre, 1991), these inputs are highly significant for phytoplankton growth (Revelante & Gillmartin, 1976; Ludwig et al., 2009). These features render the biogeochemistry of the Mediterranean Sea particularly complex, especially regarding the carbonate system. Several studies have investigated the carbonate system and air-sea $CO_2$ fluxes in these areas, typically using point measurements from various locations including, the Ligurian Sea (De Carlo et al., 2013; Kapsenberg et al., 2017), the Bay of Marseille (BoM; Wimart-Rousseau et al., 2020), the Gulf of Trieste (Ingrosso et al., 2016) and the Adriatic Sea (Urbini et al., 2020). Overall, these studies agree with findings by Roobaert et al. (2019) who showed that coastal systems mostly act like $CO_2$ sinks on a yearly basis, although the $CO_2$ uptake shows a significant intra-annual variability.

Most modelling approaches to investigate carbonate system variables typically employ 3D coupled physical-biogeochemical models and focus on larger coastal areas (e.g., Artioli et al., 2014; Bourgeois et al., 2016). If the focus is on smaller areas this requires higher spatial and temporal resolution to correctly represent the relevant processes (Bourgeois et al., 2016). However, higher spatial and temporal resolution often result in a significant increase of the calculation time which make more difficult the repetition of numerical experiments, an important step to better understand the global functioning of the area and its reaction to environmental forcings. A solution to avoid important calculation times is to use a dimensionless model. This type of model allows to conduct large amount of test in short amount of time. For instance, Lajaunie-Salla et al. (2021) used the dimensionless Eco3M-CarbOx model, which contains a carbonate module performing the resolution of the carbonate system based on total alkalinity (TA) and dissolved inorganic carbon (DIC). Even if the DIC, oceanic partial pressure of $CO_2$ ($pCO_2$) and total pH ($pH_T$) representations look reliable, Eco3M-CarbOx tends to minimize the range of TA variations during the year, resulting in a near constant TA (Lajaunie-Salla et al., 2021).

Here we try to provide a more realistic representation of carbonate system variables in the BoM. As a starting point, we used the concept of the dimensionless Eco3M-CarbOx model (Lajaunie-Salla et al., 2021), which aims to represent a small volume of surface water (i.e., 1 $m^3$) in the BoM. We developed a planktonic ecosystem model which contains, among others, mixotrophic organisms, modified the carbonate module described by Lajaunie-Salla et al. (2021) and added it to our newly

developed planktonic ecosystem model to obtain the Eco3M_MIX-CarbOx model (v1.0). We implemented two types of TA formulation and compared the simulation results to in situ observations to identify which formulation was capable to deliver
the more realistic results: (i) a formulation that only considers biological processes (referred to as autochthonous formulation) and (ii) a new TA formulation that depends only on salinity (referred to as allochthonous formulation). Furthermore, we simulate air-sea $CO_2$ fluxes to determine whether the BoM act as a sink or a source of $CO_2$ and provide a detailed analysis of drivers of seawater $p$CO$_2$ variations for two specific hydrodynamic processes typical for the BoM: (i) Rhône River intrusion and (ii) summer upwelling events. With this study, we aim to provide a new tool which allows to
obtain a reliable representation of the carbonate system in the simplest way as possible: by using a dimensionless configuration which is easy to use, adapt and gives results in a short amount of time.

Eco3M_MIX-CarbOx model contains both a mixotrophy compartment and a representation of the carbonate system. The model description is split in two parts: (i) a description of how the organisms and their dynamics are represented in the model, with a particular focus on mixotrophic organisms, and (ii) a more detailed description of the carbonate module and
90 the associated dynamics. While (ii) is presented here, (i) has been presented in a companion paper (Barré et al., 2023).

## 2 Materials and methods

### 2.1 Study area

The BoM is located in the NW Mediterranean Sea, in the eastern part of the Gulf of Lion near Marseille. Due to its proximity to Marseille, the second biggest city in France, and to other urbanized areas along the coast (e.g., Fos-sur-Mer and
95 Berre Lagoon to the west), the BoM is strongly affected by anthropogenic forcings which results in significant inputs of anthropogenic nutrients as ammonia ($NH_4^+$) and phosphate ($PO_4^{3-}$), chemical products, and organic matter (OM) (Millet et al., 2018) through urban rivers. Significant quantities of nutrients and freshwater are also provided by the Rhône River (Pont et al., 2002) of which the delta is located 35 km to the west of the bay. In specific wind conditions, Rhône River plume can be pushed eastwards, supplying the bay with nitrate which tend to boost the productivity of the area (Gatti et al., 2006;
Fraysse et al., 2013, 2014). In addition to these inputs, the biogeochemical functioning of the BoM is affected by various hydrodynamic processes including strong Mistral events (Yohia, 2017), upwelling events (Millot, 1990) which generally take place in specific locations: the Calanques of Marseille and the Côte Bleue, development of eddies (Schaeffer et al., 2011) and intrusions of oligotrophic water masses via the Northern Current (Barrier et al., 2016; Ross et al., 2016).

In Eco3M_MIX-CarbOx, environmental forcings are provided by in situ measurements of sea surface temperature (referred
as temperature in the following), salinity and atmospheric $p$CO$_2$ in combination with simulation data of wind speed and solar irradiance.

To evaluate our representation of carbonate system variables, we compared our model results to in situ measurements by using a carbonate parameters data set which includes TA, DIC and salinity data (https://www.seanoe.org, last access: 14

February 2023). Measurements are performed fortnightly at SOLEMIO station. $pH_T$ and $pCO_2$ are calculated based on measured TA and DIC, by using CO2SYSv3 (Sharp et al., 2020, originally developed by Lewis and Wallas (1998)) on MATLAB.

A detailed description of forcings used by the model and a map of the study area showing the location of stations where measurements were carried, and places of interests can be found in Sect. 2.1 of Barré et al. (2023) (Table 1 and Fig. 1 respectively).

## 2.2 Model description

In this study, we used the Eco3M_MIX-CarbOx model (v1.0) which was developed to represent the dynamics of the seawater carbonate system and mixotrophs in the BoM and was implemented using the Eco3M (Ecological Mechanistic and Molecular Modelling) platform (Baklouti et al., 2006a, b). Eco3M_MIX-CarbOx is a dimensionless model (0D): we consider a volume of 1 $m^3$ of surface water at SOLEMIO station, in this volume the state variables only vary over time as the model is not coupled with a hydrodynamic model. We chose to use a 0D configuration as this configuration has several advantages namely, calculation times are low (around 45 minutes in our case). It allows to make several test simulations to better understand the biogeochemical functioning of the BoM and its possible reactions to environmental forcings.

The Eco3M_MIX-CarbOx model includes seven compartments: zooplankton, mixotrophs, phytoplankton, heterotrophic bacteria, labile dissolved organic matter, detritic particulate organic matter, and dissolved inorganic matter with the following carbonate system variables: dissolved inorganic carbon (DIC), total alkalinity (TA), pH calculated on total scale ($pH_T$), and oceanic partial pressure of $CO_2$ ($pCO_2$). The carbonate system resolution required knowledge of at least two from among the four main variables of TA, DIC, $pH_T$ and $pCO_2$. As TA and DIC are conserved, a requirement to solve the source-sinks state equations, we used those variables to perform the system resolution. To provide a more realistic representation of the carbonate system, we modified the carbonate module described by Lajaunie-Salla et al. (2021) by focusing mainly on the state equations of TA and DIC, as a realistic implementation of TA and DIC state variables is crucial to obtain reliable estimates of the diagnostic variables $pH_T$, and $pCO_2$. In addition to a modified carbonate module, Eco3M_MIX-CarbOx contains a mixotroph compartment which is crucial for a reliable representation of TA and DIC, as the presence of mixotrophs affects total photosynthesis, total respiration, as well as uptake and precipitation fluxes (Mitra et al., 2014).

By using the dimensionless model Eco3M_MIX-CarbOx, we aim to represent a small volume of surface water (1 $m^3$) at the SOLEMIO station (Fig. 1). This small volume is closed which means that: (i) it does not exchange matter (i.e., nutrients, organic matter, organisms) with the water column, (ii) in our case, as we implemented a carbonate module which allows the representation of air-sea $CO_2$ fluxes, the only exchanges allowed between the volume and the atmosphere are the air-sea $CO_2$ fluxes, (iii) within the volume the matter is continuously recycled. As a result, when the water column is impacted by an hydrodynamic event which modifies its properties (i.e., which brings nutrients, organic matter, impact salinity or temperature for example), the event impacts only temperature and salinity of the volume, (Note: in the volume, TA may be impacted by a

specific event : Rhône river intrusion in the BoM, we detailed this particular case in Sect. 2.2.2 ; Fig. 1), and total N, and P are supposed to be conserved within the volume as, unlike what is done for the C pool, we do not consider any external source or sink from/to the water column or the atmosphere.

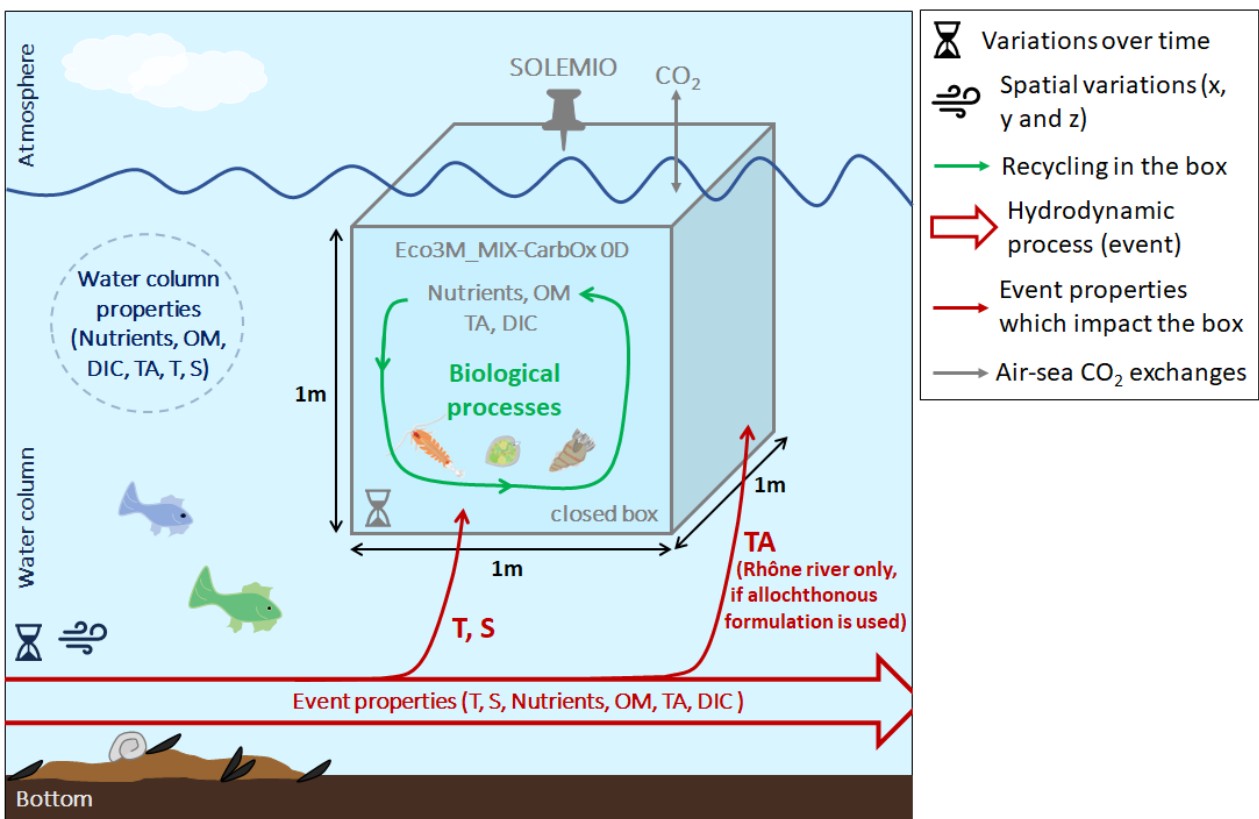

**Figure 1. Schematic representation of 0D concept used in this study with Eco3M_MIX-CarbOx. T: temperature, S: Salinity and OM: organic matter.**

In the following sections, we provide a detailed description of the carbonate system module. We also give a brief description of nutrients and organic matter representation. A detailed description of other compartments, especially of mixotrophs compartment can be found in Barré et al. (2023). Equations and parameters used by the model are also explained in this 150 previous study.

## 2.2.1 Nutrients and organic matter

As we use a dimensionless configuration, we assume that nutrients are fully the result of autochthonous biological processes. In other terms, we do not consider allochthonous inputs of nutrients (i.e., from rivers or atmosphere as instance, Fig. 1). For all the simulations, nutrients dynamics are represented by the following state equations:

$$\frac{\partial NO_3}{\partial t} = Nitrif_{NO_3}^{NH_4} - \sum_{i=1}^{2}\left(Upt_{NO_3}^{Phy_{N_i}}\right) - Upt_{NO_3}^{CM_{N_i}}$$

$$\frac{\partial NH_4}{\partial t} = Excr_{NH_4}^{COP_N} + Excr_{NH_4}^{NCM_N} + Remin_{NH_4}^{BAC_N} - \sum_{i=1}^{2}\left(Upt_{NH_4}^{Phy_{N_i}}\right) - Upt_{Nh_4}^{CM_N} - Upt_{NH_4}^{BAC_N} - Nitrif_{NH_4}^{NO_3}$$

$$\frac{\partial PO_4}{\partial t} = Excr_{PO_4}^{COP_P} + Excr_{PO_4}^{NCM_P} + Remin_{PO_4}^{BAC_P} - \sum_{i=1}^{2}\left(Upt_{PO_4}^{Phy_{P_i}}\right) - Upt_{PO_4}^{CM_P} - Upt_{PO_4}^{BAC_P}$$

(1)

where $i$ represents the number of classes of organisms. The $NO_3^-$ concentration results from nitrification and phytoplankton

and constitutive mixotrophs (CM) uptakes. $NH_4^+$ concentration results from copepods and non-constitutive mixotrophs (NCM) excretion, remineralisation by heterotrophic bacteria, phytoplankton, CM, and heterotrophic bacteria uptakes and losses from nitrification. $PO_4^{3-}$ concentration results from copepods and NCM excretion, remineralisation by heterotrophic bacteria and phytoplankton, CM and heterotrophic bacteria uptakes.

Such as nutrients dynamics, organic matter (dissolved and particulate) dynamics are only the result of autochthonous

biological processes (Eqs. 2 and 3).

$$\frac{\partial DOC}{\partial t} = \sum_{i=1}^{2}\left(Exu_{DOC}^{PHY_{C_i}}\right) + \sum_{i=1}^{2}\left(Exu_{DOC}^{MIX_{C_i}}\right) + Excr_{DOC}^{COP_C} + Mort_{DOC}^{BAC_C} - BP_{DOC}^{BAC_C}$$

$$\frac{\partial DON}{\partial t} = \sum_{i=1}^{2}\left(Exu_{DON}^{Phy_{N_i}}\right) + \sum_{i=1}^{2}\left(Exu_{DON}^{MIX_{N_i}}\right) + Mort_{DON}^{BAC_N} - Upt_{DON}^{CM_N} - Upt_{DON}^{PICO_N} - Upt_{DON}^{BAC_N}$$

$$\frac{\partial DOP}{\partial t} = \sum_{i=1}^{2}\left(Exu_{DOP}^{Phy_{P_i}}\right) + \sum_{i=1}^{2}\left(Exu_{DOP}^{MIX_{P_i}}\right) + Mort_{DOP}^{BAC_P} - Upt_{DOP}^{CM_P} - Upt_{DOP}^{PICO_P} - Upt_{DOP}^{BAC_P}$$

(2)

where $i$ represents the number of classes of organisms. The concentration of dissolved organic carbon (DOC), nitrogen (DON) and phosphorus (DOP) depends on phytoplankton and mixotrophs exudation, copepods excretion (DOC only), heterotrophic bacteria mortality (natural mortality) and CM, PICO and heterotrophic bacteria uptake.

$$\frac{\partial POC}{\partial t} = E_{POC}^{COP_C} + Predation_{POC}^{COP_C} - BP_{POC}^{BAC_C}$$

$$\frac{\partial PON}{\partial t} = E_{PON}^{COP_N} + Predation_{PON}^{COP_N} - Upt_{PON}^{BAC_N}$$

$$\frac{\partial POP}{\partial t} = E_{POP}^{COP_P} + Predation_{POP}^{COP_P} - Upt_{POP}^{BAC_P}$$

(3)

The concentration of particulate organic carbon (POC), nitrogen (PON), and phosphorus (POP) depends on copepods egestion, predation by higher trophic levels on copepods (closure terms of the model) and heterotrophic bacteria production and uptake. POM particles are large enough to sink, however, we do not consider a term to represent their removal from the

surface box by sinking. In our case, the POM, such as the DOM and nutrients, stays in the box and is constantly recycling (Fig. 1).

A detailed description and formulations of processes can be found in Barré et al. (2023). Processes notation description can be found in Appendix A (Table A1).

### 2.2.2 Carbonate system variables calculation

In Eco3M_MIX-CarbOx, we consider the four main carbonate system variables: TA, DIC, $pH_T$ and $pCO_2$. We describe their calculation by the model in this section.

In Eco3M-CarbOx, TA representation lacks variations during the year. Eco3M-CarbOx did not account for TA inputs by rivers, especially by the Rhône River which has an average alkalinity of 2885 µmol kg$^{-1}$ (Schneider et al., 2007). To remedy this shortcoming, we decided to express TA in two ways. In the first one, we considered only autochthonous TA variations

(i.e., variations of TA which result from processes which take place in the considered volume, Fig. 1). In the second one, we considered allochthonous TA variations (i.e., in the volume, TA dynamics are impacted by external contributions, Fig. 1). We then compared the outputs from each formulation to in situ data to determine which formulation delivered the more realistic results.

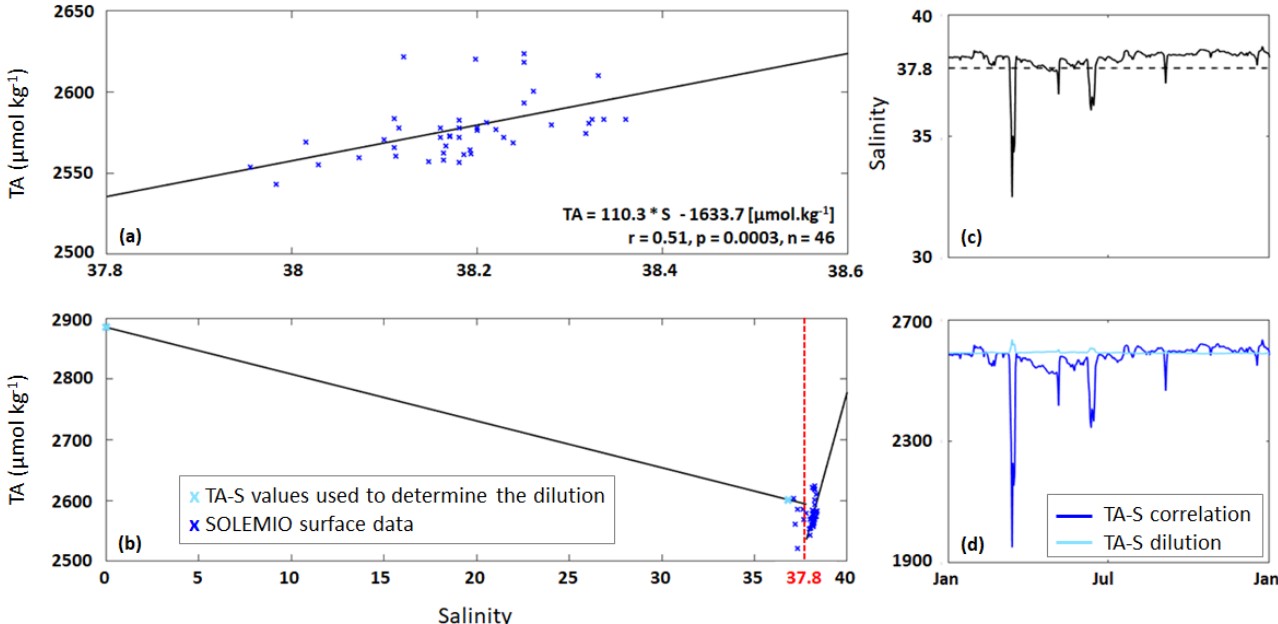

**Figure 2. (a) TA-S correlation (black line) based on SOLEMIO surface data excluding low salinities ≤ 37.8 (b) TA-S dilution (for S ≤ 37.8) and TA-S correlation (for S > 37.8) (c) Salinity data used by the model (solid line) and S = 37.8 (dashed line) (d) TA calculated from TA-S correlation (Eq. 5) and TA-S dilution (Eq. 6).**

For the autochthonous formulation, we relied on the Eco3M-CarbOx TA state equation which we modified to fit our modelled planktonic ecosystem. We first added a term of $PO_4^{3-}$ remineralisation by heterotrophic bacteria. By considering

that the uptake of one mole of $PO_4^{3-}$ by phytoplankton increases TA by one mole, and vice versa, for one mole of $PO_4^{3-}$ released during remineralisation, TA decreases by one mole (Wolf-Gladrow et al., 2007a). As a last term we included the mixotrophic uptake of nutrients. TA is calculated as follows:

$$\frac{\partial TA}{\partial t} = 2.Diss_{TA}^{CaCO_3} + \sum_{i=1}^{2}\left(Upt_{NO_3}^{Phy_{N_i}}\right) + Upt_{NO_3}^{CMN} + \sum_{i=1}^{2}\left(Upt_{PO_4}^{PHY_{P_i}}\right) + Upt_{PO_4}^{CMP} + Remin_{NH_4}^{BACN} - \sum_{i=1}^{2}\left(Upt_{NH_4}^{PHY_{N_i}}\right) -$$
$$Upt_{NH_4}^{CMN} - Remin_{PO_4}^{BACP} - 2.Prec_{TA}^{CaCO_3} - 2.Nitrif_{TA} \,,$$

(4)

where *i* represents the number of classes of organisms. Processes description can be found in Appendix A (Table A1) and formulations are available in Barré et al. (2023). In this formulation, TA only depends on biogeochemical processes (i.e., TA riverine inputs are excluded).

For the allochthonous formulation, we first determined an oceanic TA-S correlation (Eq. (5); Fig. 2a) using the measurements of carbonate system parameters at SOLEMIO station (see Sect. 2.1). We only considered the TA values associated to salinity values > 37.8 as 37.8 was used as a threshold value to identify low salinity events (LSE), associated to Rhone River plume intrusions in the BoM (Fraysse et al 2014).

$TA = 110.3 * S - 1633.7$ (µmol kg$^{-1}$)

(5)

Second, using only those TA values associated with LSE, we determined a separate TA-S formulation to quantify river water dilution (Eq. (6); Fig. 2b).

$TA = -7.7 * S + 2885$ (µmol kg$^{-1}$)

(6)

The carbonate data set did not contain sufficient LSE data to create a reliable TA-S fit. Equation (6) was therefore derived based on two TA-S data pairs: TA = 2885.0 µmol kg$^{-1}$ and S = 0, representative of water masses near Rhône River mouth (Schneider et al., 2007), and TA = 2600.6 µmol kg$^{-1}$ and S = 36.82, recorded at SOLEMIO station during a major LSE on March 15, 2017. Unlike Eq. (5), the TA-S dilution shows a negative slope typical of low salinity river water (Fig. 2b).

We implemented both TA-S formulations in our Eco3M_MIX-CarbOx model, and the formulation to be used was chosen based on the salinity : if salinity value used by the model for the time step considered ≤ 37.8, the TA-S dilution (Eq. 6) was applied; else for salinity value > 37.8 the TA-S correlation was applied (Eq. (5), Figs. 2c,d). With this method, TA only depends on salinity (i.e., biological processes are neglected).

The DIC formulation used in our Eco3M_MIX-CarbOx model is very similar to the formulation used in Eco3M-CarbOx except that we added the mixotrophic organisms' processes to our equation. As a results, DIC depends on phytoplankton,

mixotrophs, zooplankton and bacterial respiration, air-sea $CO_2$ fluxes (aeration process), dissolution of $CaCO_3$, phytoplankton and mixotrophs photosynthesis and precipitation of $CaCO_3$ (Eq. 7).

$$\frac{\partial DIC}{\partial t} = \sum_{i=1}^2 \left(Resp_{DIC}^{PHYc_i}\right) + \sum_{i=1}^2 \left(Resp_{DIC}^{MIXc_i}\right) + Resp_{DIC}^{COPc} + BR_{DIC}^{BACc} + Diss_{DIC}^{CaCO_3} - \sum_{i=1}^2 \left(Photo_{DIC}^{PHYc_i}\right) -$$

$$\sum_{i=1}^2 \left(Photo_{DIC}^{MIXc_i}\right) - Prec_{DIC}^{CaCO_3} - Aera_{DIC}$$

$$(7)$$

where $i$ represents the number of classes of organisms. Processes description can be found in Appendix A (Table A1) and formulations are available in Barré et al. (2023). As an additional modification, we use a more recent version of the gas transfer velocity calculation introduced by Wanninkhof (2014). The air-sea $CO_2$ fluxes are determined according to :

$$Aera = \frac{K_{ex}}{H} * \alpha * \left(pCO_{2,sw} - pCO_{2,atm}\right)$$

$$(8)$$

where Aera is in mmol m$^{-3}$ s$^{-1}$. $K_{ex}$ represents the gas transfer velocity (Wanninkhof, 2014) in cm h$^{-1}$, $\alpha$ the $CO_2$ solubility coefficient (Weiss, 1974) in mol L$^{-1}$ atm$^{-1}$, $pCO_{2,sw}$ the seawater $pCO_2$ modelled at the previous time step in µatm, $pCO_{2,atm}$ the atmospheric $pCO_2$ from CAV in µatm and H the magnitude of the impacted layer in meters (in Eco3M_MIX-CarbOx, H = 1 m). $K_{ex}$ is calculated using :

$$K_{ex} = 0.251 * U_{10}^2 * \left(\frac{660}{Sc}\right)^{\left(\frac{1}{2}\right)}$$

$$(9)$$

where $U_{10}$ is the wind speed in m s$^{-1}$ and Sc the Schmidt number calculated with the coefficients from Wanninkhof (2014). By convention, we will consider negative aeration values (i.e., $pCO_{2,atm} > pCO_{2,sw}$) to represent fluxes from the atmosphere into the ocean and vice versa. Furthermore, we will express air-sea $CO_2$ fluxes in the more frequently used units of mmol m$^{-2}$ per unit time.

$pH_T$ and $pCO_2$ are then obtained using the value of TA and DIC. Their calculation is detailed in Appendix B. Simulations were conducted using both formulations (autochthonous and allochthonous) for the year 2017 (Table 1, SIMC0 and SIMC1). In addition, we ran a simulation in which TA is set to a constant (TA = 2591.2 µmol kg$^{-1}$, Table 1, SIMCSTE). This simulation and its results are detailed in supplementary material.

**Table 1. Summary of simulation properties. Simulation with constant TA is detailed in supplementary material.**

| Simulation name | Total Alkalinity | Temperature | Salinity | Air-sea $CO_2$ fluxes | Biology |
|---|---|---|---|---|---|
| **SIMCSTE-Constant TA** | Constant : TA = 2591.2 µmol kg$^{-1}$ | Temperature file | Salinity file | Allowed | Yes |

| | | | | | |
|---|---|---|---|---|---|
| **SIMC0-Modelled TA (autochthonous formulation)** | Modelled | Temperature file | Salinity file | Allowed | Yes |
| **SIMC1-Calculated TA (allochthonous formulation)** | Calculated: TA = f(S) | Temperature file | Salinity file | Allowed | Yes |
| **SIMC2-Aeration effect** | Calculated: TA = f(S) | Temperature file | Salinity file | Not allowed | Yes |
| **SIMC3-Biology effect** | Calculated: TA = f(S) | Temperature file | Salinity file | Not allowed | No |
| **SIMC4-Solubility effect** | Calculated: TA = f(S) | Constant: T = 16.4°C | Constant: S = 38.1 | Not allowed | No |

### 2.3 $\Delta p$CO$_2$ decomposition

To determine the drivers of temporal variability of $p$CO$_2$, we use two types of $\Delta p$CO$_2$ decomposition. The first is based on Lovenduski et al. (2007) and evaluates TA, DIC, temperature, and salinity contributions to $p$CO$_2$ variations, while the second is based on Turi et al. (2014) and consider the contributions of biology, air-sea CO$_2$ fluxes and solubility.

#### 2.3.1 TA, DIC, T, and S drivers

Following the reasoning presented in Lovenduski et al. (2007), $p$CO$_2$ variations can be expressed as the sum of variations
generated by changes in TA, DIC, temperature and salinity as follow:

$$\Delta p\text{CO}_2 = \Delta p\text{CO}_2^{\text{TA}} + \Delta p\text{CO}_2^{\text{DIC}} + \Delta p\text{CO}_2^{\text{T}} + \Delta p\text{CO}_2^{\text{S}}$$

$$\Delta p\text{CO}_2 = \frac{\partial p\text{CO}_2}{\partial \text{TA}} * (\text{TA} - \overline{\text{TA}}) + \frac{\partial p\text{CO}_2}{\partial \text{DIC}} * (\text{DIC} - \overline{\text{DIC}}) + \frac{\partial p\text{CO}_2}{\partial \text{T}} * (\text{T} - \overline{\text{T}}) + \frac{\partial p\text{CO}_2}{\partial \text{S}} * (\text{S} - \overline{\text{S}})$$

(10)

Where $\Delta p\text{CO}_2$ is in µatm. The overbar in $\overline{\text{TA}}, \overline{\text{DIC}}, \overline{\text{T}}$, and $\overline{\text{S}}$ denotes the annual mean. Freshwater inputs can induce changes
in TA and DIC. Though, we isolate the changes of TA and DIC due to variations in freshwater inputs using the salinity-normalised TA (nTA = $\overline{\text{S}}$/S × TA) and DIC (nDIC = $\overline{\text{S}}$/S × DIC) and adding another term to regroup them. For simplicity, we only use one term to designate salinity and freshwater inputs (i.e., S+Fw term). Equation (10) can thus be rewritten as:

$$\Delta p\text{CO}_2 = \Delta p\text{CO}_2^{\text{nTA}} + \Delta p\text{CO}_2^{\text{nDIC}} + \Delta p\text{CO}_2^{\text{S+Fw}} + \Delta p\text{CO}_2^{\text{T}}$$

$$\Delta p\text{CO}_2 = \text{rS} * \frac{\partial p\text{CO}_2}{\partial \text{TA}} * (\text{nTA} - \overline{\text{nTA}}) + \text{rS} * \frac{\partial p\text{CO}_2}{\partial \text{DIC}} * (\text{nDIC} - \overline{\text{nDIC}}) + \frac{\partial p\text{CO}_2}{\partial \text{S}} * (\text{S} - \overline{\text{S}}) + \left[ \text{rSTA} * \frac{\partial p\text{CO}_2}{\partial \text{TA}} * (\text{S} - \overline{\text{S}}) + \text{rSDIC} * \right.$$
$$\left. \frac{\partial p\text{CO}_2}{\partial \text{DIC}} * (\text{S} - \overline{\text{S}}) \right] + \frac{\partial p\text{CO}_2}{\partial \text{T}} * (\text{T} - \overline{\text{T}})$$

$$\text{rS} = \frac{\text{S}}{\overline{\text{S}}} \,|\, \text{rSTA} = \frac{\overline{\text{TA}}}{\overline{\text{S}}} \,|\, \text{rSDIC} = \frac{\overline{\text{DIC}}}{\overline{\text{S}}}$$

(11)

See Appendix A in Lovenduski et al., (2007) for more details about the computation. Derivatives are obtained using the approach suggested by Sarmiento and Gruber (2006).

## 2.3.2 Contributing processes

The second decomposition (Turi et al., 2014) aims to estimate the contribution of air-sea $CO_2$ exchanges, biological processes, and solubility effects to $pCO_2$ variations:

$$\Delta pCO_2 = \Delta pCO_2^{Aeration} + \Delta pCO_2^{Biology} + \Delta pCO_2^{Solubility}$$

(12)

With the modelling approach used here, we can easily identify the individual processes and evaluate their effect on $pCO_2$ variations. Several simulations are required to identify and separate the effects of the underlying processes (see Table 1, SIMC2 to SIMC4). SIMC2 aimed to quantify the effect of aeration process on $pCO_2$ variations. Starting from SIMC1, we disabled the air-sea $CO_2$ exchanges. SIMC3 aimed to estimate the effects of biology. Using the above reasoning, we deactivated all biological processes, i.e., neither the biology nor aeration was activated in SIMC3. Finally, SIMC4 aimed to evaluate the effect of solubility on $pCO_2$ variations. This was achieving by keeping both temperature and salinity constant, using their annual means. The three terms of the Eq. (12) can be calculated as follow:

$$\Delta pCO_2^{process_i} = pCO_2^{SIMC(i-1)} - pCO_2^{SIMC(i)},$$

(13)

where $i$ is the simulation number for the process considered ($2 \leq i \leq 4$). The order in which the simulations are run is particularly important. For instance, we quantified the aeration effect (by deactivating aeration) before examining the effect of biological processes (also by deactivating them) because of the impact the biology can have on seawater $pCO_2$ and on aeration fluxes. Using similar reasoning, the impact of the biology is assessed before the impact of solubility (obtained by setting temperature and salinity constant) temperature itself has a significant effect on the biology (Lajaunie-Salla et al., 2021).

## 3 Results

### 3.1 Carbonate system variables

First, we performed an initial qualitative evaluation of Eco3M_MIX-CarbOx, comparing the output of SIMC0 (using the autochthonous TA formulation) and SIMC1 (using allochthonous TA formulation) for TA, DIC, $pCO_2$ and $pH_T$ to the corresponding SOLEMIO surface data for 2017 (Figs. 3a-d). Next, we used four statistical indicators to compare model outputs and SOLEMIO data quantitatively: the percentage bias (%BIAS), the average error (AE), the average absolute error (AAE) and the root mean square deviation (RMSD, also refer as root mean square error in the literature - RMSE). They were used with both Eco3M_MIX-CarbOx simulations, SIMC0 and SIMC1 (Table 1), and the reference Eco3M-CarbOx simulation (Lajaunie-Salla et al., 2021). By comparing the statistical indicators obtained for SIMC0, SIMC1 and Eco3M-

CarbOx we obtained an indication of how changes in the carbonate variables formulation affected the results. Statistical

indicators calculation is detailed in Appendix C.

The different TA formulations yielded very different model outputs for DIC, $pCO_2$ and $pH_T$ (Figs. 3f-h). TA observations varied between 2560.8 and 2623.9 µmol kg$^{-1}$, with no apparent seasonal pattern (Fig. 3a). This variability is successfully represented by SIMC1, but not SIMC0 (SIMC1 range: 2540 to 2635 µmol kg$^{-1}$). SIMC0 produces TA values that show a gradual and near-linear decrease from 2578 µmol kg$^{-1}$ in early January to 2572 µmol kg$^{-1}$ at the end of the year. The

differences between SIMC0 and SIMC1 are most pronounced between August and December where SIMC1 delivers systematically higher TA values compared to SIMC0 (Fig. 3e).

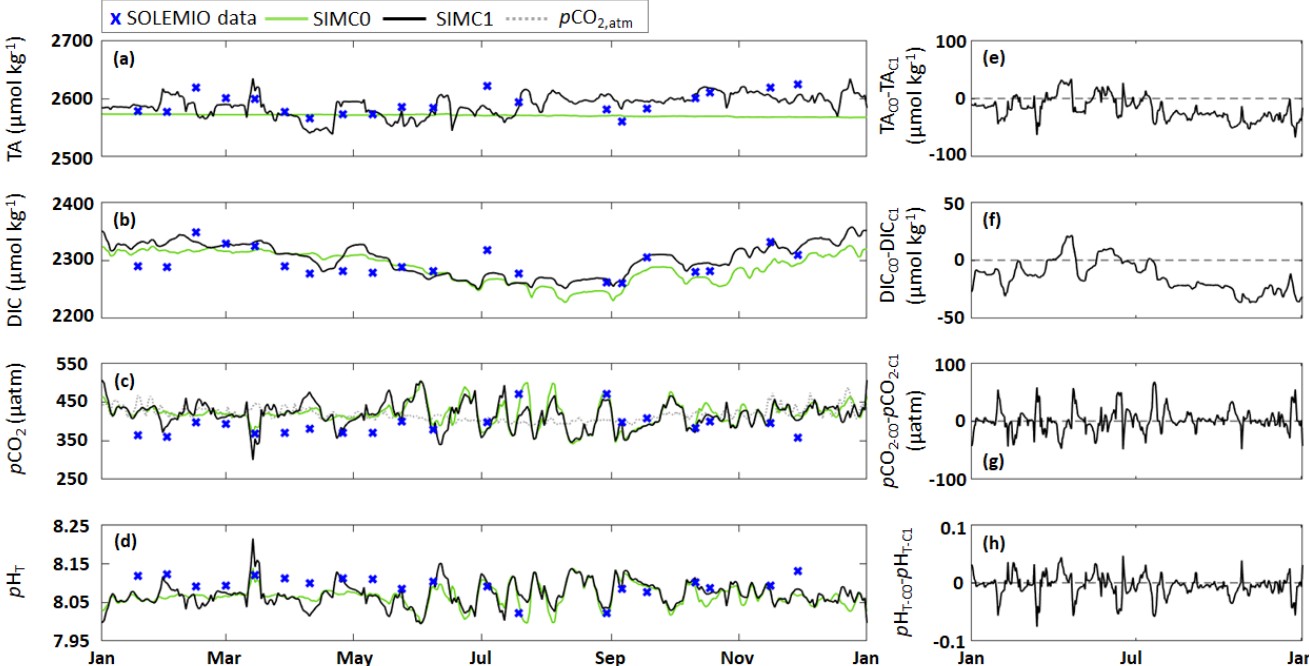

**Figure 3. (a-d) Comparison of model outputs from the SIMC0 (autochthonous formulation) and SIMC1 (allochthonous formulation), model runs showing daily averages of (a) TA, (b) DIC, (c) seawater $pCO_2$ and CAV atmospheric $pCO_2$ and (d) $pH_T$.**
**(e-h) Differences between SIMC0 and SIMC1 outputs for each variable (VARC0 – VARC1).**

With regard to DIC, both SIMC0 and SIMC1 are capable of reproducing the seasonal variability present in the in situ data. From November to April, DIC has higher values (around 2320 µmol kg$^{-1}$ in both simulations), with lower values during the rest of the year (both have a minimum August, SIMC0: 2234 µmol kg$^{-1}$ and SIMC1: 2254 µmol kg$^{-1}$; Fig. 3b). At the beginning of the year, SIMC1 seems to be closer to the observations than SIMC0 which shows fewer variations (e.g., SIMC1

appears to be better at reproducing the decrease visible at the end of April). Differences between SIMC0 and SIMC1 for DIC are similar to those observed for TA (Figs. 3e,f) although in absolute terms, they are only about half of what we observed for TA. Nevertheless, these results show that the choice of the TA formulation strongly affects the DIC model results (Fig. 3f).

The in situ $pCO_2$ data exhibits strong variations throughout the year, especially from May to November which are well represented in both simulations (Fig. 3c). Between January and April, both simulations overestimate the in situ $pCO_2$ values: while the simulations both predict $pCO_2$ values close to the CAV atmospheric $pCO_2$ of about 415 µatm, $pCO_2$ observed at SOLEMIO is lower indicating under-saturation. For both simulations a strong decrease of $pCO_2$ is modelled on March 15[th], in response to a Rhône River intrusion in the BoM. This event is particularly marked in the SIMC1 model results which show a decrease from 450 to 300 µatm (compared to a decrease from 415 to 358 µatm with SIMC0). While this decrease is also visible in the in situ data it is more moderate (392 to 367 µatm).

**Table 2. Comparing the different model results to surface observations at SOLEMIO station for TA, DIC, seawater $pCO_2$, and $pH_T$. N represents the number of observations. Mean, SD, AE, AAE and RMSD are in the same unit than the considered variable, i.e.: µmol kg$^{-1}$ for TA and DIC and µatm for $pCO_2$. %BIAS is without unit.**

|  |  | TA | DIC | $pCO_2$ | $pH_T$ |
|---|---|---|---|---|---|
| **N** | Observations | 20 | 20 | 20 | 20 |
| **Mean ± SD** | Observations | 2591.2 ± 19.4 | 2294.9 ± 24.0 | 391.0 ± 31.0 | 8.09 ± 0.030 |
| **Mean ± SD** | SIMC0 | 2576.1 ± 1.5 | 2293.6 ± 25.1 | 413.5 ± 16.5 | 8.07 ± 0.015 |
|  | SIMC1 | 2588.6 ± 16.4 | 2301.1 ± 24.5 | 409.1 ± 21.4 | 8.07 ± 0.020 |
|  | CarbOx | 2574.5 ± 3.6 | 2292.5 ± 26.0 | 413.9 ± 15.9 | 8.07 ± 0.010 |
| **%BIAS** | SIMC0 | 0.58 | 0.05 | -5.75 | 0.29 |
|  | SIMC1 | 0.09 | -0.27 | -4.61 | 0.21 |
|  | CarbOx | 0.64 | 0.1 | -5.86 | 0.29 |
| **AE** | SIMC0 | 15.12 | 1.25 | -22.5 | 0.02 |
|  | SIMC1 | 2.57 | -6.2 | -18.02 | 0.02 |
|  | CarbOx | 16.7 | 2.4 | -22.9 | 0.02 |
| **AAE** | SIMC0 | 18.7 | 20.4 | 35.9 | 0.03 |
|  | SIMC1 | 16.3 | 17.2 | 34.7 | 0.03 |
|  | CarbOx | 20.1 | 21.2 | 35.3 | 0.03 |
| **RMSD** | SIMC0 | 24.90 | 24.26 | 38.75 | 0.04 |
|  | SIMC1 | 20.03 | 21.83 | 40.27 | 0.04 |
|  | CarbOx | 26.56 | 24.90 | 38.29 | 0.04 |

Regarding $pH_T$, both simulations produced similar dynamics as for $pCO_2$ (Figs. 3d vs 3c). Both simulations deliver good representations of the observed $pH_T$ variations between May and November while from January to April both simulations underestimate the in situ (in situ: 8.12 vs simulations: 8.07). The Rhône River intrusion is also visible in the $pH_T$ data which

exhibits a sudden increase. While both simulations show this increase, it is more pronounced in the SIMC1 results (increase from 8.04 to 8.21) compared to SIMC0 (8.07 to 8.14), but in both cases larger than in the observations (8.09 to 8.12).

The differences between both simulations for $pCO_2$ and $pH_T$ do not exhibit any noticeable trend (Figs. 3g, h). However,

looking at the annual average, SIMC1 produces lower (higher) $pCO_2$ ($pH_T$) values compared to SIMC0 with a mean difference of 2.3 µatm ($-5 \times 10^{-3}$). Moreover, for both variables, the differences between SIMC0 and SIMC1 are more pronounced at the beginning of the year.

For statistical indicators, %BIAS values are systematically lower than 10 %, with the highest values obtained for $pCO_2$ with ~6 % while the remaining variables had values < 1 %. Similarly, $pCO_2$ had the highest RMSD, AAE and AE which suggests

that this parameter is not as well represented in the model as the other variables. Furthermore, SIMC1 produced the best TA representation resulting in the lowest values for %BIAS, AE, AAE and RMSD (Table 2). Moreover, SIMC1 produced an annual mean-TA that was closest to the observations. While the SIMC0 and Eco3m-CarbOx results are fairly similar. SIMC0 produced a slightly better representation of TA compared to Eco3m-CarbOx (%BIAS, AE, AAE and RMSD slightly lower). For $pH_T$, SIMC1 outperformed SIMC0 based on %BIAS (Table 2), however, AE, AAE and RMSD values are

similar for the three simulations. We then performed the calculation of statistical indicators on $H^+$ concentration as, according to some authors (Kwiatkowski & Orr, 2018), comparing $H^+$ concentrations is a better practice than comparing pH. Results are available in Appendix C. Based on Table C1, SIMC1 also outperformed SIMC0 based on AE and AAE. For studying DIC and $pCO_2$, the situation is less clear as the simulations performed differently for different indicators, making it difficult to pick a clear winner. Still SIMC1 shows the best AAE and RMSD values for DIC, and the best %BIAS, AE, and

AAE for $pCO_2$. In conclusion, SIMC1 shows the best overall indicator values for the examined variables (more specifically, it outperformed the other simulations in 13 of 20 indicator comparisons when including $H^+$ concentrations comparison).

## 3.2 Air-sea $CO_2$ fluxes

Throughout 2017 temperature varied from 13.3 to 25.9 °C (Fig. 4a) with the highest variability visible during the summer upwelling period (SUP). Apart from four low salinity events in March, May, June, and September (all corresponding to the

Rhône River intrusions) the salinity remained close to its mean value of 38.1 (Fig. 4a).

Wind speed was highly variable with several strong gusts, especially during winter when wind speeds often exceeded 10 m s$^{-1}$ (Fig. 4b). Wind speed tends to be lower during summer and SUP, although these periods also show numerous strong wind events (> 10 m s$^{-1}$).

The sea-air $pCO_2$ difference exhibits the same seasonality as temperature, with high positive values during summer while

oscillating about zero during the rest of the year. In general, the sea-air $pCO_2$ difference combines the patterns from temperature, salinity and wind speed which are the main underlying forcings. The local minimum in March, corresponds to an extremely low salinity event (Fig. 4c). However, during the SUP the sea-air $pCO_2$ difference is mostly driven by temperature as seen by the high variability between May and October which coincide with the largest temperature variations.

In contrast, air-sea $CO_2$ fluxes do not show any seasonality, with values oscillating about zero throughout the year (Fig. 4d) yielding an integrated total of -0.21 mmol m$^{-2}$ per year. Maximum positive values are obtained from November to March when wind speeds are highest. Extreme negative value (-13 mmol m$^{-2}$ per day) can be seen in July coinciding with high wind speed, negative sea-air $pCO_2$ difference and a significant drop in temperature.

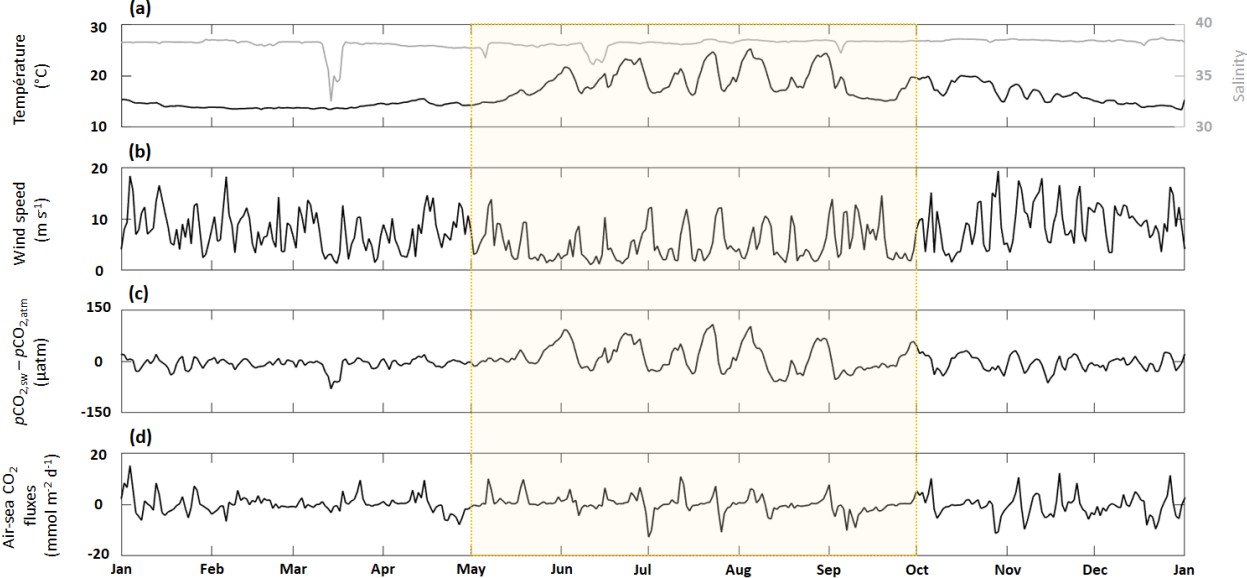

**Figure 4. Time series of (a, e) in situ daily average sea surface temperature (black line) and salinity (grey line) (b, f) SIMC1 daily average wind speed (c, g) the difference between SIMC1 daily average seawater $pCO_2$ and in situ daily average atmospheric $pCO_2$ (d, h) SIMC1 daily average air-sea $CO_2$ fluxes (aeration process). The summer upwelling period (from 1 May to 1 October) is highlighted in yellow.**

### 3.3 Main drivers of $pCO_2$ dynamics

### 3.3.1 Annual scale

Following the approach from Lovenduski et al. (2007), we used temperature (Fig. 5a), as well as salinity (S), freshwater inputs (Fw), nTA and nDIC (Fig. 5b) contributions to identify the underlying dynamics in the observed $pCO_2$ variations (Fig. 5c). Seasonal variations in temperature (Fig. 5a) produce seasonal anomalies in $pCO_2$ with negative anomalies dominating from November to May and mostly positive anomalies throughout the remainder of the year (Fig. 5d). Anomalies generated by S+Fw do not exhibit any seasonality but remain close to zero throughout the year, unless there is an LSE, during which the anomalies turn negative (-101 µatm, -30 µatm, -40 µatm and -20 µatm for the four LSE). Anomalies generated by nDIC show the opposite seasonal trend compared to the anomalies generated by temperature, i.e., from November to May the nDIC-generated anomalies are positive and negative during the rest of the year. The four LSE are also clearly visible in the nDIC-generated anomalies which exhibit sharp increases (increase of 506 µatm, 253 µatm, 243 µatm and 152 µatm respectively). Also, nTA does not produce any seasonality in the anomalies but exhibits sharp decrease during the four LSE (decrease of 548 µatm, 242 µatm, 239 µatm and 90 µatm respectively).

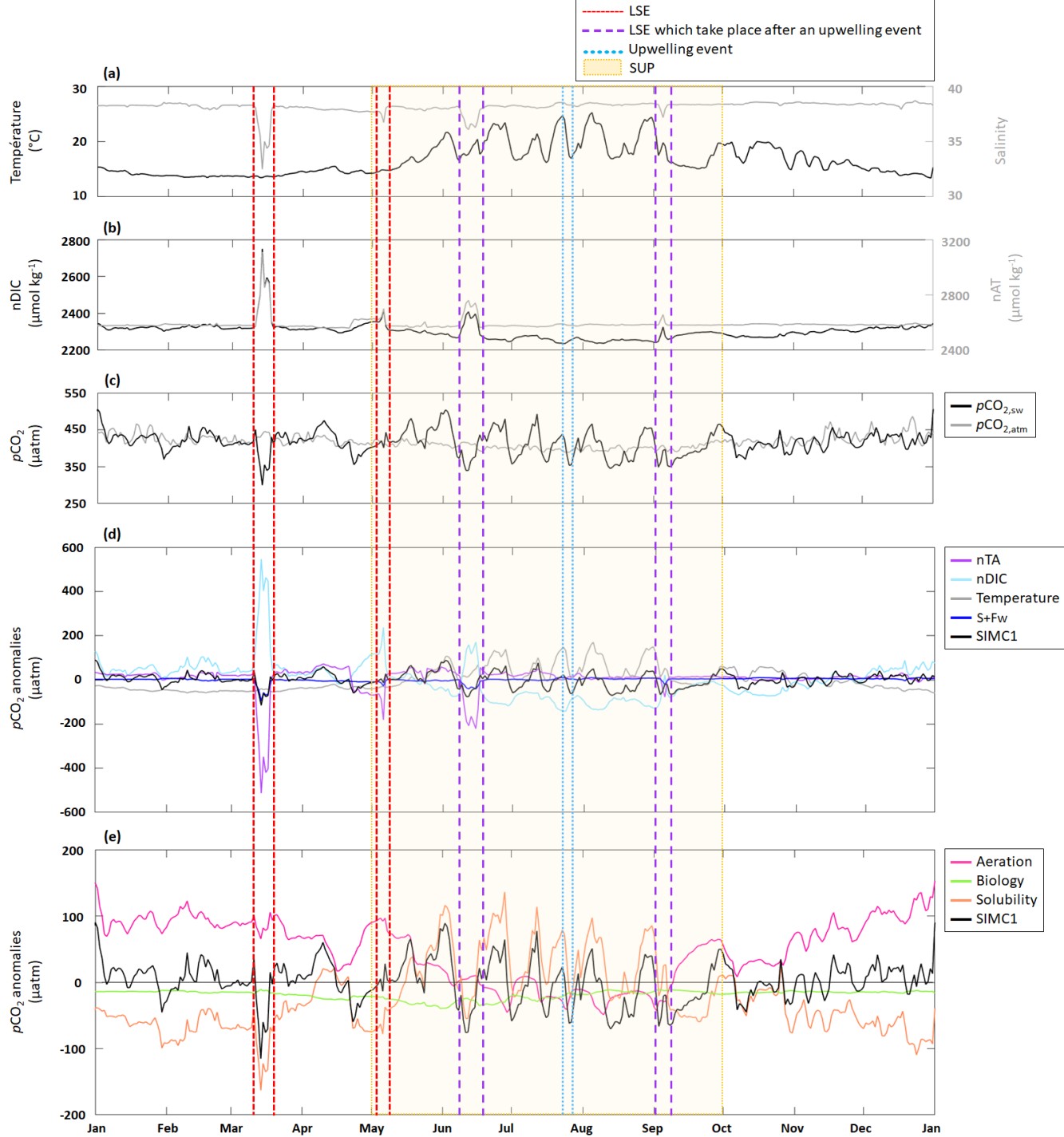

**Figure 5. Time series for 2017 of daily average (a) in situ temperature and salinity (b) modelled nDIC and nTA (c) modelled seawater and in situ atmospheric $p$CO₂ (d) $p$CO₂ anomalies generated by DIC, TA, S+Fw and temperature based on the approach in Lovenduski et al. (2007) (Note: the dark blue line is sometimes obscured by the black line, especially in March. An enlargement**

**of the panel d is available in Appendix D) (e, j) *p*CO₂ anomalies generated by aeration, solubility, and biological processes based on the approach in Turi et al. (2014). LSE and an upwelling event have been highlighted. The summer upwelling period (SUP) is indicated by yellow shading.**

Following the approach by Turi et al. (2014), we examined the effects of aeration, biological processes, and solubility on $p$CO₂ variability (Fig. 5e). Aeration produced anomalies very similar to those observed for nDIC (Fig. 5d): positive from

November to May and negative during the rest of the year. Since CO₂ solubility is controlled by temperature and salinity, solubility-generated anomalies essentially follow the trends and seasonality seen in temperature and S+Fw-generated anomalies (Fig. 5d): negative from November to May and mostly positive during the rest of the year (mean of +9.2 µatm).

The four LSE are also visible in the solubility-generated anomalies generating strong decreases (Fig. 5e). However, only two LSE are easily identifiable (15 March with a drop from -41 µatm to -163 µatm and 6 May with a drop from 8 µatm to -75

µatm) while the other two appear to be obscured by temperature-related counter-movements. Since aeration- and solubility-generated anomalies show opposite seasonality, they partly cancel each other out. While aeration seems to dominate from November to May, (apart from LSE), solubility appears to dominate from May to November and during LSE. Biological processes are never the dominant driver of $p$CO₂ variations as they are systematically smaller (by a factor of 2 to 3) than aeration and solubility-generated anomalies (Fig. 5e). Biology-induced anomalies are always negative, providing evidence

that biological processes always decrease $p$CO₂.

### 3.3.2 During the summer upwelling period (SUP)

The SUP is characterized by significant temperature variations (Fig. 5a) due to periodic upwelling events. During the 2017 SUP, there were three LSE which will be excluded here as we discuss them in the following section. nTA is nearly constant during the SUP while nDIC shows marked variations (Fig. 5b) that are directly linked to variations in DIC (see Sect. 3.1).

$p$CO₂ is also highly variable during the SUP (Fig. 5c). Using the approach from Lovenduski et al. (2007) (Fig. 5d), the SUP is characterized by a strong contribution of temperature which shows strong positive anomalies (maximum of 170 µatm reached on 5 August), and nDIC which shows strong negative anomalies (minimum of -142 µatm reached on 24 July). S+Fw and nTA do not represent significant drivers with anomalies remaining close to zero. Using the approach in Turi et al. (2014) (Fig. 5e), we can see that solubility is a major driver producing large amplitude variations in the $p$CO₂ anomalies connected

to similar variations in temperature (a drop in temperature causes the anomaly to change from positive to negative and vice versa) (Fig. 5a). Aeration, which mostly generates negative anomalies, counteracts solubility. During the SUP, we also observed an increase of biological processes contribution since associated anomalies further decrease at the beginning of the period (from -22 µatm on 1 May to -40 µatm on 31 May).

Focusing on the upwelling event that took place between 23-27 July, we observe a sharp decrease in temperature (from 24.6

°C to 16.9 °C; Fig. 5a), no variation in nTA, and a slight increase in nDIC (from 2242 µmol kg⁻¹ to 2269 µmol kg⁻¹; Fig. 5b). The event is also associated with a strong $p$CO₂ decrease (from 438 µatm to 353 µatm; Fig. 5c). Using the approach in Lovenduski et al., (2007) we observed a decrease of the temperature-generated anomaly (from 148 µatm at the beginning of

the event to 5 µatm at the peak of the event). At the same time, the nDIC-generated anomaly become less negative (from -142 µatm at the beginning of the event to -79 µatm at the peak of the event). Neither nTA nor S+Fw seem to have any

significant impact on $pCO_2$ anomalies. Using the approach in Turi et al. (2014) (Fig. 5e), the upwelling event is characterized by decrease of solubility-generated anomalies (from 79 µatm at the beginning of the event to -24 µatm at the end of the event). Anomalies generated by aeration and biological processes tend to respectively become positive and less negative at the end of the event (aeration: -45 µatm to 3 µatm ; biological processes: -30 µatm to -20 µatm).

### 3.3.3 During a low salinity event (LSE)

There were four LSE during 2017: on 15 March, 6 May, 15 June, and 5 September. All four LSE show similar patterns, namely a strong decrease in salinity (Fig. 5a) which in turn leads to an increase in both nTA and nDIC (Fig. 5, Table 3). Apart from the 5 September LSE which shows an increase in $pCO_2$, the remaining LSE coincide with significant $pCO_2$ decreases (Fig. 5c, Table 3).

When using the approach of Lovenduski et al., (2007), LSE that do not take place immediately after an upwelling event (i.e.,

15 March and 6 May) exhibit similar combinations of driver contributions, e.g., nTA and S+Fw create strong negative anomalies in both LSE (with combined (nTA+S+Fw) contributions of: -614 µatm on 15 March and -211 µatm on 6 May), which are partially cancelled out by nDIC opposite contribution (547 µatm on 15 March and 235 µatm on 6 May). While temperature-generated anomalies showed no change during either event, it is still negative and by adding its effect to those obtained for nTA and S+Fw, we obtain a combined effect of  -656 µatm on 15 March and -241 µatm on 6 May.

**Table 3. Change in S, nTA, nDIC and $pCO_2$ from before to during a LSE.**

| | S | nTA (µmol kg$^{-1}$) | nDIC (µmol kg$^{-1}$) | $pCO_2$ (µatm) |
|---|---|---|---|---|
| **15 March** | 38.3 to 32.5 | 2570 to 3110 | 2320 to 2750 | 450 to 300 |
| **6 May** | 37.8 to 36.7 | 2560 to 2700 | 2308 to 2420 | 420 to 401 |
| **15 June** | 38.1 to 36.0 | 2580 to 2760 | 2273 to 2409 | 504 to 340 |
| **5 September** | 38.3 to 37.1 | 2583 to 2658 | 2241 to 2327 | 348 to 396 |

LSE that take place immediately after a summer upwelling event (i.e., 15 June and 5 September), show similar variations of salinity, nTA, nDIC and $pCO_2$ but also show an increase of temperature (from 16.5 °C to 20.5 °C on 15 June and 17.5 °C to 19.8 °C on 5 September; Fig. 5a). Also, the factors driving the anomalies are similar to those for the non-upwelling related

LSE discussed in the previous paragraph. The combined nTA and S+Fw anomalies (-260 µatm on 15 June and -108 µatm on 5 September) are partially compensated by nDIC contribution (171 µatm and 22 µatm respectively). Unlike for the previous events, we do see a significant temperature effect for the upwelling-related LSE: temperature-generated anomalies are positive (45 µatm on 15 June and 53 µatm on 5 September) and support nDIC contribution.

When following Turi et al. (2014) (Fig. 5e), all LSE, with the exception of the 5 September LSE, are characterized by strong
negative solubility-generated anomalies (-163 µatm on 15 March, -78 µatm on 6 May and -55 µatm on 15 June) partially
compensated by positive aeration-generated anomalies (65 µatm, 97 µatm and 8 µatm respectively). The odd one out which
take place on 5 September shows positive solubility-generated anomaly (27 µatm) and negative aeration-generated anomaly
(-30 µatm). In all the four LSE, biological processes did not have any significant impact on $pCO_2$ variations (anomalies
generated by biological processes are 2 to 3 times lower than those generated by aeration or solubility).

## 4 Discussion

### 4.1 Impact of Rhône River inputs on TA variations

Due to its location near the Rhône River mouth, the BoM is particularly affected by freshwater inputs. In 2017, there were
four LSE in the BoM. Apart from being low in salinity, the Rhone River water entering the BoM also contains organic
matter, nutrients, DIC and alkalinity, with a mean TA of 2885 µmol kg$^{-1}$ (Schneider et al., 2007). This input adds up to the
effect of biological processes. We have seen that TA measurements in the BoM exhibit significant variability throughout the
year (Fig. 3a), although no obvious seasonality. By considering autochthonous (i.e., dependant on biological processes only)
and allochthonous (i.e., dependant on rivers inputs only) formulations of TA, we were able to isolate the effects of the
biology and riverine inputs and quantify their relative importance for the TA variations seen in the BoM.

With the autochthonous formulation, TA remained fairly constant throughout the year, which is similar to the results
obtained by Lajaunie-Salla et al. (2021). In contrast, the allochthonous formulation produced a much high variability in TA
that was close to in situ observations. Several authors suggested that biological processes could have a large effect on TA
dynamics in coastal areas (Krumins et al., 2013; Gustafsson et al., 2014). These findings are not confirmed by our model
results where changes in TA due to biology did not exceed 5 µmol kg$^{-1}$ (Fig. 3a), which is insignificant compared to the
changes attributed to other drivers, including riverine inputs. This suggests that TA variations in the BoM are mostly driven
by allochthonous factors. The importance of allochthonous contributions to TA variations have already been highlighted by
several authors at the Mediterranean Sea scale (Copin-Montegut, 1993; Schneider et al., 2007; Hassoun et al., 2015). Other
important drivers in the Mediterranean include TA exchanges with the Atlantic Ocean and Black Sea, as well as TA inputs
from sediments and rain. For the particular location of our study area, we only considered river contributions. Having
neglected other allochthonous drivers seems to be justified by the results which showed a close match to observations and a
generally better representation of the other carbonate system variables since DIC, $pCO_2$ and pH$_T$ are all closely related to TA
(Fig. 3 and Table 2). Several studies of TA variations in the Mediterranean Sea have been conducted at the sub-basins scale
yielding different TA-S correlation for different study areas (Cossarini et al., 2015; Hassoun et al., 2015). For instance, the
correlation proposed for the north-western Mediterranean Sea, suggests that local TA dynamics are mainly controlled by
evaporation. We did not include this in our study as the BoM is strongly impacted by the Rhône River. By focussing on a
smaller area, we could provide a TA formulation that represents this particular part of the Mediterranean very well.

While our results seem to provide a realistic representation of TA dynamics in the BoM, we could have included other factors such as sediments, which have been shown to be important for TA dynamics, particularly in coastal areas (Brenner et al., 2016; Gustafsson et al., 2014). We plan to add TA supplies by sediments in our future work. Moreover, from a more conceptual perspective, the use of the present TA allochthonous formulation allowed to manage two cases of salinity, namely $S \leq 37.8$ and $S > 37.8$ with two different equations (Eqs. 5 and 6), however the switch from one to another, in other words crossing the thresholds value, may lead to instabilities in TA representation. A solution to better manage the threshold crossing case is to represent the Rhone River inputs more realistically. Here, we used two TA-S couples (TA and S at the mouth of the Rhône River and TA and S measured at SOLEMIO during the most significant Rhone River intrusion event) to obtain the dilution formulation. With this method, we do not take into account the seasonality of TA in the Rhône River which can bring significant variations (Figure S2 and Table S3 of supplementary material).

## 4.2 Impact of hydrodynamic processes on $p$CO$_2$ variations

### 4.2.1 Low salinity events (LSE)

The four LSE observed in 2017 had several common characteristics: a salinity decrease (Fig. 5a) and apparent nTA and nDIC increases (Fig. 5b). Three of the four LSE resulted in a $p$CO$_2$ decrease (15 March, 6 May, and 15 June, Fig. 5c). Rhône River intrusion events are often associated with a $p$CO$_2$ decrease since the introduced nutrients stimulate phytoplanktonic growth (Fraysse et al., 2014; Lajaunie-Salla et al., 2021). However, in our case, the decrease of $p$CO$_2$ observed on 15 March, 6 May and 15 June was entirely caused by nTA and solubility effects (Figs. 5d,e). Generally, a TA increase is associated with a $p$CO$_2$ decrease that is proportional to the buffering state of the considered water mass (for high TA:DIC ratios, changes in $p$CO$_2$ are lower since the water mass is well buffered; Middelburg et al., 2020), which explains the negative $p$CO$_2$ anomalies associated with these three LSE. Solubility depends on both salinity and temperature. Depending on the size and the duration of the Rhône River intrusion, salinity effect to solubility can vary. When salinity is decreasing, the solubility of CO$_2$ in seawater also decreases, which results in a decrease in $p$CO$_2$ (Middelburg, 2019). The effects of temperature to solubility vary throughout the year. For instance, during the 15 March and 6 May LSE, temperatures were low and fairly constant (Fig. 5a) and therefore only contributed a small amount to the negative anomaly (Fig. 5d). In contrast, the 15 June, temperature cause a positive $p$CO$_2$ anomaly (Fig. 5d). This difference can be explained by the fact that the 15 June LSE took place right after an upwelling event, probably facilitated by the Marseille eddy presence near the BoM, which tend to be observed just after Mistral events (Fraysse et al., 2014). While the temperature dropped as a result of the upwelling, once the event was over the temperature increased again which caused the observed positive $p$CO$_2$ anomaly. Despite this positive temperature-related anomaly, the overall anomaly remained negative due to the strong effects of salinity and nTA during the LSE (Fig. 5c).

The 5 September LSE was associated with a $p$CO$_2$ increase (Fig. 5c), caused by nDIC and solubility effects (Figs. 5d,e): as salinity and nTA contributions remain weak, they are completely counterbalanced by nDIC and temperature contribution,

resulting in an increase of $p$CO$_2$. During September 5th LSE, observed salinity and temperature showed opposite patterns: the decrease of salinity is associated to an increase of temperature, and the increase of salinity after the peak of the LSE, is

associated to a temperature decrease (Fig. 5a). Unlike for the 15 June LSE, the temperature increase seen during the 5 September event was not caused by the end of the upwelling event preceding as the temperature was decreasing right after the LSE peak (Fig. 5a). We assume that this temperature increase was instead caused by the intruding Rhône River water, which brought about the observed $p$CO$_2$ increase ($p$CO$_2$ increases exponentially with temperature; Middelburg, 2019).

In all four LSE, biological processes did not have any significant impact on $p$CO$_2$ variations (Fig. 5e). To interpret this

result, it is important to consider the assumptions used by Eco3M_MIX-CarbOx (Sect. 2.2). Rhône River intrusion can significantly modify the biogeochemistry of the bay as they are typically associated with temperature and salinity changes and TA, DIC and nutrients inputs (Gatti et al., 2006; Fraysse et al., 2014; Lajaunie-Salla et al., 2021). Due to its 0D configuration, Eco3M_MIX-CarbOx only represents temperature and salinity changes and TA inputs (only if the allochthonous formulation is used for the latter, Fig. 1). For the studied events, linking measured surface salinity to measured

DIC (Appendix E) showed that the four events are not systematically associated to a DIC increase at SOLEMIO even though the Rhône River mouth DIC value (2877 µmol kg$^{-1}$, value calculated by using TA and pH from Schneider et al. (2007) and Aucour et al. (1999) respectively) is much higher than the mean value at the station (2294.9 µmol kg$^{-1}$). Based on this observation, we can assume that, for DIC, the riverine signal is quickly lost when moving away from the Rhone River mouth and is not reaching SOLEMIO station. Contrary to TA which is mainly affected by Rhone River inputs in the area, DIC is

impacted by air-sea CO$_2$ exchanges and biological processes which can explain this pattern. However, for more realism and as these inputs could affect $p$CO$_2$ variations by increasing the nDIC contribution, considering them could be an interesting addition to the present configuration. Moreover, linking measured surface salinity to measured nutrient concentrations (Appendix E) showed that only the first and last events (15 March and 5 September respectively) have an impact on nutrient concentrations at SOLEMIO with the first event being the most significant. Lajaunie-Salla et al. (2021) showed that these

nutrient inputs led to an increase in chlorophyll concentration. This phytoplankton growth leads to further decrease in $p$CO$_2$, which means that by neglecting them we possibly underestimated the importance of biological processes, and especially of autotrophic processes during these Rhône River intrusions.

### 4.2.2 Summer upwelling period (SUP)

During the SUP, regardless of whether there is an LSE, $p$CO$_2$ variations mostly depend on temperature and nDIC which tend

to produce anomalies of opposite signs (Fig. 5d). Temperature was highly variable during the SUP due to the succession of upwelling events which explains its significant contribution to $p$CO$_2$ variations. nDIC contribution can be defined as the sum of aeration and biological processes contributions. During the SUP, biological processes represent 29 % of DIC variations (with 14 % attributed to primary production and 15 % to respiration; results not shown). The remaining 71 % are contributions by aeration. While the contribution of aeration decreased during summer, this decrease was compensated by a

9 % increase in the contribution by biological processes (Fig. 5e). The maximum negative anomaly generated by biological

processes occurred at the beginning of the SUP, on 31 May (Fig. 5e), evidence that biological processes and more precisely autotrophic processes are enhanced during late spring. This feature is explained by the change in organisms' limitations. At the end of spring, organisms are less limited by temperature and light. Nevertheless, the overall contribution of biological processes was low compared to aeration and temperature ones. This agrees with observations by Wimart-Rousseau et al. (2020) and Lajaunie-Salla et al. (2021) who showed that, $pCO_2$ variations and associated $CO_2$ fluxes are mostly driven by temperature in the BoM.

We showed that upwelling events were associated with strong decreases in $pCO_2$ (Fig. 5c) mostly as a result of temperature changes. The associated decrease in temperature further decreased $pCO_2$. This feature is only observed during upwelling events in summer when both temperatures and $pCO_2$ are high (Figs. 5a,c), stressing the importance of upwelling events for these variables. During upwelling events, aeration-generated anomalies change sign and become positive (Fig. 5e). The observed decrease in temperature resulted in a decrease in seawater $pCO_2$ to below atmospheric levels, thereby facilitating the absorption of atmospheric $CO_2$ which caused the reversal sign of aeration-generated anomaly. During upwelling events, the contribution by biological processes is low compared to temperature and aeration which both varied significantly (Fig. 5e). While upwelling events only occur at very specific locations (Côte Bleue and Calanques de Marseille) in our study area, they impact the temperature of the entire BoM (Pairaud et al., 2011). Although upwelling events also bring nutrients and DIC to the surface. In Eco3M_MIX-CarbOx, these effects are not considered, and upwelling events are only represented through temperature decrease in the volume. During the SUP, by linking surface temperature measurements and surface DIC and nutrients concentration measurements at SOLEMIO (Appendix E), we showed that: (i) among the upwelling events, only two (at the beginning of July and mid-September) are linked to a noticeable DIC variation, and (ii) surface nutrients concentration dynamics seems only slightly affected by upwelling events (nutrients concentrations remain close to 0 for most of the time) explained by the fact that, when the upwelling takes place, nutrients which are upwelled are quickly consumed by the phytoplankton present in the area, then not systematically reaching the station. Even though the effect of upwelling events on DIC and nutrients concentration seems limited at SOLEMIO station, it may be interesting to consider them for more realism as, the temporal coverage of SOLEMIO measurements remains low (15 days) and we cannot exclude the fact that an impact can be observed but not caught by the measurements. Indeed, even if low, a nutrient input can promote primary production (Fraysse et al., 2013), then increase the contribution of biological processes (especially of autotrophic processes) resulting in a stronger decrease in $pCO_2$ while DIC inputs would increase the importance of nDIC thereby reducing the decrease of $pCO_2$ associated with these events.

### 4.3 Air-sea $CO_2$ fluxes

We have shown that air-sea $CO_2$ fluxes oscillated between -13 and 15 mmol m$^{-2}$ per day (Fig. 4d) which is a range similar to the one obtained by Wimart-Rousseau et al. (2020) (-15 and 10 mmol m$^{-2}$ per day) suggesting that our model correctly represents the range of variations of air-sea $CO_2$ daily fluxes values during the year. $CO_2$ sinks associated to upwelling events (Lajaunie-Salla et al., 2021) are reproduced by our model. By calculating the daily mean value of air-sea $CO_2$ fluxes

during the SUP, we obtained a positive value of 0.15 mmol m$^{-2}$ per day (or 24.2 mmol m$^{-2}$ for the entire SUP). To examine
this result in more detail, we performed a sensitivity analysis of our air-sea $CO_2$ flux calculation (see Appendix F for details)
which allowed us to identify the contributions of all relevant parameters (Table 4).

**Table 4. Results of the sensitivity analysis showing the effect of varying the relevant parameters by 10%.**

|  | Temperature | | Salinity | | Wind speed | | $p$CO$_2$ difference | |
|---|---|---|---|---|---|---|---|---|
|  | +10 % | -10 % | +10 % | -10 % | +10 % | -10 % | +10 % | -10 % |
| **Air-sea CO$_2$ flux difference (mmol m$^{-2}$ d$^{-1}$)** | 0.016 | -0.017 | 0.044 | -0.045 | -0.440 | 0.398 | -0.210 | 0.210 |

On average, air-sea $CO_2$ fluxes values during the SUP were mostly driven by wind speed term followed by sea-air $p$CO$_2$
difference, salinity and finally temperature. According to Eqs. (8) and (9), wind speed, salinity, and temperature only affect
the magnitude of air-sea $CO_2$ fluxes while their sign is determined by the sea-air $p$CO$_2$ difference which also impacts their
magnitude significantly (Table 4). We have shown that, during the SUP, this difference is mostly driven by temperature
since seawater $p$CO$_2$ variations are controlled by temperature at this time (Figs. 5d, e). A realistic representation of seawater
$p$CO$_2$ is crucial to calculate air-sea $CO_2$ fluxes. Since seawater $p$CO$_2$ variations were correctly represented by the model
during the SUP (Fig. 3c), the modelled air-sea $CO_2$ fluxes during the SUP should be reliable.

Over the entire year, air-sea $CO_2$ fluxes in the BoM essentially evened out yielding only a slightly negative balance of -0.21
mmol m$^{-2}$ per year. This is much lower than the -803 mmol m$^{-2}$ per year suggested by Wimart-Rousseau et al. (2020). The
reason for this discrepancy may be related to the fact that our model overestimates seawater $p$CO$_2$ during winter, resulting in
a sea-air difference close to zero (Fig. 4d). As a result, despite strong winds and low temperatures which would favour $CO_2$
absorption (Middelburg, 2019), the winter $CO_2$ sink is not well represented.

Seawater $p$CO$_2$, air-sea $CO_2$ fluxes and DIC are closely connected (Appendix B). In Eco3M_MIX-CarbOx, aeration is
simulated by applying Eq. (8) to 1 m$^3$ of surface water at SOLEMIO station which tends to overestimate the impact of
aeration process on DIC and, due to the close link between DIC and $p$CO$_2$, also on $p$CO$_2$. Indeed, when the aeration process
is applied to this small volume, the balance between atmosphere and the volume is quickly reached, which then impacts the
representation of $p$CO$_2$. To overcome this problem, we need to consider a larger layer of water on which aeration process is
applied. Consequently, we ran a simulation in which we considered a larger thickness of water (H = 30.5 m, annual mean
value of the mixed layer depth in the area ; Eq. 8) to apply the aeration process. This simulation and its results are described
in supplementary material. By increasing the volume on which aeration process is applied, the annual mean value of air-sea
$CO_2$ fluxes is more realistic (-113.6 mmol m$^{-2}$ yr$^{-1}$), but still, much lower than the one obtained by Wimart-Rousseau et al.
(2020) in the area. In fact, to represent the air-sea $CO_2$ fluxes, especially their annual mean value in a more realistic way, we
must consider, on the one hand, a realistic volume of water on which the aeration process is applied and on the other hand,
all the processes that take place in the water column and impact this flux, especially vertical mixing and matter transfer to the
bottom of the water column. Consequently, in the present state, Eco3M_MIX-CarbOx is unable to represent the annual cycle

of air-sea $CO_2$ fluxes. Overcoming this problem requires the switch to a 3D configuration, which is planned for our future
work.

Most studies that investigated air-sea $CO_2$ fluxes and other carbonate system variables in various Mediterranean locations at different locations (Ligurian Sea, North Adriatic Sea, BoM) were based on measurements only and concluded that their study areas acted as $CO_2$ sinks during their study periods (e.g., Begovic, 2003; De Carlo et al., 2013; Ingrosso et al., 2016; Urbini et al., 2020; Wimart-Rousseau et al., 2020). To the best of our knowledge, the only other study examining air-sea $CO_2$
fluxes in the BoM using a modelling approach was conducted by Lajaunie-Salla et al. (2021) using Eco3m-CarbOx model, which is also dimensionless and based on a 1 $m^3$ volume like Eco3M_MIX-CarbOx and therefore also tend to underestimate the yearly fluxes. Most modelling studies have focussed on larger scales and employed at least 1D models. For instance, D'Ortenzio et al. (2008), used a coupled 1D model, and found that the Mediterranean Sea, as a whole, was nearly balanced as the western and eastern basins act as $CO_2$ sink and a source, respectively, and therefore cancel each other out. Using a 3D
coupled model and looking at even larger scales, Bourgeois et al. (2016) provided a complete analysis of the air-sea $CO_2$ fluxes in various coastal environments and have shown that they represent 4.5 % of the anthropogenic $CO_2$ uptake of the global ocean. 3D models typically allow more realistic representations of the water column, they would allow us to (i) consider a more realistic water column (volume and processes which impact it) to perform our air-sea $CO_2$ fluxes calculation, (ii) consider autochthonous and allochthonous contributions to TA variations, (iii) consider the effects of
nutrients and DIC inputs from the Rhône River intrusions and local upwellings. Nevertheless, dimensionless model also offers some advantages including short simulation time and easy adaptability which allowed us to provide a detailed analysis of drivers of seawater $pCO_2$ variations, particularly during specific hydrodynamic processes typical for the BoM. This type of study is still uncommon in the area, as few of them investigated the carbonate system dynamics, especially the $pCO_2$ variations drivers and would have been a tedious task to realize in 3D (i.e., longer simulations and isolation of $pCO_2$
variation drivers' contributions more difficult due to the complexity of the model).

## 5 Conclusion

Using the concept of the dimensionless Eco3M-CarbOx biogeochemical model as a starting point, we developed a new planktonic ecosystem model which contains, in addition to mixotroph organisms, a modified version of the carbonate module described by Lajaunie-Salla et al. (2021), to represent the carbonate system variables more realistically. First, we
improved the parametrisation of TA by developing two different formulations: (i) an autochthonous formulation that only considers biological contributions to TA variations and (ii) an allochthonous formulation that only depends on salinity, thus considers riverine contributions to TA variations. A comparison of both TA formulations showed that TA variations in the BoM were mostly due to allochthonous contributions. Then, we adapted the allochthonous formulation for modelling TA variations in the BoM which, yielded a helpful tool to complement the low frequency in situ measurements. We use this new

formulation to study air-sea $CO_2$ fluxes and seawater $p$CO$_2$ variations at SOLEMIO station in 2017, focussing on two hydrodynamic processes that are typical for the BoM: (i) Rhône River intrusions and (ii) summer upwelling events.

During the SUP, our model represented the $CO_2$ sinks generated by summer upwelling events which are suggested by Lajaunie-Salla et al., (2021), and identified the underlying drivers of $CO_2$ variability. Furthermore, our model was able to simulate the expected decrease in $p$CO$_2$ associated with summer upwelling events (Lajaunie-Salla et al., 2021). This decrease

was mainly generated by temperature effects on $p$CO$_2$. LSE were also represented by the model. They often generated a decrease in $p$CO$_2$ as a result of the decreasing salinity and increasing TA, especially when those two contributions were not counterbalanced by temperature effects. However, in winter, the model was unable to reproduce the undersaturation seen in seawater $p$CO$_2$ measurements at SOLEMIO station and rather overestimate it. As a result, the present configuration of Eco3M_MIX-CarbOx is unable to reproduce the commonly observed seasonality of air-sea $CO_2$ fluxes in the north-western

Mediterranean. This pattern directly impacts our estimates of the overall yearly air-sea $CO_2$ flux, as, even if the model clearly identifies the bay as a $CO_2$ sink, it does not allow to reproduce the observed mean annual value of air-sea $CO_2$ fluxes (our model : -0.21 mmol m$^{-2}$ per year, Wimart-Rousseau et al., (2020): -803 mmol m$^{-2}$ per year).

The present work clearly highlighted the limitations of dimensionless models. Although this type of model possesses some advantages that facilitate an improved understanding of complex coastal systems, it has clear limitations when it comes to

the representation of specific processes or variables with obvious impacts on the results. The accuracy could be improved by employing a 3D coupled model which would allow us to (i) improve our representation of air-sea $CO_2$ fluxes by applying them to the whole water column, (ii) improve our representation of TA by considering autochthonous and other allochthonous sources and (iii) improve our representation of LSE and upwelling events by allowing us to consider the inputs of nutrients and DIC.


## Appendix A: State equations processes description

**Table A1. Description of state equation processes.**

| Notation | Process |
|---|---|
| **Copepods** | |
| $Excr_{NutX}^{COP_X}$<br>$NutX \in [NH_4^+, PO_4^{3-}]$<br>$X \in [N, P]$ | Excretion of nutrient X by copepods |
| $Excr_{DOC}^{COP_C}$ | DOC excretion by copepods |
| $Resp_{DIC}^{COP_C}$ | Copepods respiration |
| $E_{POX}^{COP_X}$<br>$X \in [C, N, P]$ | Copepods egestion |
| $Predation_{POX}^{COP_X}$<br>$X \in [C, N, P]$ | Predation by higher trophic levels on copepods |
| **Mixotrophs (Mix $\in$ [NCM, CM])** | |
| $Exu_{DOX}^{Mix_{X_i}}$<br>$X \in [C, N, P]$ | DOX exudation by mixotrophs |
| $Resp_{DIC}^{Mix_C}$ | Mixotrophs respiration |
| $Photo_{DIC}^{Mix_C}$ | Mixotrophs photosynthesis |
| $Excr_{NutX}^{NCM_X}$<br>$NutX \in [NH_4^+, PO_4^{3-}]$<br>$X \in [N, P]$ | Excretion of nutrient X by NCM |
| $Upt_{NutX}^{CM_X}$<br>$X \in [N, P]$<br>$NutX \in [NO_3^-, NH_4^+, PO_4^{3-}]$ | Uptake of nutrient X by constitutive mixotrophs |
| $Upt_{DOX}^{CM_X}$<br>$X \in [N, P]$ | Uptake of DOX by constitutive mixotrophs |
| **Phytoplankton (Phy $\in$ [NMPHYTO, PICO])** | |
| $Resp_{DIC}^{Phy_C}$ | Phytoplankton respiration |
| $Photo_{DIC}^{Phy_C}$ | Phytoplankton photosynthesis |
| $Upt_{NutX}^{Phy_X}$<br>$NutX \in [NO_3^-, NH_4^+, PO_4^{3-}]$ | Uptake of nutrient X by phytoplankton |
| $Exu_{DOX}^{Phy_X}$<br>$X \in [C, N, P]$ | DOX exudation by phytoplankton |
| $Upt_{DOX}^{PICO_X}$<br>$X \in [N, P]$ | Uptake of DOX by picophytoplankton |
| **Heterotrophic bacteria** | |
| $BP_X^{BAC_C}$ | Bacterial production |

| | |
|---|---|
| $X \in [DOC, POC]$ | |
| $BR_{DIC}^{BAC_C}$ | Bacterial respiration |
| $Upt_{POX}^{BAC_X}$ $X \in [N, P]$ | POX uptake by heterotrophic bacteria |
| $Exu_{DOX}^{Phy_{X_i}}$ $X \in [C, N, P]$ | DOX exudation by phytoplankton |
| $Remin_{BAC_X}^{NutX}$ $NutX \in [NH_4^+, PO_4^{3-}]$ $X \in [N, P]$ | Remineralisation of nutrient X by heterotrophic bacteria |
| $Mort_{DOX}^{BAC_X}$ | Heterotrophic bacteria natural mortality |
| **Dissolved inorganic matter (DIM)** | |
| $Diss_{DIC}^{CaCO_3}$ | $CaCO_3$ dissolution |
| $Prec_{DIC}^{CaCO_3}$ | $CaCO_3$ precipitation |
| Nitrif | Nitrification |
| $Aera_{DIC}$ | Air-sea $CO_2$ gas exchanges (aeration) |

## Appendix B: pH$_T$ and $p$CO$_2$ calculation

The calculation method performed in the Eco3M_MIX-CarbOx model to obtain pH$_T$ and $p$CO$_2$ is detailed below. As specified in Sect. 2, we used the method introduced by Lajaunie-Salla et al. (2021), which is based on CO2SYSv3 (Sharp et al., 2020), a software originally developed by Lewis and Wallas (1998) to perform the resolution of carbonate system, to perform this calculation. This appendix aims to complete Appendix A from Lajaunie-Salla et al. (2021) by providing some corrections. It also introduces the possibility to choose between two types of TA formulation (autochthonous or

allochthonous) to perform the calculation of pH$_T$ and $p$CO$_2$.

### B.1 Equilibrium constants and conservative elements concentrations calculation

In the following formulations, S represents the practical salinity.

### B.1.1 Conservative elements concentrations and ionic strength

**Table B1. Formulations of conservative elements concentrations and ionic strength.**

| Description | Formulation | Units |
|---|---|---|
| Concentration in total fluoride (Riley, 1965) | $TF = \dfrac{0.000067}{18.998} * \dfrac{S}{1.80655}$ | mol kg$^{-1}$ |
| Concentration in total sulfate (Morris & Riley, 1966) | $TS = \dfrac{0.14}{96.062} * \dfrac{S}{1.80655}$ | mol kg$^{-1}$ |
| Concentration in total Boron (Uppström, 1974) | $TB = \dfrac{0.000416 * S}{35}$ | mol kg$^{-1}$ |
| Concentration in calcium ion (Riley & Tongudai, 1967) | $Ca^{2+} = \dfrac{0.02128}{40.087} * \dfrac{S}{1.80655}$ | mol kg$^{-1}$ |
| Ionic strength (DOE, 1994) | $IonS = \dfrac{19.924 * S}{1000 - 1.005 * S}$ | $\emptyset$ |

**B.1.2 Equilibrium constants**

In the following formulations, T represents temperature value converted in Kelvin (i.e., T(°C) + 273.15).

K$_F$ (mol kg$^{-1}$): HF dissociation constant (Dickson & Riley, 1979)

$$ln(K_F) = \frac{1590.2}{T} - 12.641 + 1.525 * IonS^{0.5}$$

$$K_F = exp\big(ln(K_F) * (1 - 0.001005 * S)\big) \tag{B1}$$

$K_F$ is expressed on free pH scale.

$K_S$ (mol kg$^{-1}$): HSO$_4^-$ dissociation constant (Dickson, 1990a)

$$ln(K_S)_{temp} = -\frac{4276.1}{T} + 141.328 - 23.093 * ln(T) + \left(-\frac{13856}{T} + 324.57 - 47.986 * ln(T)\right) * IonS^{0.5}$$

$$ln(K_S) = ln(K_S)_{temp} + \left(\frac{35474}{T} - 771.54 + 114,723 * ln(T)\right) * IonS - \frac{2698}{T} * IonS^{1.5} + \frac{1776}{T} * IonS^2$$

$$K_S = exp\left(ln(K_S) * (1 - 0.001005 * S)\right) \tag{B2}$$

$K_S$ is expressed on free pH scale.

$K_B$ (mol kg$^{-1}$): B(OH)$_3$ dissociation constant (Dickson, 1990b)

$$ln(K_B)_{temp} = \frac{-8996.9 - 2890.53*S^{0.5} - 77.942*S + 1.728*S^{1.5} - 0.0996*S^2}{T} + 148.0248 + 137.1942 * S^{0.5}$$

$$ln(K_B) = ln(K_B)_{temp} + 1.62142 * S + (-24.4344 - 25.085 * S^{0.5} - 0.2474 * S) * ln(T) + 0.053105 * S^{0.5} * T$$

$$K_B = exp\left(ln(K_B)\right) \tag{B3}$$

$K_B$ is expressed on total pH scale.

$K_{ca}$ (mol kg$^{-1}$)$^2$: Calcite formation constant (Mucci, 1983)

$$log(K_{ca})_{temp} = -171.9065 - 0.077993 * T + \frac{2839.319}{T} + 71.595 * log(T)$$

$$log(K_{Ca}) = log(K_{ca})_{temp} + \left(-0.77712 + 0.0028426 * T + \frac{178.34}{T}\right) * S^{0.5} - 0.07711 * S + 0.0041249 * S^{1.5}$$

$$K_{ca} = 10^{(log(K_{Ca}))} \tag{B4}$$

$K_e$ (mol kg-1): H$_2$0 dissociation constant (Millero, 1995)

$$ln(K_e) = -\frac{13847.26}{T} + 148.9802 - 23.6521 * ln(T) + \left(-5.977 + \frac{118.67}{T} + 1.0495 * ln(T)\right) * S^{0.5} - 0.01615 * S$$

$$K_e = exp\left(ln(K_e)\right) \tag{B5}$$

$K_e$ is expressed on SWS pH scale.

$K_0$ (mol kg$^{-1}$ atm$^{-1}$): CO$_2$ solubility (Weiss, 1974)

$$ln(K_0)_{temp} = -60.2409 + 93.4517 * \frac{100}{T} + 23.3585 * ln\left(\frac{T}{100}\right)$$

$$ln(K_0) = ln(K_0)_{temp} + S * \left(0.023517 - 0.023656 * \frac{T}{100} + 0.0047036 * \left(\frac{T}{100}\right)^2\right)$$

$$K_0 = exp\left(ln(K_0)\right) \tag{B6}$$

$K_1$ (mol kg-1): H$_2$CO$_3$ dissociation (Lueker et al., 2000)

$$pK_1 = \frac{3633.86}{T} - 61.2172 + 9.6777 * ln(T) - 0.011555 * S + 0.0001152 * S^2$$

$K_1 = 10^{(-pK_1)}$                                                                (B7)

$K_1$ is expressed on total pH scale.

$K_2$ (mol kg$^{-1}$): $HCO_3^-$ dissociation (Lueker et al., 2000)

$$pK_2 = \frac{471.78}{T} + 25.929 - 3.16967 * ln(T) - 0.01781 * S + 0.0001122 * S^2$$

$$K_2 = 10^{(-pK_2)}$$         (B8)

$K_2$ is expressed on total pH scale.

## B.1.3 pH scale conversion

pH calculation is performed on total scale. Accordingly, the previous constants are converted if necessary (i.e., expressed on total pH scale) using the following conversion factors. Except $K_S$ and $K_F$ which must be expressed on free pH scale, the other equilibrium constants must be converted to total pH scale.

**Table B2. Formulation of pH scale conversion factors.**

| Description | Conversion factor |
| --- | --- |
| From SWS pH scale to total pH scale | $\dfrac{1 + \dfrac{T_S}{K_S}}{1 + \dfrac{T_S}{K_S} + \dfrac{T_F}{K_F}}$ |
| From free pH scale to total pH scale | $1 + \dfrac{T_S}{K_S}$ |

## B.1.4 Pressure correction

All the constants are corrected by the effect of hydrostatic pressure using the following formulations (Millero, 1995). We define $T_K$ and $T_C$ which represents respectively the temperature in Kelvin and in Celsius degree. R represents the gas constant in ml bar$^{-1}$ K$^{-1}$ mol$^{-1}$ (R = 83.1451 ml bar$^{-1}$ K$^{-1}$ mol$^{-1}$) and P the pressure in bar.

Corrected $K_F$ (mol kg$^{-1}$):

$$K_F CorrFac = \frac{\left(9.78 + 0.009*T_C + 0.0009429*T_C^2 + 0.5*\left(\frac{-3.91 + 0.054*T_C}{1000}\right)*P\right)*P}{R*T_K}$$

$$K_F = K_F * exp(K_F CorrFac)$$         (B9)

Corrected $K_S$ (mol kg$^{-1}$):

$$K_S CorrFac = \frac{\left(18.03 - 0.0466*T_C - 0.000316*T_C^2 + 0.5*\left(\frac{-4.53 + 0.09*T_C}{1000}\right)*P\right)*P}{R*T_K}$$

$K_S = K_S * exp(K_S CorrFac)$ (B10)

Corrected $K_B$ (mol kg$^{-1}$):

$$K_B CorrFac = \frac{\left(29.48 - 0.1622 * T_C + 0.002608 * T_C^2 + 0.5 * \left(-\frac{2.84}{1000}\right) * P\right) * P}{R * T_K}$$

$K_B = K_B * exp(K_B CorrFac)$ (B11)

Corrected $K_{ca}$ (mol kg$^{-1}$)$^2$:

$$K_{ca} CorrFac = \frac{\left(48.76 - 0.5304 * T_C + 0.5 * \left(\frac{-11.76 + 0.3692 * T_C}{1000}\right) * P\right) * P}{R * T_K}$$

$K_{ca} = K_{ca} * exp(K_{ca} CorrFac)$ (B12)

Corrected $K_e$ (mol kg$^{-1}$):

$$K_e CorrFac = \frac{\left(20.02 - 0.1119 * T_C + 0.001409 * T_C^2 + 0.5 * \left(\frac{-5.13 + 0.0794 * T_C}{1000}\right) * P\right) * P}{R * T_K}$$

$K_e CorrFac = K_e * exp(K_e CorrFac)$ (B13)

Corrected $K_1$ (mol kg$^{-1}$):

$$K_1 CorrFac = \frac{\left(25.5 - 0.1271 * T_C + 0.5 * \left(\frac{-3.08 + 0.0877 * T_C}{1000}\right) * P\right) * P}{R * T_K}$$

$K_1 = K_1 * exp(K_1 CorrFac)$ (B14)

Corrected $K_2$ (mol kg$^{-1}$):

$$K_2 CorrFac = \frac{\left(15.82 + 0.0219 * T_C + 0.5 * \left(\frac{1.13 + 0.1475 * T_C}{1000}\right) * P\right) * P}{R * T_K}$$

$K_2 = K_2 * exp(K_2 CorrFac)$ (B15)

## B.1.5 Fugacity factor

To perform the calculation of the fugacity factor (FugFac), we supposed that the pressure value is close or equal to an atmosphere (Weiss, 1974).

T represents the temperature in Kelvin. We define $P_{atm}$, as the atmospheric pressure in bar: $P_{atm} = 1.01325$ bar.

$$ln(FugFac) = \frac{\left((-1636.75 + 12.0408 * T - 0.0327957 * T^2 + 3.16528 * 0.00001 * T^3) + 2 * (57.7 - 0.118 * T)\right) * P_{atm}}{R * T}$$

$FugFac = exp\left(ln(FugFac)\right)$ (B16)

## B.2 pH$_T$ and $p$CO$_2$ calculation

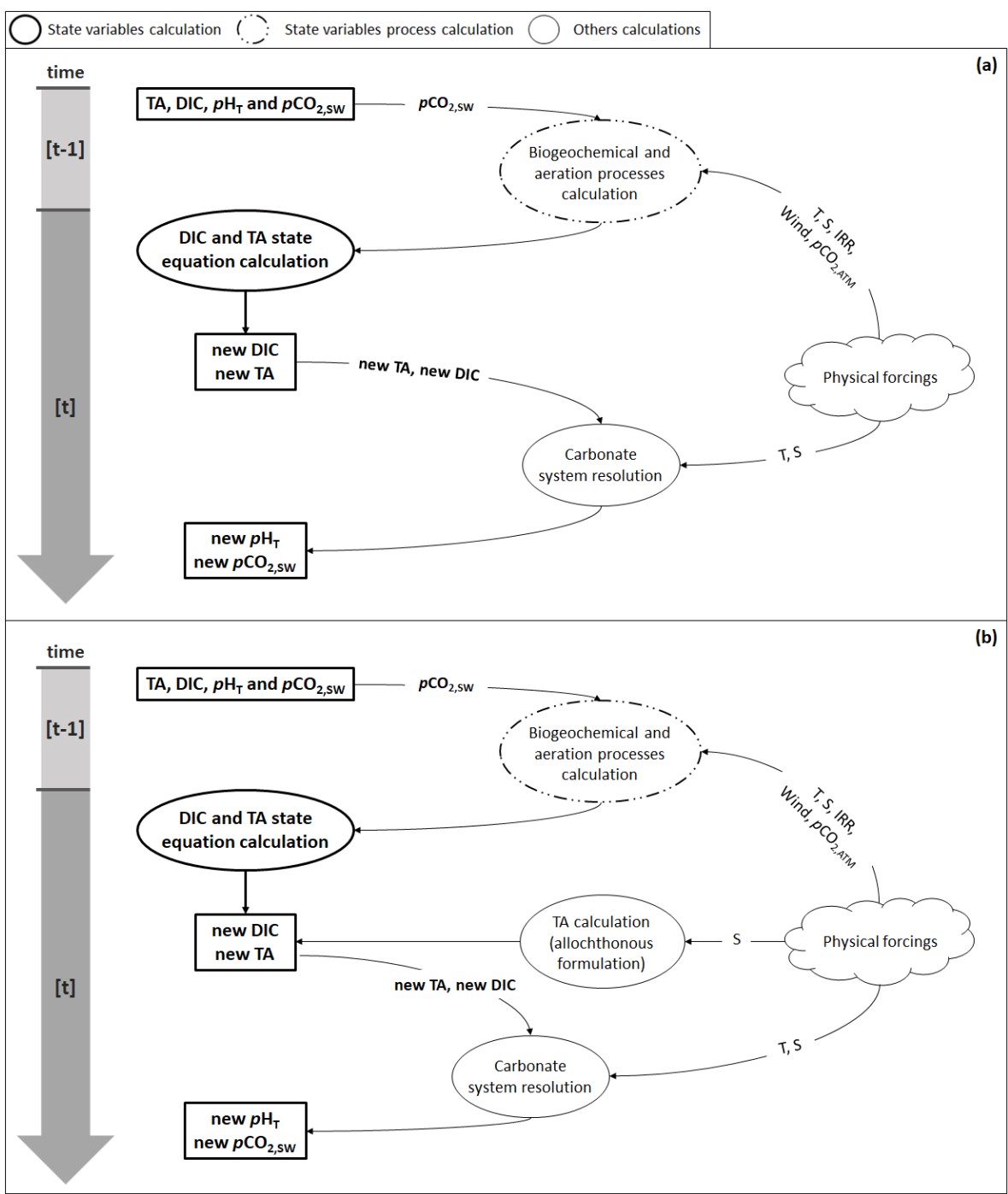

Figure B1. Flow diagram illustrating the steps needed to calculate pH$_T$ and $p$CO$_2$ (a) using the autochthonous formulation (Eq. 4) and (b) with the allochthonous formulation (Eq. 5 and 6). Physical forcings include temperature (T), salinity (S), solar irradiance (IRR), wind speed (Wind) and atmospheric $p$CO$_2$ ($p$CO$_{2,ATM}$).

Solving the equations of the carbonate system requires knowledge of TA and DIC. Depending on the TA formulation used, the steps followed by the model to issue the new $pH_T$ and $pCO_2$ are described on Fig. B1. If TA is calculated using the Eq. (4), biogeochemical and aeration processes are applied as described in Eqs. (4) and (7) in order to deliver new ([t] time step) TA and DIC: air-sea $CO_2$ fluxes are calculated from temperature, salinity, wind speed, atmospheric $pCO_2$ and seawater $pCO_2$, and biogeochemical processes required, at least, temperature to be computed and solar irradiance. When calculated, processes are applied in the form of fluxes to the previous TA and DIC ([t-1] time step values) to solve their respective state equation. The $pH_T$ and $pCO_2$ calculation is, then, performed using in addition to TA and DIC, temperature and salinity data. When TA is calculated using Eqs. (5) and (6), the biogeochemical and aeration fluxes computed during the first stage are only applied to DIC from the preceding time step, while TA is calculated after DIC based on the salinity data from the current time step. All subsequent steps are unchanged (Fig. B1b).

## B.2.1 $pH_T$ calculation

```
! pH initial value = 8.0
! pHTol = Tolerance threshold --> 0.0001
! deltapH = pH difference between two model iterations
! pH is calculated on total scale

if (nbIter < 1) pH = 8.0

pHTol = 0.0001
deltapH = pHTol + 1

do while (abs(deltapH) > pHTol)
 H = 10^(-pH)
 Denom = H^2 + K1 * H + K1 * K2
 CAlk = DIC * K1 * ((H + 2 * K2)/Denom) !Carbonate Alkalinity
 BAlk = (TB * KB)/(KB + H) ! Borate Alkalinity
 OH = Ke/H
 FreeToTot = 1 + (TS/KS)
 HFree = H/FreeToTot
 HSO4 = TS/(1+(KS/HFree))
 HF = TF/(1+(KF/H))
 Residual = TA - CAlk - BAlk - OH + HFree + HSO4 + HF
 Slope = DIC * H * K1 * (H^2 + K1 * K2 + 4 * H * K2)
 Slope = Slope/(Denom^2) + OH + H + (BAlk * H)/(KB + H)
 Slope = log(10) * Slope
 deltapH = Residual/Slope

 do while (abs(deltapH) > 1)
  deltapH = deltapH/2
 enddo

pH = pH + deltapH
enddo
```

**Figure B2. $pH_T$ calculation**

$pH_T$ is calculating using a buffering value (B) defined as the pH variation induced by an addition of acid or base to a specific solution (Van Slycke, 1922). In seawater, B can be expressed in terms of TA (Middelburg, 2019) which yields:

$$B = \frac{\partial TA}{\partial pH_T} \Leftrightarrow \Delta pH_T = \frac{\partial TA}{\sum_{i=1}^{n} B_i},$$

(B17)

where i represents a chemical species contributing to TA.

Accordingly, we calculate the $pH_T$ difference between two model time steps ($\Delta pH_T$) using an iterative method. We set the $pH_T$ initial value to 8.0. We chose this value by considering the Mediterranean and Rhône River $pH_T$ which are respectively close and equal to 8.0. Finally, considering that the measurements precision is rather close to 0.0004 (Clayton & Byrne, 1993), we set the tolerance threshold to 0.0001. $pH_T$ calculation is detailed in Fig. B2.

### B.2.2 $p$CO$_2$ and carbonate system species concentrations

$p$CO$_2$ is deduced using DIC, pH (via $H^+$ concentration) and equilibrium constants. We also calculate the concentrations of $CO_2$, $HCO_3^-$, $CO_3^{2-}$ and $CaCO_3$ saturation ($\Omega$).

**Table B3. Formulation of $p$CO$_2$ and carbonate system species concentrations.**

| Description | Formulation | Units |
|---|---|---|
| $p$CO$_2$ | $pCO_2 = \dfrac{DIC * [H^+]^2}{[H^+]^2 + K_1 * [H^+] + K_1 * K_2} * \dfrac{10^6}{K_0 * FugFac}$ | µatm |
| CO$_2$ concentration | $[CO_2^*] = \dfrac{(DIC * 10^6)}{\left(1 + \dfrac{K_1}{[H^+]} + \dfrac{(K_1 * K_2)}{[H^+]^2}\right)}$ | µmol kg$^{-1}$ |
| HCO$_3^-$ concentration | $[HCO_3^-] = \dfrac{K_1 * [CO^2]}{[H^+]}$ | µmol kg$^{-1}$ |
| CO$_3^{2-}$ concentration | $[CO_3^{2-}] = \dfrac{K_2 * [HCO_3^-]}{[H^+]}$ | µmol kg$^{-1}$ |
| CaCO$_3$ saturation state | $\Omega = \dfrac{[Ca^{2+}] * [CO_3^{2-}] * 10^{-6}}{K_{ca}}$ | $\varnothing$ |

**Appendix C: Statistic indicators calculation and application to H⁺ concentration**

We used four statistical indicators for the comparison between simulation and SOLEMIO data: the percentage bias (%BIAS), the average error (AE), the average absolute error (AAE) and the root mean square deviation (RMSD, also refer as root mean square error in the literature - RMSE). They were used with two Eco3M_MIX-CarbOx simulations (SIMC0 and SIMC1) and the reference Eco3M-CarbOx simulation (Lajaunie-Salla et al., 2021). The %BIAS is calculated as follow:

$$\%BIAS = \frac{\sum_{i=1}^{N}(O_i - M_i)}{\sum_{i=1}^{N} O_i} * 100$$

(C1)

where O represents the observations and M the model results (Allen et al., 2007). This indicator allows to quantify the model's tendency to under- or overestimate the observations. The closer the value is to 0, the better the model. Here, a positive %BIAS means that the model underestimated the in situ observations and vice versa. On an indicative basis, the %BIAS can be interpreted according to Marechal (2004): Absolute values of %BIAS allow to assess the overall agreement between the model results and observations and the agreement is considered: excellent if %BIAS < 10 %, very good if 10 % ≤ %BIAS < 20 %, good if 20 % ≤ %BIAS < 40 % and poor otherwise. We based our calculation of AE, AAE and RMSD on Stow et al. (2009). Together, these three statistical indicators provide an indication of model prediction accuracy.

$$AE = \frac{\sum_{i=1}^{N}(O_i - M_i)}{n}$$

(C2)

$$AAE = \frac{\sum_{i=1}^{N}(|O_i - M_i|)}{n}$$

(C3)

$$RMSD = \sqrt{\frac{\sum_{i=1}^{N}(O_i - M_i)^2}{N}}$$

(C4)

The three of them aim to measure the size of the discrepancies between model results and observations, the closer the value is to 0, the better the agreement between model results and observations. However, when interpreting AE, it is important to note that value near zero can be misleading because negative and positive discrepancies can cancel each other. That is why it is important to calculate, in addition to AE, AAE and RMSD which allow to overcome this effect (Stow et al., 2009). Such as %BIAS, a positive value of AE means that the model underestimated the in situ observations and vice versa. The model data is averaged using the mean of the output from the date in question ± five days. Using temporal mean and standard deviation of model results allowed us to better account of variability at SOLEMIO station.

In addition to TA, DIC, pH$_T$ and $p$CO$_2$, statistical indicators were calculated for H⁺ concentrations (Table C1).

**Table C1. Comparing the different model results to surface observations at SOLEMIO station for H$^+$ concentration. N represents the number of observations. Mean, SD, AE, AAE and RMSD are in the same unit than the considered variable, i.e.: mmol m$^{-3}$ for H$^+$ concentrations. %BIAS is without unit.**

|  |  | [H$^+$] |
| --- | --- | --- |
| **N** | Observations | 20 |
| **Mean ± SD** | Observations | $8.08 \times 10^{-9} \pm 5.52 \times 10^{-10}$ |
| **Mean ± SD** | SIMC0 | $8.89 \times 10^{-9} \pm 2.91 \times 10^{-10}$ |
|  | SIMC1 | $8.39 \times 10^{-9} \pm 4.06 \times 10^{-10}$ |
|  | CarbOx | $8.52 \times 10^{-9} \pm 2.80 \times 10^{-10}$ |
| **%BIAS** | SIMC0 | -5.33 |
|  | SIMC1 | -3.91 |
|  | CarbOx | -5.47 |
| **AE** | SIMC0 | $-4.30 \times 10^{-10}$ |
|  | SIMC1 | $-3.15 \times 10^{-10}$ |
|  | CarbOx | $-4.42 \times 10^{-10}$ |
| **AAE** | SIMC0 | $6.45 \times 10^{-10}$ |
|  | SIMC1 | $6.05 \times 10^{-10}$ |
|  | CarbOx | $6.36 \times 10^{-10}$ |
| **RMSD** | SIMC0 | $6.98 \times 10^{-10}$ |
|  | SIMC1 | $7.14 \times 10^{-10}$ |
|  | CarbOx | $6.93 \times 10^{-10}$ |

**Appendix D: Time series of daily average $p$CO₂ anomalies generated by DIC, TA, S+Fw and temperature based on the approach described by Lovenduski et al. (2007), for 2017. Enlargement of the panel d, of figure 5.**

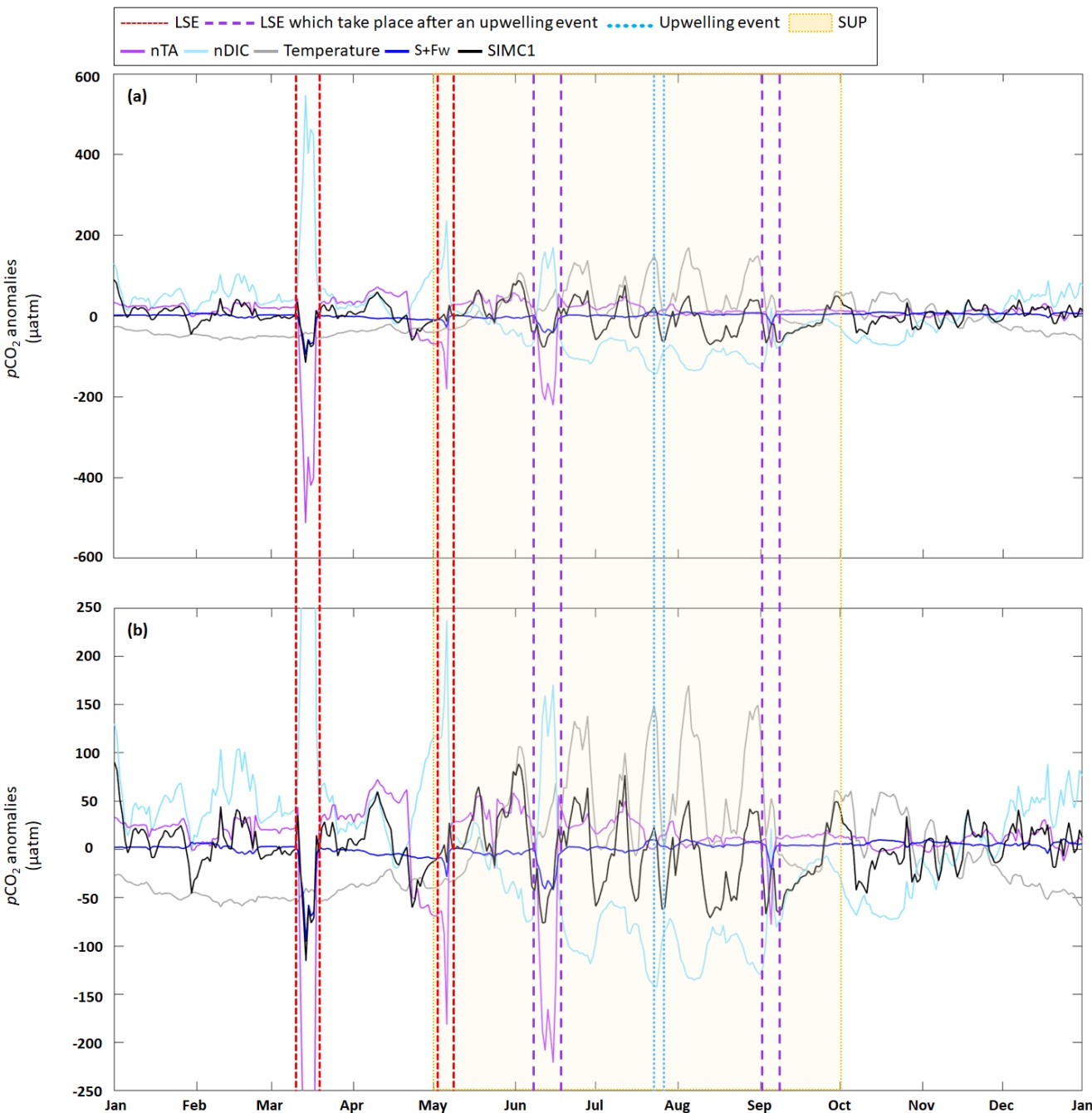


**Figure D1. Time series for 2017 of daily average (a) $p$CO₂ anomalies generated by DIC, TA, S+Fw and temperature based on the approach in Lovenduski et al. (2007) (Note: the dark blue line is sometimes obscured by the black line, especially in March), (b) Enlargement of the panel a between -250 and 250 µatm. LSE and an upwelling event have been highlighted. The summer upwelling period (SUP) is indicated by yellow shading.**

**Appendix E: DIC and nutrients SOLEMIO data interpolation**

As we represent a closed volume, we do not consider nutrients and DIC inputs which could be associated with LSE or upwelling events (Gatti et al., 2006, Fraysse et al., 2013, 2014, Lajaunie-Salla et al., 2021). To assess if these inputs impact SOLEMIO, we interpolated DIC and nutrients measurements performed at the station, then studying the trend observed during the events studied in the present study (Fig. E1).

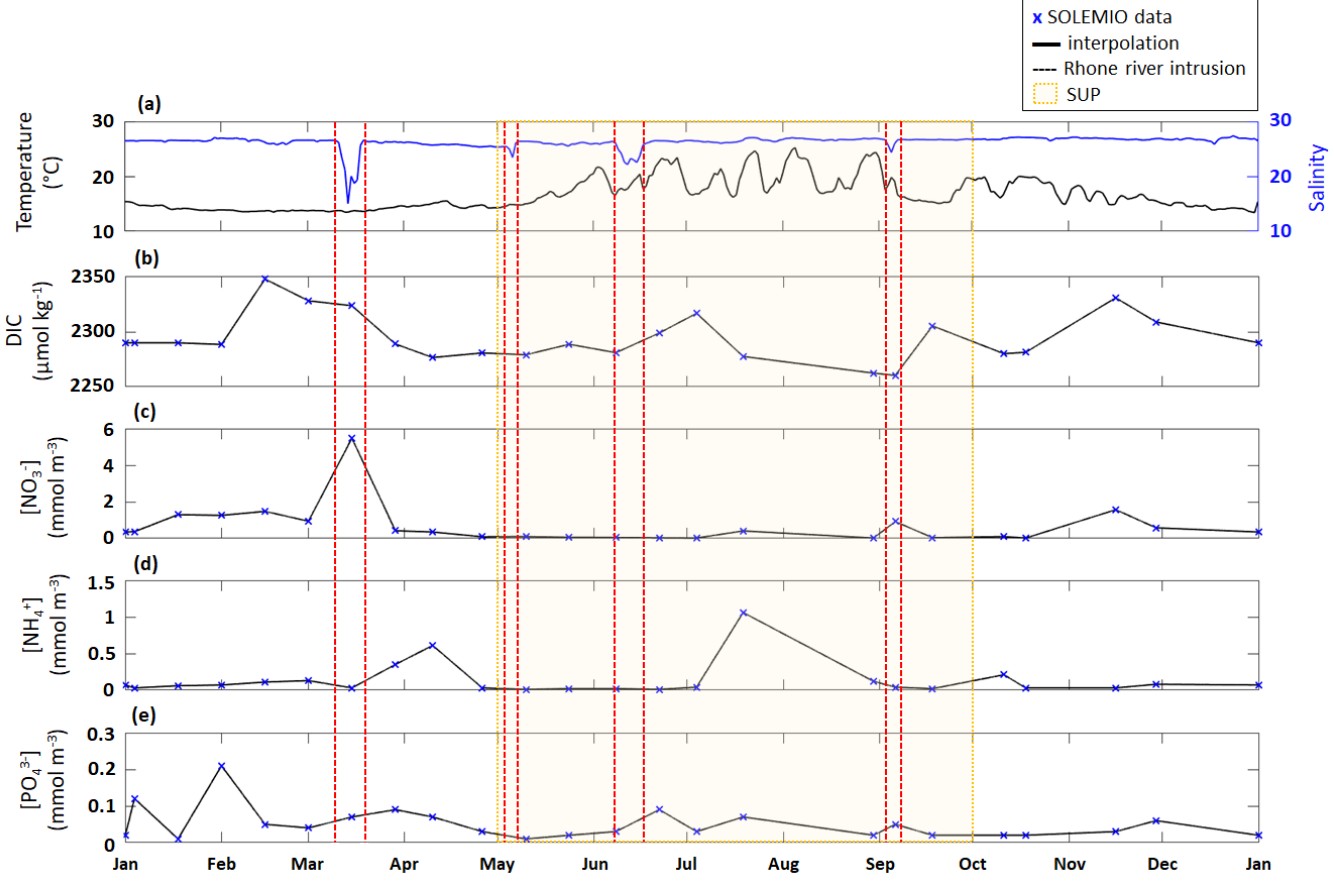


**Figure E1. Time series of surface (a) temperature (PLANIER measurements) and salinity (CARRY measurements) and interpolated (b) DIC, (c) NO₃⁻, (d) NH₄⁺ and (e) PO₄³⁻ concentration at SOLEMIO station. SOLEMIO data are represented by blue markers. Rhone River intrusions studied here are indicated by the red dotted lines and the SUP is shaded in yellow.**

**Table E2. Surface DIC and nutrients concentration measurements at SOLEMIO station during LSE and SUP for the year 2017.**

| Date | Event | DIC (μmol kg$^{-1}$) | NO$_3^-$ (mmol m$^{-3}$) | NH$_4^+$ (mmol m$^{-3}$) | PO$_4^{3-}$ (mmol m$^{-3}$) |
|---|---|---|---|---|---|
| 15 March | LSE | 2323.8 | 5.5 | 0.03 | 0.07 |
| 6 May | LSE | No measurement available | | | |
| 10 May | SUP | 2279.1 | 0.1 | 0.01 | 0.01 |
| 24 May | SUP | 2288.7 | 0.06 | 0.02 | 0.02 |
| 8 June | SUP | 2281.0 | 0.05 | 0.02 | 0.03 |
| 15 June | LSE | No measurement available | | | |
| 22 June | SUP | 2299.0 | 0.09 | 0.01 | 0.09 |
| 4 July | SUP | 2316.9 | 0.03 | 0.04 | 0.03 |
| 19 July | SUP | 2277.6 | 0.4 | 1.05 | 0.07 |
| 30 August | SUP | 2262.4 | 0.02 | 0.12 | 0.02 |
| 5 September | LSE and SUP | 2260.3 | 0.9 | 0.04 | 0.05 |
| 18 September | SUP | 2305.4 | 0.04 | 0.02 | 0.02 |

**Appendix F: Sensibility analysis performed on air-sea $CO_2$ fluxes calculation**

A sensibility analysis was performed to evaluate the importance of temperature, salinity, wind speed and seawater-
atmospheric $pCO_2$ difference terms in the air-sea $CO_2$ fluxes calculation. Previous terms are one by one increased (decreased) by 10 %. Air-sea $CO_2$ fluxes are then, post-processed using the Eqs. (8) and (9). Calculation is performed using MATLAB. We present in Table 4 the mean difference between the reference air-sea $CO_2$ fluxes (i.e., calculated without increasing (decreasing) by 10 % one of the calculation terms) and the air-sea $CO_2$ fluxes obtained by adding (removing) 10 % to one of the terms of the calculation (Eq. F1).

$\Delta_{\text{Air-sea}} CO_2 \text{Fluxes} = \frac{1}{N} * \sum_{i=1}^{N} \big( abs(Ref) - abs(X_{10\%}) \big),$                                           (F1)

where $\Delta_{\text{Air-sea}} CO_2 \text{Fluxes}$ is expressed in mmol m$^{-2}$ s$^{-1}$ N is the number of modelled values. X represents temperature, salinity, wind speed or the difference between seawater and atmospheric $pCO_2$

**Code availability**

The current version of Eco3M_MIX-CarbOx is available from the Zenodo website (https://zenodo.org/record/7669658#.Y_dAJ0NKg2w, last access: 23 February 2023) under the Creative Commons Attribution 4.0 international licence. The exact version of the model used to produce the results in this paper is archived on Zenodo (Barré Lucille, Diaz Frédéric, Wagener Thibaut, Van Wambeke France, Mazoyer Camille, Yohia Christophe, & Pinazo Christel. (2022). Eco3M_MIX-CarbOx (v1.0). Zenodo. https://doi.org/10.5281/zenodo.7669658), as are input data
and scripts to run the model and produce the plots for all the simulation presented in this paper.

**Data availability**

SOLEMIO time serie data is available on https://www.seanoe.org,. Temperature data is available on www.t-mednet.org by filling out the request form for station and years pre-selected. Salinity data is available on https://erddap.osupytheas.fr. The non-processed atmospheric $pCO_2$ data can be found on https://servicedata.atmosud.org/donnees-stations. Request for
processed atmospheric $pCO_2$ data should be addressed to alexandre.armengaud@airpaca.org and irene.xueref-remy@imbe.fr.

**Author contribution**

LB conceptualized this study, developed the Eco3M_MIX-CarbOx model v1.0, and  it, designed the numerical experiments, developed MATLAB software to visualize and process the model results, processed, and analysed the model results, wrote
the initial draft. FD provided the initial version of the model code (without carbonate module and with an initial implementation of the mixotroph organisms) and helped to develop the Eco3M_MIX-CarbOx v1.0. TW participated to the conceptualization of this study, participated to the data acquisition of carbonate variables, helped to design the numerical experiments, analysed the model results, reviewed, and edited the initial draft. CM helped in the model development process by giving expertise on the code development to reduce calculation time. CY provided the wind and irradiance data,
maintained computing resources. CP acquired the fundings, participated to the conceptualization of this study and supervised it, participated to the model development, designed the numerical experiments, analysed the model results, and reviewed and edited the initial draft.

**Competing interests**

The authors declare that they have no conflict of interest.

**Acknowledgements**

We thank the National Service d'Observation en Milieu LITtoral (SOMLIT) for its permission to use SOLEMIO data. We would like to thank the crew members of the RV Antedon II, operated by the DT-INSU, for making these samplings possible, the team of the SAM platform (Service Atmosphère Mer) of the MIO for help with the field work. We also thank Michel Lafont and Véronique Lagadec of the PACEM (Plateforme Analytique de Chimie des Environnenments Marins) platform of the MIO and the SNAPO-$CO_2$ at LOCEAN, Paris. The SNAPO-$CO_2$ service at LOCEAN is supported by CNRS-INSU and OSU Ecce-Terra. We acknowledge the TMEDNet team for its permission to use the Planier-Souquet temperature data. We thank the ROMARIN network team for its permission to use the salinity data from Carry buoy. We thank the observatoire de la qualité de l'air en Région Sud Provence-Alpes-Côte d'Azur (ATMOSUD) in particular, Alexandre Armengaud, and the AMC (Aix-Marseille Carbon Pilot Study) project leaders, Irène Xueref-Remy and Dominique Lefèvre for providing the atmospheric $CO_2$ data at the Cinq Avenue station. We acknowledge the staff of the "Cluster de calcul intensif HPC" platform of the OSU Institut PYTHEAS (Aix–Marseille Université, INSU-CNRS) for providing the computing facilities. We would like to thank Julien Lecubin from the Service Informatique de l'OSU Institut Pytheas for its technical assistance. We thank XpertScientific team for the manuscript correction.

**Fundings**

This work takes part of the IAMM project (Évaluer l'Impact de la métropole Aix-Marseille sur l'Acidification de la baie de Marseille et les conséquences sur les microorganismes marins, approche par Modélisation) funded by the public establishment of the Ministry of the Environment, l'Agence de l'eau Rhône Mediterranée Corse.

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
