# Peer review of "Lucille Barré[1], Frédéric Diaz[1,†], Thibaut Wagener[1], Camille Mazoyer[1], Christophe Yohia[2] and Christel Pinazo[1]"

_Geoscientific Model Development, 2023_

## Author Comment (AC1)

First, we would like to thank Julien Palmiéri for his careful evaluation of our manuscript and his interesting comments which we believe will help us to improve it. Please, find hereafter our response to these comments.

**Major comments**

1) First, the reason of the choice is not given. I understand it is greatly needed for developments like adding the mixotrophs in Eco3M. It's a huge task that require such lightweight configuration to test, verify that nothing is broken, and make sure the fluxes between each element of the model are reasonable. So, this choice is easily explained for the part 1, but less for this one.

We agree with this point. The use of a 0D configuration may seem more justified for the first part of our study as adding mixotrophs required many tests.

We chose to also use a dimensionless configuration for the part II as we wanted to provide a reliable representation of the carbonate system, which consider mixotrophs organisms, in the simplest way possible. In other words, we wanted to provide a tool which is easy to use, easy to adapt to other coastal area (by modifying the environmental forcings and the AT-S correlation) and give reliable results in a short amount of time.

Another reason to use a dimensionless configuration rather than a 3D configuration was the possibility to compare Eco3M\_MIX-CarbOx to Eco3M-CarbOx in the same type of implementation. By comparing the models, we can answer several questions and especially: How mixotrophs affect carbonate variables ? Does adding them provide a more accurate representation of these variables ? and then build further relevant simulation strategies for 3D.

Considering these reasons, this dimensionless configuration can be seen as a laboratory to test and then build further relevant simulation strategies for 3D.

To justify the use of a 0D configuration, we first clarified the aim of the study at the end of the introduction (1.79):

[Here we try to provide a more realistic representation of carbonate system variables in the BoM. As a starting point, we used the concept of the dimensionless Eco3M-CarbOx model (Lajaunie-Salla et al., 2021), which aims to represent a small volume of surface water (i.e.,  $1 \text{ m}^3$ ) in the BoM. We developed a planktonic ecosystem model which contains, among others, mixotrophic organisms, modified the carbonate module described by Lajaunie-Salla et al. (2021) and added it to our newly developed planktonic ecosystem model to obtain the Eco3M\_MIX-CarbOx model (v1.0). We implemented two types of TA formulation and compared the simulation results to in situ observations to identify which formulation was capable to deliver the more realistic results: (i) a formulation that only considers biological processes (referred to as autochthonous formulation) and (ii) a new TA formulation that depends only on salinity (referred to as allochthonous formulation). Furthermore, we simulate air-sea CO2 fluxes to determine whether the BoM act as a sink or a source of CO2 and provide a detailed analysis of drivers of seawater  $pCO_2$  variations for two specific hydrodynamic processes typical for the BoM: (i) Rhône River intrusion and (ii) summer upwelling events. With this study, we aim to provide a new tool which allow to obtain a reliable representation of the carbonate system in the simplest way as possible: by using a dimensionless configuration which is easy to use, adapt and give results in a short amount of time.]

And we gave a justification of this choice in the Section 2.2 (I.111):

[In this study, we used the Eco3M\_MIX-CarbOx model (v1.0) which was developed to represent the dynamics of the seawater carbonate system and mixotrophs in the BoM and was implemented using the Eco3M (Ecological Mechanistic and Molecular Modelling) platform (Baklouti et al., 2006a, b). Eco3M\_MIX-CarbOx is a dimensionless model (0D): we consider a volume of 1 m3 of surface water at SOLEMIO station, in this volume the state variables only vary over time as the model is not coupled with a hydrodynamic model. We chose to use a 0D configuration as this configuration has several advantages, especially, calculation times are low (around 45 minutes in our case). It allows us to make several test simulations to better understand the biogeochemical functioning of the BoM and its possible reactions to environmental forcings.]

In the introduction, it's made mention of the need for high resolution model for coastal regional study, and the next sentence announce the use of a dimensionless (0D hereafter) configuration.

We understand that these sentences can be confusing. We add a sentence to make a better link between 3D introduction and use of 0D (I.65).

[Most modelling approaches to investigate carbonate system variables typically employ 3D coupled physical-biogeochemical models and focus on larger coastal areas (e.g., Artioli et al., 2014; Bourgeois et al., 2016). If the focus is on smaller areas this requires higher spatial and temporal resolution to correctly represent the relevant processes (Bourgeois et al., 2016). However, higher spatial and temporal resolution often result in a significant increase of the calculation time which make more difficult the repetition of numerical experiments, an important step to better understanding the global functioning of the area and its reaction to environmental forcings. A solution to avoid important calculation times is to use a dimensionless model. This type of model allows to conduct large amount of test in short amount of time. For instance, Lajaunie-Salla et al. (2021) used the dimensionless Eco3M-CarbOx model, which contains a carbonate module performing the resolution of the DIC, oceanic partial pressure of  $CO_2$  ( $pCO_2$ ) and total pH ( $pH_T$ ) representations look reliable, Eco3m-CarbOx tends to minimize the range of TA variations during the year, resulting in a near constant TA (Lajaunie-Salla et al., 2021).]

2) A second point is that this OD brings more questions than answers. Because it is a surface box, the model does not represent advection and mixing. The physics variable/forcing come from observations and hence include annual cycle and external forcing, including specific phenomena like summer upwellings or the Rhône waters passing by. But what about the nutrients ?

We agree that hydrodynamic processes, especially upwelling and Rhône River intrusions in the BoM are associated with nutrients inputs. In Eco3M\_MIX-CarbOx, upwellings are only represented by strong variations of temperature and Rhône River intrusions are only represented by strong variations of salinity and TA inputs when allochthonous TA formulation is used. In other words, we do not consider the possible inputs of nutrients associated with these events and assumed that nutrients are fully the result of autochthonous biological processes (due to 0D configuration) which means that they are modelled based on the following state equations :

$$\frac{\partial \text{NO}_3}{\partial t} = \text{Nitrif}_{\text{NO}_3}^{\text{NH}_4} - \sum_{i=1}^{2} \text{Upt}_{\text{NO}_3}^{\text{PhyN}_i} - \text{Upt}_{\text{NO}_3}^{\text{CM}_{N_i}}$$
$$\frac{\partial \text{NH}_4}{\partial t} = \text{Excr}_{\text{NH}_4}^{\text{COP}_N} + \text{Excr}_{\text{NH}_4}^{\text{NCM}_N} + \text{Remin}_{\text{NH}_4}^{\text{BAC}_N} - \sum_{i=1}^{2} \left( \text{Upt}_{\text{NH}_4}^{\text{PhyN}_i} \right) - \text{Upt}_{\text{NH}_4}^{\text{CM}_N} - \text{Upt}_{\text{NH}_4}^{\text{BAC}_N} - \text{Nitrif}_{\text{NH}_4}^{\text{NO}_3}$$

2

$$\frac{\partial PO_4}{\partial t} = \text{Excr}_{PO_4}^{\text{COPP}} + \text{Excr}_{PO_4}^{\text{NCMP}} + \text{Remin}_{PO_4}^{\text{BACP}} - \sum_{i=1}^{2} \left( \text{Upt}_{PO_4}^{\text{PHY}_{Pi}} \right) - \text{Upt}_{PO_4}^{\text{CMP}} - \text{Upt}_{PO_4}^{\text{BACP}}$$
(Eq. S1)

Nitrate concentration results from nitrification and phytoplankton and CM uptakes. Ammonium concentration results from copepods and NCM excretion, bacterial remineralisation, heterotrophic bacteria, phytoplankton, and CM uptakes and losses from nitrification. Finally, phosphate concentration results from copepods and NCM excretion, bacterial remineralisation and heterotrophic bacteria, phytoplankton, and CM uptakes.

**What are the external forces driving the biology of the model ?**

Biology is impacted by temperature and/or irradiance. Depending on the biogeochemical process considered, both or only one of them can have an impact and this impact could be direct or indirect. Nitrification which is performed by nitrifying bacteria (organisms not considered explicitly in the model) depends on temperature and modelled dissolved oxygen concentration:

Nitrif\_{NH\_4}^{NO\_3} = tx\_{NITRIF} \* NH\_4 \* f\_{Q\_{10},nitrif}^T \* \frac{O\_2}{O\_2 + K\_{O\_2}}
$$f_{Q_{10},nitrif}^T = Q_{10,nitrif}^{\frac{T-10}{10}}$$
(Eq. S2)

where  $tx_{NITRIF}$  represents the fraction of  $NH_4^+$  used for nitrification,  $K_{O2}$  is the dissolved oxygen half saturation constant and  $Q_{10,nitrif}$  is the temperature coefficient for nitrification.

We detailed the modelling of planktonic organisms in the companion paper (Barré et al., 2023a). To sum up, heterotrophic bacteria processes are directly impacted by temperature. Phytoplankton and CM processes are directly impacted by temperature (all processes) and irradiance (photosynthesis and grazing). Copepods and NCM processes are indirectly impacted by temperature and irradiance through the consumed preys.

**This is not explained until the discussion, what is extremely frustrating, as we don't really understand what the model sees and feels or not, until the very end, when the author reveals some of the experiment limitations.**

We provided all the balance equations, detailed formulations of biogeochemical processes and parameters values in the companion paper (Barré et al., 2023a). For organisms, we did not find relevant to provide them again. However, we understand the necessity of adding explanations about the nutrients in the manuscript. We decided to add a subsection in Section 2.2.1, in which we detail the modelling of nutrients and organic matter.

**[2.2.1 Nutrients and organic matter representation in the model**

As we use a dimensionless configuration, we assume that nutrients are fully the result of autochthonous biological processes. In other terms, we do not consider allochthonous inputs of nutrients (i.e., from rivers or atmosphere as instance). For all the simulations, nutrients dynamics are represented by the following state equations:

$$\frac{\partial NO_3}{\partial t} = \text{Nitrif}_{NO_3}^{NH_4} - \sum_{i=1}^2 \text{Upt}_{NO_3}^{Phy_{N_i}} - \text{Upt}_{NO_3}^{CM_{N_i}}$$

$$\frac{\partial NH_{4}}{\partial t} = \text{Excr}_{NH_{4}}^{\text{COP}_{N}} + \text{Excr}_{NH_{4}}^{\text{NCM}_{N}} + \text{Remin}_{NH_{4}}^{\text{BAC}_{N}} - \sum_{i=1}^{2} \left( \text{Upt}_{NH_{4}}^{\text{Phy}_{i}} \right) - \text{Upt}_{NH_{4}}^{\text{CM}_{N}} - \text{Upt}_{NH_{4}}^{\text{BAC}_{N}} - \text{Nitrif}_{NH_{4}}^{\text{NO}_{3}}$$
$$\frac{\partial PO_{4}}{\partial t} = \text{Excr}_{PO_{4}}^{\text{COP}_{P}} + \text{Excr}_{PO_{4}}^{\text{NCM}_{P}} + \text{Remin}_{PO_{4}}^{\text{BAC}_{P}} - \sum_{i=1}^{2} \left( \text{Upt}_{PO_{4}}^{\text{PHY}_{Pi}} \right) - \text{Upt}_{PO_{4}}^{\text{CM}_{P}} - \text{Upt}_{PO_{4}}^{\text{BAC}_{P}}$$
(1)

The concentration of NO3- results from nitrification and phytoplankton and CM uptakes. Ammonium concentration results from copepods and NCM excretion, bacterial remineralisation, heterotrophic bacteria, phytoplankton, and CM uptakes and losses from nitrification. Phosphate concentration results from copepods and NCM excretion, bacterial remineralisation and heterotrophic bacteria, phytoplankton, and CM uptakes.

Such as nutrients dynamics, organic matter (dissolved and particulate) dynamic is only the result of autochthonous biological processes (Eq. 2 and 3).

$$\frac{\partial DOC}{\partial t} = \sum_{i=1}^{2} \left( Exu_{DOC}^{PHY_{C_{i}}} \right) + \sum_{i=1}^{2} \left( Exu_{DOC}^{MIX_{C_{i}}} \right) + Excr_{DOC}^{COP_{C}} + Mort_{DOC}^{BAC_{C}} - BP_{DOC}^{BAC_{C}}$$

$$\frac{\partial DON}{\partial t} = \sum_{i=1}^{2} \left( Exu_{DON}^{PHY_{N_{i}}} \right) + \sum_{i=1}^{2} \left( Exu_{DON}^{MIX_{N_{i}}} \right) + Mort_{DON}^{BAC_{N}} - Upt_{DON}^{CM_{N}} - Upt_{DON}^{PICO_{N}} - Upt_{DON}^{BAC_{N}}$$

$$\frac{\partial DOP}{\partial t} = \sum_{i=1}^{2} \left( Exu_{DOP}^{PHY_{P_{i}}} \right) + \sum_{i=1}^{2} \left( Exu_{DOP}^{MIX_{P_{i}}} \right) + Mort_{DOP}^{BAC_{P}} - Upt_{DOP}^{CM_{P}} - Upt_{DOP}^{PICO_{P}} - Upt_{DOP}^{BAC_{P}}$$
(2)

The concentration of dissolved organic carbon (DOC), nitrogen (DON) and phosphorus (DOP) depends on phytoplankton and mixotrophs exudation, copepods excretion, heterotrophic bacteria mortality (natural mortality) and CM, PICO and heterotrophic bacteria uptake.

$$\frac{\partial POC}{\partial t} = E_{POC}^{COP_{C}} + Predation_{POC}^{COP_{C}} - BP_{POC}^{BAC_{C}}$$
$$\frac{\partial PON}{\partial t} = E_{PON}^{COP_{N}} + Predation_{PON}^{COP_{N}} - Upt_{PON}^{BAC_{N}}$$
$$\frac{\partial POP}{\partial t} = E_{POP}^{COP_{P}} + Predation_{POP}^{COP_{P}} - Upt_{POP}^{BAC_{P}}$$

(3)

The concentration of particulate organic carbon (POC), nitrogen (PON) and phosphorus (POP) depends on copepods egestion, predation by higher trophic levels on copepods (closure term of the model) and heterotrophic bacteria production and uptake. POM particles are large enough to sink, however, we do not consider a term to represent their removal from the surface box by sinking. In our case, the POM, such as the DOM, stay in the box and is constantly recycling.

A detailed description and formulations of processes can be found in Barré et al. (2023a). Processes notation description can be found in Table A1 (Appendix A).]

We modified the manuscript consequently (I.111):

[In the following, we provide a detailed description of the carbonate module. We also give a brief description of nutrients and organic matter representation. A detailed description of other compartments, especially of mixotrophs compartment can be found in Barré et al. (2023a).]

**Appendix A:**

[Table A1: Description of state equation processes.

| Notation                                                                                                                                   | Process                                         |  |  |  |
|--------------------------------------------------------------------------------------------------------------------------------------------|-------------------------------------------------|--|--|--|
|                                                                                                                                            | Copepods                                        |  |  |  |
| Excr COP X
NutX € [NH4 + , PO4 3- ]
X € [N, P]                                                      | Excretion of nutrient X by copepods             |  |  |  |
| Excr COP c                                                                                                                      | DOC excretion by copepods                       |  |  |  |
| Resp COP C
DIC                                                                                                               | Copepods respiration                     |  |  |  |
|                                                                                                                                            | Copepods egestion                        |  |  |  |
| Predation COP X
X € [C, N, P]                                                                                                | Predation by higher trophic levels on copepods  |  |  |  |
|                                                                                                                                            | Mixotrophs (Mix e [NCM, CM])             |  |  |  |
| Exu Mix Xi
Dox
X є [C, N, P]                                                                                   | DOX exudation by mixotrophs                     |  |  |  |
| Resp Mix C                                                                                                           | Mixotrophs respiration                   |  |  |  |
| Photo Mix C                                                                                                          | Mixotrophs photosynthesis                       |  |  |  |
| Excr NutX
NutX € [NH4 + , PO4 3- ]
X € [N, P]                                                       | Excretion of nutrient X by NCM                  |  |  |  |
| Upt CM X
X € [N, P]
NutX € [NO 3 - , NH 4 + , PO 4 3- ] | Uptake of nutrient X by constitutive mixotrophs |  |  |  |
| $Upt_{DOX}^{CM_X}$                                                                                                                         | Uptake of DOX by constitutive mixotrophs        |  |  |  |
|                                                                                                                                            | Phytoplankton (Phy $\in$ [NMPHYTO, PICO])       |  |  |  |
| Resp Phy C                                                                                                                      | Phytoplankton respiration                       |  |  |  |
| Photo Phy C
DIC                                                                                                              | Phytoplankton photosynthesis                    |  |  |  |
| Upt Phy X
NutX
NutX ε [NO3 - , NH4 + , PO4 3- ]                                          | Uptake of nutrient X by phytoplankton           |  |  |  |
| Exu Phy X
DOX
X { [C, N, P]                                                                                               | DOX exudation by phytoplankton                  |  |  |  |
| $ \begin{array}{l} Upt_{DOX}^{PICO_X} \\ X \in [N, P] \end{array} \end{array} $                                                            | Uptake of DOX by picophytoplankton              |  |  |  |
| Heterotrophic bacteria                                                                                                                     |                                                 |  |  |  |
| $BP_X^{BAUC}$
X $\epsilon$ [DOC, POC]                                                                                                   | Bacterial production                            |  |  |  |
| BR BAC C                                                                                                                        | Bacterial respiration                           |  |  |  |
| $Upt_{POX}^{BAC_X} X \in [N, P]$                                                                                                           | POX uptake by heterotrophic bacteria            |  |  |  |

| $Exu_{DOX}^{Phy_{X_i}}$
X $\epsilon$ [C, N, P]                                                     | DOX exudation by phytoplankton                           |  |
|-------------------------------------------------------------------------------------------------------|----------------------------------------------------------|--|
| $\begin{array}{l} Remin_{BAC_{X}}^{NutX} \\ NutX \in [NH4^{+}, PO4^{3-}] \\ X \in [N, P] \end{array}$ | Remineralisation of nutrient X by heterotrophic bacteria |  |
| Mort BAC x                                                                                 | Heterotrophic bacteria natural mortality                 |  |
|                                                                                                       | Dissolved inorganic matter (DIM)                         |  |
| Diss CaCO 3                                                                     | CaCO 3 dissolution                            |  |
| Prec CaCO3                                                                      | CaCO 3 precipitation                          |  |
| Nitrif                                                                                                | Nitrification                                            |  |
| Aera DIC                                                                                   | Air-sea CO 2 gas exchanges (aeration)         |  |

Also, we propose to add the figure 2, which resumes the hypothesis used in this study and allows to visualize the 0D concept, at the end of part 2.2.2 (2.2.3 in the revised manuscript).

Figure 2: Schematic representation of 0D concept and summary of the hypotheses used in this study with Eco3M\_MIX-CarbOx. T: temperature, S: Salinity and OM: Organic matter.

We modify the manuscript consequently:

]

(I.181): Figure X illustrates the concept of 0D and summarize the hypothesis used in this study with Eco3M\_MIX-CarbOx.

3) Still about the 0D, what happens to the POM ? Do they sink ? Are they removed from the surface box ? Or do they float there and are slowly remineralized (as if the bay is mixed enough to keep the particles around) ? This is important as it has an impact on TA and DIC and all other nutrients concentration.

In the model, the POC, PON and POP dynamics result from copepods egestion, higher trophic level predation on copepods (closure terms of the model) and heterotrophic bacteria uptake (Eq. S3). They do not sink and are constantly recycling in the surface box using the processes indicated above.

 $\frac{\partial POC}{\partial t} = E_{POC}^{COP_{C}} + Predation_{POX}^{COP_{X}} - BP_{POC}^{BAC_{C}}$  $\frac{\partial PON}{\partial t} = E_{PON}^{COP_{N}} + Predation_{PON}^{COP_{N}} - Upt_{PON}^{BAC_{N}}$  $\frac{\partial POP}{\partial t} = E_{POP}^{COP_{P}} + Predation_{POP}^{COP_{P}} - Upt_{POP}^{BAC_{P}}$

(Eq. S3)

Using these state equations, the POM compartment is balanced (Fig. S1) which is why we do not consider a term to represent their removal from the surface box by sinking.

---

## Author Comment (AC2)

First, we would like to thank referee 2 for his careful evaluation of our manuscript and his interesting comments which we believe will help us to improve it. Please, find hereafter our response to these comments.

1) Firstly, I would say that it does not fit the scope of GMD, which is there to present new developments in models. While the companion paper, with its presentation of a mixotroph compartment, meets this criterium, the main new thing in this manuscript is a diagnostic relation between TA and salinity, which may improve the results, but conceptually is a fairly small step and has been used in many different models so far. From this side I would rather recommend publication in a different journal, where the focus is more on the considered system itself, i.e. a model for the BoM.

We understand your concern, however we believe that this manuscript has its place in GMD. Eco3M\_MIX-CarbOx is a new model which has been developed to consider both mixotrophs and carbonate system. We decided to present Eco3M\_MIX-CarbOx in two parts to show both sides of the model distinctly. It allowed us to propose clear studies, easier to read than one which would have been longer, and at the same time to highlight the two main developments which, together, constitute the originality of our model. However, it is important to keep in mind that both parts of the study aim to present this new model. We believe that the study, as a whole (both parts), fits well the scope of GMD and especially, meet the criteria of 'model description papers'.

Moreover, both studies are strongly linked. In the first part, we focused on the planktonic ecosystem description, especially on mixotrophs. We detailed their implementation in the model and study their dynamics in the area. In the second part, we focused on the carbonate system which is barely mentioned in the first part and detailed its representation in the model. This second study is based on the first one as we present a representation of carbonate system by considering the impact of mixotrophs (photosynthesis, respiration...) on these variables. We think that the strong connection between both studies also justify their publication in the same journal.

**2) The switch to a salinity-TA relationship is motivated by the desire to represent the episodic intrusion of freshwater from the nearby Rhone into the BoM, and also the influence of evaporation and precipitation. My first question here is: If these freshwater fluxes affect the TA balance so strongly, should they not also influence DIC?**

First, we would like to stress the fact that "an excess of alkalinity" which likely reflects alkalinity inputs to coastal areas has been described for the entire Mediterranean Sea (Schneider et al., 2007). This study, at the global scale, has forged our conviction that, in a coastal area close to the Rhone River, alkalinity inputs from Rhone River needed to be considered even in this 0D configuration.

The switch to a salinity-TA relationship is possible thanks to the fact that in the bay of Marseille, TA variations are mainly the results of rivers contributions, particularly the Rhône River one. It was demonstrated by the lack of variations observed when we modelled TA only based on biogeochemical processes which take place in the box. For DIC, a different reasoning must be adopted, mainly because the processes which impact DIC dynamics are very different than the one which impact TA dynamic. As we consider a surface layer, DIC dynamics is mostly the results of temperature and salinity changes (which are considered by the model) and biogeochemical processes (especially air-sea CO2 exchanges) (Hassoun et al., 2015). The Rhône River can bring DIC to the BoM, these inputs are diluted (far from 2877 µmol kg-1, the value observed in the Rhône River, Table S1) and, due to the action of other processes (solubility effects and biogeochemical processes) on DIC dynamics, which is more pronounced in this case, have a less significant impact than on TA dynamics.

Moreover, it is important to note that, in Eco3M\_MIX-CarbOx, TA is the main driver of carbonate system. In other words, a change in TA results in significant changes in DIC,  $pH_T$  and  $pCO_2$  as demonstrated by the Figure 4 of the manuscript. By representing the contribution of the Rhône River on TA we then indirectly apply it to the three other variables of the carbonate system. We are aware that the contribution of the Rhône River to the DIC considered in that way (through TA) is only an indirect effect. It is important to highlight that in the case of our OD configuration, it is not always possible to consider allochthonous contributions. So far, we experimented two ways to consider external contributions in Eco3M\_MIX-CarbOx, which gave satisfactory results:

- By using a VAR=f(S) relation (done in this paper for TA),
- By forcing the variable with a file which include an interpolation of the measurements performed at SOLEMIO for this variable (done in our first paper for nutrient concentrations)

Using a DIC=f(S) relation did not produce the expected results. In fact, it is not recommended to use such a formulation to represent DIC dynamics, especially in surface water due to the numerous processes from which its dynamic results (Hassoun et al., 2015). Using an interpolation of SOLEMIO DIC measurements force us to represent the variable dynamics as a forcing, which had no real interest here given that the representation of DIC obtained was already rather correct when based on biogeochemical processes.

**Table S1**: Salinity-DIC couples for LSE events measured at SOLEMIO between 6 June 2016 and 26 June2019 (last data available).

| Salinity | DIC (µmol.kg -1 ) |
|----------|------------------------------|
| 37.11    | 2321.3                       |
| 37.78    | 2280.1                       |
| 37.30    | 2259.3                       |
| 36.82    | 2323.8                       |
| 37.62    | 2288.7                       |
| 37.18    | 2260.3                       |
| 37.66    | 2269.8                       |
| 37.32    | 2249.0                       |

**and nutrients as well?**

**Figure S1:** Time series of surface (a) salinity (CARRY measurements), and interpolated (b)  $NO_3^-$  concentration, (c)  $NH_4^+$  concentration and (d)  $PO_4^{3-}$  concentration at SOLEMIO station. SOLEMIO data are represented by blue markers and the four LSE are indicated by the red dotted lines.

Rhône River intrusion events are associated with an increase of nutrient concentrations in the area, especially nitrate and phosphate (Fraysse et al., 2014). However, in our case, the four low salinity events are not systematically associated with a nutrient increase at the station. In fact, only the first and last events (15 March and 5 September respectively) have an impact on nutrient concentrations at SOLEMIO with the first event being the most significant (Fig. S1). This pattern can be explained by the salinity data used by the model. The measurements are performed at CARRY station (near the Côte Bleue, see figure 1 of the manuscript for location) which is more significantly impacted by the Rhône River plume as it is closer to the river mouth than SOLEMIO station. In consequence, decreases of salinity measurements are also performed at SOLEMIO, however their temporal resolution is low (fortnightly measurements) compared to the CARRY one (hourly measurements). In fact, Rhône River intrusion events duration is variable and can be less than 15 days (ex: short-lived intrusions, Fraysse et al., 2014), therefor it is important to consider the highest temporal resolution possible to better catch them with measurements which is why we chose to work with CARRY measurements instead of SOLEMIO measurements (Fig. S1a).

**That this may lead to biases is discussed in lines 477 to 484; but given the extremely high DIC concentration in Rhone water quoted on line 482, I wonder whether this inconsistency may not invalidate the main results.**

We understand your concerns, however, as indicated in the previous point, the strong DIC value (2877  $\mu$ mol.kg-1) observed in the Rhône River never reaches SOLEMIO as the Rhône River plume is quickly diluted and the DIC values which really reaches SOLEMIO station are rarely higher than 2300  $\mu$ mol.kg-1 (mean value of 2281.5  $\mu$ mol.kg-1, Table S1).

Moreover, as indicated in the previous point, it is important to note that in Eco3M\_MIX-CarbOx, TA is the main driver of carbonate system. In other words, a change in TA results in significant changes in DIC, pH and pCO2 as demonstrated by the Figure 4 of the manuscript. By representing the contribution of the Rhône River on TA we then indirectly apply it to the three other variables of the carbonate system.

3) This leads me to a more conceptual difficulty with the approach. The model concept is that of an arbitrary one cubic metre volume at the surface of the bay, and that the model just represents fluxes within this volume. Spatial fluxes are excluded (except for CO2 flux, more on that below). This only allows either to model a variable as purely forced from what is happening inside the box, or to prescribe it, e.g. as a function of salinity. For a proper modelling of how external fluxes (e.g. in mol/s) change concentrations (mol/m^3/s) inside the modelled region, one would have to define the volume that is affected by these fluxes. A reasonable choice might be to model a column of water within the mixed layer, as was done in many zero-dimensional models, e.g. Fasham et al, 1990 or Hurtt and Armstrong 1999. That would allow a consistent treatment of the effects of mixing on TA, DIC, nutrients.

This difficulty becomes especially clear when the authors discuss the possible reasons for their low net annual air-sea flux of CO2, which is in contrast to observation-based estimates. Here they state that "aeration is is simulated by applying Eq (5) to 1 m^3 of surface water at the SOLEMIO station, which tends to overestimate the effect of aeration processes on DIC..." (line 534 ff). Indeed: if the control volume is that shallow, it will be lead to a too fast approach of DIC towards equilibrium, and hence an underestimate of fluxes.

We understand your concern. We think that it is important to note that, in this study, we relied on Eco3M-CarbOx (Lajaunie-Salla et al., 2021) for the calculation of carbonate variables. So, Eco3M-CarbOx was our starting point to implement carbonate system variables in Eco3M\_MIX-CarbOx. With our study, we aim to bring answers to the concerns raised by this previous study, we then use the same concept and try to improve the representation of carbonate cycle variables (by adding mixotrophs organisms processes to the state equations and switching TA formulation by a newly implemented allochthonous formulation). In this way, we were able to compare both models, then knowing how our modifications impact the carbonate system variables representation. Even if we did not manage to obtain a realistic representation of air-sea CO2 fluxes, we provide some improvements to the initial concept and give some examples of suitable and unsuitable use of it (first part and second part of this study respectively), then confirming that the only way to obtain realistic fluxes is to consider a larger layer. We could have done it the way you propose but, considering that it required a complete review of our OD configuration, we decided to focus directly on the coupling of Eco3M\_MIX-CarbOx, in 3D (which is still in test phase).

4) And finally, while the diagnostic TA leads to an improvement in model results, as evidenced by decreases in %BIAS, RMSD and a 'cost function' presented in Table 3, the improvements are fairly modest. Indeed I would be interested in knowing whether a model with TA prescribed constant at the average of observations would not have fared at least similarly good as the two presented model cases.

As suggested, we run a simulation with a constant TA (mean of SOLEMIO measurements for  $2017 = 2591.2 \mu mol kg^{-1}$ ). In Table S2, we presented the calculation of the statistical indicators presented in the manuscript. As we decided to modify them, based on your suggestions, we also provided average absolute error (AAE), and average error (AE) calculated as in Stow et al. (2009) except that, to be consistent with calculations of statistical indicators used previously (Allen et al., 2007) the difference is applied between observations and model which means that for %BIAS and AE, if a positive value is obtained the model underestimates the observations.

**Table S2**: Statistical indicators calculation for the simulation with a constant TA (TA = 2591.2  $\mu$ mol kg-1). Mean, SD, AE, AAE and RMSD are in the same unit than the considered variable, i.e.:  $\mu$ mol kg-1 for TA and DIC and  $\mu$ atm for *p*CO2. CF and %BIAS are without unit.

|           |             | ТА            | DIC           | pCO₂         | pH⊤         |
|-----------|-------------|---------------|---------------|--------------|-------------|
| N         | Observation | 20            | 20            | 20           | 20          |
| Mean ± SD | Observation | 2591.2 ± 19.4 | 2294.9 ± 24.0 | 391.0 ± 31.0 | 8.09 ± 0.03 |
| Mean ± SD | Model       | 2591.2 ± 0.22 | 2305.7 ± 26.1 | 418.0 ± 28.9 | 8.07 ± 0.03 |
| CF        | Model       | 0.85          | 0.82          | 1.14         | 1.14        |
| %BIAS     | Model       | -0.002        | -0.50         | -5.79        | 0.26        |
| RMSD      | Model       | 18.90         | 26.14         | 38.45        | 0.03        |
| AAE       | Model       | 16.5          | 19.7          | 35.5         | 0.03        |
| AE        | Model       | -0.06         | -11.5         | -22.6        | 0.02        |

We also represented the daily mean values of TA, DIC,  $pH_T$  and  $pCO_2$  for the simulations SIMCO, SIMC1 and constant TA (Fig. S2) to compare the three simulations carbonate system variables representation.

---

## Author Response (AR1)

First, we would like to thank Julien Palmiéri for his careful evaluation of our manuscript and his interesting comments which we believe will help us to improve it. Please, find hereafter our response to these comments.

Major comments

**1) First, the reason of the choice is not given. I understand it is greatly needed for developments like adding the mixotrophs in Eco3M. It's a huge task that require such lightweight configuration to test, verify that nothing is broken, and make sure the fluxes between each element of the model are reasonable. So, this choice is easily explained for the part 1, but less for this one.**

We agree with this point. The use of a 0D configuration may seem more justified for the first part of our study as adding mixotrophs required many tests.

We chose to also use a dimensionless configuration for the part II as we wanted to provide a reliable representation of the carbonate system, which consider mixotrophs organisms, in the simplest way possible. In other words, we wanted to provide a tool which is easy to use, easy to adapt to other coastal area (by modifying the environmental forcings and the AT-S correlation) and give reliable results in a short amount of time.

Another reason to use a dimensionless configuration rather than a 3D configuration was the possibility to compare Eco3M_MIX-CarbOx to Eco3M-CarbOx in the same type of implementation. By comparing the models, we can answer several questions and especially: How mixotrophs affect carbonate variables ? Does adding them provide a more accurate representation of these variables ? and then build further relevant simulation strategies for 3D.

Considering these reasons, this dimensionless configuration can be seen as a laboratory to test and then build further relevant simulation strategies for 3D.

To justify the use of a 0D configuration, we first clarified the aim of the study at the end of the introduction (l.79):

[Here we try to provide a more realistic representation of carbonate system variables in the BoM. As a starting point, we used the concept of the dimensionless Eco3M-CarbOx model (Lajaunie-Salla et al., 2021), which aims to represent a small volume of surface water (i.e., 1 m$^3$ ) in the BoM. We developed a planktonic ecosystem model which contains, among others, mixotrophic organisms, modified the carbonate module described by Lajaunie-Salla et al. (2021) and added it to our newly developed planktonic ecosystem model to obtain the Eco3M_MIX-CarbOx model (v1.0). We implemented two types of TA formulation and compared the simulation results to in situ observations to identify which formulation was capable to deliver the more realistic results: (i) a formulation that only considers biological processes (referred to as autochthonous formulation) and (ii) a new TA formulation that depends only on salinity (referred to as allochthonous formulation). Furthermore, we simulate air-sea $CO_2$ fluxes to determine whether the BoM act as a sink or a source of $CO_2$ and provide a detailed analysis of drivers of seawater $p$$CO_2$ variations for two specific hydrodynamic processes typical for the BoM: (i) Rhône River intrusion and (ii) summer upwelling events. **With this study, we aim to provide a new tool which allow to obtain a reliable representation of the carbonate system in the simplest way as possible: by using a dimensionless configuration which is easy to use, adapt and give results in a short amount of time.**]

And we gave a justification of this choice in the Section 2.2 (l.111):

[In this study, we used the Eco3M_MIX-CarbOx model (v1.0) which was developed to represent the dynamics of the seawater carbonate system and mixotrophs in the BoM and was implemented using the Eco3M (Ecological Mechanistic and Molecular Modelling) platform (Baklouti et al., 2006a, b). **Eco3M_MIX-CarbOx is a dimensionless model (0D): we consider a volume of 1 m³ of surface water at SOLEMIO station, in this volume the state variables only vary over time as the model is not coupled with a hydrodynamic model. We chose to use a 0D configuration as this configuration has several advantages, especially, calculation times are low (around 45 minutes in our case). It allows us to make several test simulations to better understand the biogeochemical functioning of the BoM and its possible reactions to environmental forcings.**]

**In the introduction, it's made mention of the need for high resolution model for coastal regional study, and the next sentence announce the use of a dimensionless (0D hereafter) configuration.**

We understand that these sentences can be confusing. We add a sentence to make a better link between 3D introduction and use of 0D (l.65).

[Most modelling approaches to investigate carbonate system variables typically employ 3D coupled physical-biogeochemical models and focus on larger coastal areas (e.g., Artioli et al., 2014; Bourgeois et al., 2016). If the focus is on smaller areas this requires higher spatial and temporal resolution to correctly represent the relevant processes (Bourgeois et al., 2016). **However, higher spatial and temporal resolution often result in a significant increase of the calculation time which make more difficult the repetition of numerical experiments, an important step to better understanding the global functioning of the area and its reaction to environmental forcings. A solution to avoid important calculation times is to use a dimensionless model. This type of model allows to conduct large amount of test in short amount of time. For instance,** Lajaunie-Salla et al. (2021) used the dimensionless Eco3M-CarbOx model, which contains a carbonate module performing the resolution of the carbonate system based on total alkalinity (TA) and dissolved inorganic carbon (DIC). Even if the DIC, oceanic partial pressure of $CO_2$ ($pCO_2$) and total pH ($pH_T$) representations look reliable, Eco3m-CarbOx tends to minimize the range of TA variations during the year, resulting in a near constant TA (Lajaunie-Salla et al., 2021).]

**2) A second point is that this 0D brings more questions than answers. Because it is a surface box, the model does not represent advection and mixing. The physics variable/forcing come from observations and hence include annual cycle and external forcing, including specific phenomena like summer upwellings or the Rhône waters passing by. But what about the nutrients ?**

We agree that hydrodynamic processes, especially upwelling and Rhône River intrusions in the BoM are associated with nutrients inputs. In Eco3M_MIX-CarbOx, upwellings are only represented by strong variations of temperature and Rhône River intrusions are only represented by strong variations of salinity and TA inputs when allochthonous TA formulation is used. In other words, we do not consider the possible inputs of nutrients associated with these events and assumed that nutrients are fully the result of autochthonous biological processes (due to 0D configuration) which means that they are modelled based on the following state equations :

$$\frac{\partial NO_3}{\partial t} = \text{Nitrif}_{NO_3}^{NH_4} - \sum_{i=1}^{2} \text{Upt}_{NO_3}^{Phy_{N_i}} - \text{Upt}_{NO_3}^{CM_{N_i}}$$

$$\frac{\partial NH_4}{\partial t} = \text{Excr}_{NH_4}^{COP_N} + \text{Excr}_{NH_4}^{NCM_N} + \text{Remin}_{NH_4}^{BAC_N} - \sum_{i=1}^{2}\left(\text{Upt}_{NH_4}^{Phy_{N_i}}\right) - \text{Upt}_{NH_4}^{CM_N} - \text{Upt}_{NH_4}^{BAC_N} - \text{Nitrif}_{NH_4}^{NO_3}$$

$$\frac{\partial PO_4}{\partial t} = Excr_{PO_4}^{COP_P} + Excr_{PO_4}^{NCM_P} + Remin_{PO_4}^{BAC_P} - \sum_{i=1}^{2} \left( Upt_{PO_4}^{PHY_{P_i}} \right) - Upt_{PO_4}^{CM_P} - Upt_{PO_4}^{BAC_P}$$

(Eq. S1)

Nitrate concentration results from nitrification and phytoplankton and CM uptakes. Ammonium concentration results from copepods and NCM excretion, bacterial remineralisation, heterotrophic bacteria, phytoplankton, and CM uptakes and losses from nitrification. Finally, phosphate concentration results from copepods and NCM excretion, bacterial remineralisation and heterotrophic bacteria, phytoplankton, and CM uptakes.

**What are the external forces driving the biology of the model ?**

Biology is impacted by temperature and/or irradiance. Depending on the biogeochemical process considered, both or only one of them can have an impact and this impact could be direct or indirect. Nitrification which is performed by nitrifying bacteria (organisms not considered explicitly in the model) depends on temperature and modelled dissolved oxygen concentration:

$$Nitrif_{NH_4}^{NO_3} = tx_{NITRIF} * NH_4 * f_{Q_{10},nitrif}^{T} * \frac{O_2}{O_2 + K_{O_2}}$$

$$f_{Q_{10},nitrif}^{T} = Q_{10,nitrif}^{\frac{T-10}{10}}$$

(Eq. S2)

where $tx_{NITRIF}$ represents the fraction of $NH_4^+$ used for nitrification, $K_{O2}$ is the dissolved oxygen half saturation constant and $Q_{10,nitrif}$ is the temperature coefficient for nitrification.

We detailed the modelling of planktonic organisms in the companion paper (Barré et al., 2023a). To sum up, heterotrophic bacteria processes are directly impacted by temperature. Phytoplankton and CM processes are directly impacted by temperature (all processes) and irradiance (photosynthesis and grazing). Copepods and NCM processes are indirectly impacted by temperature and irradiance through the consumed preys.

**This is not explained until the discussion, what is extremely frustrating, as we don't really understand what the model sees and feels or not, until the very end, when the author reveals some of the experiment limitations.**

We provided all the balance equations, detailed formulations of biogeochemical processes and parameters values in the companion paper (Barré et al., 2023a). For organisms, we did not find relevant to provide them again. However, we understand the necessity of adding explanations about the nutrients in the manuscript. We decided to add a subsection in Section 2.2.1, in which we detail the modelling of nutrients and organic matter.

**[2.2.1 Nutrients and organic matter representation in the model**

**As we use a dimensionless configuration, we assume that nutrients are fully the result of autochthonous biological processes. In other terms, we do not consider allochthonous inputs of nutrients (i.e., from rivers or atmosphere as instance). For all the simulations, nutrients dynamics are represented by the following state equations:**

$$\frac{\partial NO_3}{\partial t} = Nitrif_{NO_3}^{NH_4} - \sum_{i=1}^{2} Upt_{NO_3}^{Phy_{N_i}} - Upt_{NO_3}^{CM_{N_i}}$$

$$\frac{\partial NH_4}{\partial t} = Excr_{NH_4}^{COP_N} + Excr_{NH_4}^{NCM_N} + Remin_{NH_4}^{BAC_N} - \sum_{i=1}^{2}\left(Upt_{NH_4}^{Phy_{N_i}}\right) - Upt_{NH_4}^{CM_N} - Upt_{NH_4}^{BAC_N} - Nitrif_{NH_4}^{NO_3}$$

$$\frac{\partial PO_4}{\partial t} = Excr_{PO_4}^{COP_P} + Excr_{PO_4}^{NCM_P} + Remin_{PO_4}^{BAC_P} - \sum_{i=1}^{2}\left(Upt_{PO_4}^{PHY_{P_i}}\right) - Upt_{PO_4}^{CM_P} - Upt_{PO_4}^{BAC_P}$$

(1)

The concentration of NO$_3^-$ results from nitrification and phytoplankton and CM uptakes. Ammonium concentration results from copepods and NCM excretion, bacterial remineralisation, heterotrophic bacteria, phytoplankton, and CM uptakes and losses from nitrification. Phosphate concentration results from copepods and NCM excretion, bacterial remineralisation and heterotrophic bacteria, phytoplankton, and CM uptakes.

Such as nutrients dynamics, organic matter (dissolved and particulate) dynamic is only the result of autochthonous biological processes (Eq. 2 and 3).

$$\frac{\partial DOC}{\partial t} = \sum_{i=1}^{2}\left(Exu_{DOC}^{PHY_{C_i}}\right) + \sum_{i=1}^{2}\left(Exu_{DOC}^{MIX_{C_i}}\right) + Excr_{DOC}^{COP_C} + Mort_{DOC}^{BAC_C} - BP_{DOC}^{BAC_C}$$

$$\frac{\partial DON}{\partial t} = \sum_{i=1}^{2}\left(Exu_{DON}^{PHY_{N_i}}\right) + \sum_{i=1}^{2}\left(Exu_{DON}^{MIX_{N_i}}\right) + Mort_{DON}^{BAC_N} - Upt_{DON}^{CM_N} - Upt_{DON}^{PICO_N} - Upt_{DON}^{BAC_N}$$

$$\frac{\partial DOP}{\partial t} = \sum_{i=1}^{2}\left(Exu_{DOP}^{PHY_{P_i}}\right) + \sum_{i=1}^{2}\left(Exu_{DOP}^{MIX_{P_i}}\right) + Mort_{DOP}^{BAC_P} - Upt_{DOP}^{CM_P} - Upt_{DOP}^{PICO_P} - Upt_{DOP}^{BAC_P}$$

(2)

The concentration of dissolved organic carbon (DOC), nitrogen (DON) and phosphorus (DOP) depends on phytoplankton and mixotrophs exudation, copepods excretion, heterotrophic bacteria mortality (natural mortality) and CM, PICO and heterotrophic bacteria uptake.

$$\frac{\partial POC}{\partial t} = E_{POC}^{COP_C} + Predation_{POC}^{COP_C} - BP_{POC}^{BAC_C}$$

$$\frac{\partial PON}{\partial t} = E_{PON}^{COP_N} + Predation_{PON}^{COP_N} - Upt_{PON}^{BAC_N}$$

$$\frac{\partial POP}{\partial t} = E_{POP}^{COP_P} + Predation_{POP}^{COP_P} - Upt_{POP}^{BAC_P}$$

(3)

The concentration of particulate organic carbon (POC), nitrogen (PON) and phosphorus (POP) depends on copepods egestion, predation by higher trophic levels on copepods (closure term of the model) and heterotrophic bacteria production and uptake. POM particles are large enough to sink, however, we do not consider a term to represent their removal from the surface box by sinking. In our case, the POM, such as the DOM, stay in the box and is constantly recycling.

A detailed description and formulations of processes can be found in Barré et al. (2023a). Processes notation description can be found in Table A1 (Appendix A).]

We modified the manuscript consequently (l.111):

[In the following, we provide a detailed description of the carbonate module. **We also give a brief description of nutrients and organic matter representation.** A detailed description of other compartments, especially of mixotrophs compartment can be found in Barré et al. (2023a).]

Appendix A:

[Table A1: Description of state equation processes.

| Notation | Process |
|---|---|
| **Copepods** | |
| $Excr_{NutX}^{COP_X}$
 NutX ϵ [$NH_4^+$, $PO_4^{3-}$]
 X ϵ [N, P] | Excretion of nutrient X by copepods |
| $Excr_{DOC}^{COP_C}$ | DOC excretion by copepods |
| $Resp_{DIC}^{COP_C}$ | Copepods respiration |
| $E_{POX}^{COP_X}$
 X ϵ [C, N, P] | Copepods egestion |
| $Predation_{POX}^{COP_X}$
 X ϵ [C, N, P] | Predation by higher trophic levels on copepods |
| **Mixotrophs (Mix ϵ [NCM, CM])** | |
| $Exu_{DOX}^{Mix_{X_i}}$
 X ϵ [C, N, P] | DOX exudation by mixotrophs |
| $Resp_{DIC}^{Mix_C}$ | Mixotrophs respiration |
| $Photo_{DIC}^{Mix_C}$ | Mixotrophs photosynthesis |
| $Excr_{NutX}^{NCM_X}$
 NutX ϵ [$NH_4^+$, $PO_4^{3-}$]
 X ϵ [N, P] | Excretion of nutrient X by NCM |
| $Upt_{NutX}^{CM_X}$
 X ϵ [N, P]
 NutX ϵ [$NO_3^-$, $NH_4^+$, $PO_4^{3-}$] | Uptake of nutrient X by constitutive mixotrophs |
| $Upt_{DOX}^{CM_X}$
 X ϵ [N, P] | Uptake of DOX by constitutive mixotrophs |
| **Phytoplankton (Phy ϵ [NMPHYTO, PICO])** | |
| $Resp_{DIC}^{Phy_C}$ | Phytoplankton respiration |
| $Photo_{DIC}^{Phy_C}$ | Phytoplankton photosynthesis |
| $Upt_{NutX}^{Phy_X}$
 NutX ϵ [$NO_3^-$, $NH_4^+$, $PO_4^{3-}$] | Uptake of nutrient X by phytoplankton |
| $Exu_{DOX}^{Phy_X}$
 X ϵ [C, N, P] | DOX exudation by phytoplankton |
| $Upt_{DOX}^{PICO_X}$
 X ϵ [N, P] | Uptake of DOX by picophytoplankton |
| **Heterotrophic bacteria** | |
| $BP_X^{BAC_C}$
 X ϵ [DOC, POC] | Bacterial production |
| $BR_{DIC}^{BAC_C}$ | Bacterial respiration |
| $Upt_{POX}^{BAC_X}$
 X ϵ [N, P] | POX uptake by heterotrophic bacteria |

| | |
|---|---|
| $\text{Exu}_{DOX}^{Phy_{X_i}}$
 $X \in [C, N, P]$ | **DOX exudation by phytoplankton** |
| $\text{Remin}_{BAC_X}^{NutX}$
 $NutX \in [NH_4^+, PO_4^{3-}]$
 $X \in [N, P]$ | **Remineralisation of nutrient X by heterotrophic bacteria** |
| $\text{Mort}_{DOX}^{BAC_X}$ | **Heterotrophic bacteria natural mortality** |
| | **Dissolved inorganic matter (DIM)** |
| $\text{Diss}_{DIC}^{CaCO_3}$ | **CaCO₃ dissolution** |
| $\text{Prec}_{DIC}^{CaCO_3}$ | **CaCO₃ precipitation** |
| **Nitrif** | **Nitrification** |
| $\text{Aera}_{DIC}$ | **Air-sea CO₂ gas exchanges (aeration)** |

]

Also, we propose to add the figure 2, which resumes the hypothesis used in this study and allows to visualize the 0D concept, at the end of part 2.2.2 (2.2.3 in the revised manuscript).

[Figure]

**Figure 2: Schematic representation of 0D concept and summary of the hypotheses used in this study with Eco3M_MIX-CarbOx. T: temperature, S: Salinity and OM: Organic matter.**

We modify the manuscript consequently:

(l.181): **Figure X illustrates the concept of 0D and summarize the hypothesis used in this study with Eco3M_MIX-CarbOx.**

**3) Still about the 0D, what happens to the POM ? Do they sink ? Are they removed from the surface box ? Or do they float there and are slowly remineralized (as if the bay is mixed enough to keep the particles around) ? This is important as it has an impact on TA and DIC and all other nutrients concentration.**

In the model, the POC, PON and POP dynamics result from copepods egestion, higher trophic level predation on copepods (closure terms of the model) and heterotrophic bacteria uptake (Eq. S3). They do not sink and are constantly recycling in the surface box using the processes indicated above.

$$\frac{\partial \text{POC}}{\partial t} = \text{E}_{\text{POC}}^{\text{COP}_\text{C}} + \text{Predation}_{\text{POX}}^{\text{COP}_\text{X}} - \text{BP}_{\text{POC}}^{\text{BAC}_\text{C}}$$

$$\frac{\partial \text{PON}}{\partial t} = \text{E}_{\text{PON}}^{\text{COP}_\text{N}} + \text{Predation}_{\text{PON}}^{\text{COP}_\text{N}} - \text{Upt}_{\text{PON}}^{\text{BAC}_\text{N}}$$

$$\frac{\partial \text{POP}}{\partial t} = \text{E}_{\text{POP}}^{\text{COP}_\text{P}} + \text{Predation}_{\text{POP}}^{\text{COP}_\text{P}} - \text{Upt}_{\text{POP}}^{\text{BAC}_\text{P}}$$

(Eq. S3)

Using these state equations, the POM compartment is balanced (Fig. S1) which is why we do not consider a term to represent their removal from the surface box by sinking.

[Figure]

**Figure S1**: Time-series of daily averaged (a) POC, (b) PON and (c) POP for the three years of simulation (repetition of 2017 three times) for the reference simulation (SIMC0, Table 2 of the manuscript).

We hope that adding the part 2.2.1 (see point 2) which contains a description of dissolved and particulate organic matter representation in the model will clarify this point.

**4) Somehow it looks like (and I am sorry to say that, but I am sure you agree with me) the work you've done here (changing from autochthonous TA formulation – what is what you ideally want to use – to the abiotic, allochthonous formulation) is a way to fix a problem due to the configuration choice, that is not done for this kind of study. Your conclusion (you need to switch from 0D to 3 or at least 1D) should have been one of Lajaunie-Salla et al. 2021's study.**

We agree, by changing the formulation of TA from autochthonous to allochthonous, our aim was to correct a bias due to the configuration choice.

By implementing an allochthonous formulation of TA in Eco3M_MIX-CarbOx, we first wanted to explain the result obtained by Lajaunie-Salla et al. (2021) and make sure that this result was not due to a poor representation of a biological process that could affect TA dynamics. Then, with the results obtained in our study, we could confirm that the lack of variation in the AT representation of Lajaunie-Salla et al.

(2021) is explained by the fact that the 0D configuration does not allow (at first sight) the consideration of allochthonous contributions and especially of the Rhône River which is the main source of TA variations in the area.

Nevertheless, the 0D configuration has several advantages, including its short calculation time which allowed us to provide a detailed analysis of drivers of seawater $pCO_2$ variations, particularly during specific hydrodynamic processes typical for the BoM. This type of study is still uncommon in the area, as few of them investigated the carbonate system dynamics, especially the $pCO_2$ variations drivers (reference study: Wimart-Rousseau et al., 2020) and would have been more difficult to conduct in 3D. That is why we chose to keep working on the 0D configuration (therefore, looking for a way to better represent the TA on which $pCO_2$ calculation depends) and present the results obtained with this configuration as we think that, even if some points deserve to be reworked in 3D, the 0D already allows to obtain interesting results.

**5) Before publication I would require the author to better explain the choice and implications of the 0D in the method section, so that the reader can really understand the experiments and the results.**

**- How are the nutrients managed (initialized with annual average value)?**

Nutrients dynamics are the results of biological processes which take place in the box only, please see point 2 for details.

**- Are total N, P, SI, Fe, Alk supposed to be conserved within the box ? Or are they allowed to fluctuate with some external sources and sinks from/to outside the box, apart from the air-sea CO₂ flux?**

We do not consider Si and Fe in our model. Total N, and P are supposed to be conserved within the box as we do not consider any external source or sink from/to the water column (Fig. S2).

[Figure]

**Figure S2**: Time-series of daily averaged total nitrogen ($N_{TOT}$) and phosphorus ($P_{TOT}$) for the three years of simulation (repetition of 2017 three times) for the reference simulation (SIMC0, Table 2 of the manuscript).

When TA calculation is based on autochthonous formulation, TA is supposed to be conserved in the box. However, when we repeat the year 2017 three times, we observe a decreasing trend for TA (-16µmol kg$^{-1}$ in three years) (Fig. S3). This decrease is explained by the prevailing of precipitation compared to dissolution. For the bay of Marseille, this result seems consistent as other studies (Bensoussan & Gattuso, 2007 ; Wimart-Rousseau et al., 2020) suggest a net calcifying system, however, in our case, it means that TA is not conserved in the box.

When TA calculation is based on allochthonous formulation, the TA result is the balance between Rhône River TA sources and sinks due to the net calcifying system.

[Figure]

**Figure S3**: Time series of daily averaged TA for the three years of simulation (repetition of 2017 three times) for the reference simulation (SIMC0, Table 2 of the manuscript).

**- What happens to all sinking materials ? Even if they stay in the box, we need to know.**

Sinking materials (POM, organisms larger than CM and $CaCO_3$ in our case) stay in the box. They are constantly recycled by the biogeochemical processes which impact them (see point 3).

**I ask for major revision, just to be sure this part is improved.**

We hope that our responses, especially adding 2.2.1 and the figure 2 to the manuscript, clarify these points.

**For the rest I cannot ask you to re-do everything in 1 or 3D, this will most probably be your next paper anyway.**

Indeed, we plan to study carbonate system variables dynamics in the BoM using a 3D coupled model and to present the results of this study in another manuscript.

Modifications in the text:

**The English might need some rewording. I am not an english native, so I cannot help much for that, but I would recommend a second read. For example, you make an extensive use of the word "yielding". It is a nice word, but you should replace some of them with relevant synonyms.**

As we are not native speakers, we have sent our manuscript to a native speaker before submission. So, the version provided to you was already corrected and rephrased by a native speaker, however, we took into account your suggestion and minimize the use of "yielding".

l.140: We rephrased [As a last term we included the mixotrophic uptake of nutrients which yields the following state equation for TA] to [As a last term we included the mixotrophic uptake of nutrients. **TA is calculated as follows: ]**

l.288: We modified [Both simulations yield a strong decrease of $p$CO$_2$ on March 15th , in response to a Rhône River intrusion in the BoM.] to [**For both simulations a strong decrease of $p$CO$_2$ is modelled on March 15th , in response to a Rhône River intrusion in the BoM.**]

l.301: We deleted [Regarding the coast function, simulations yielded CF < 2 for all variables which is considered very good (CF < 1) or good (1 ≤ CF < 2) (Table 3).].

l.306: We modified [Furthermore, SIMC1 produced the best TA representation yielding the lowest values for CF, %BIAS and RMSD (Table 3).] to [Furthermore, SIMC1 produced the best TA representation **resulting in** the lowest values for CF, %BIAS and RMSD (Table 3).]

l.430: We changed [In contrast, the allochthonous formulation yielded a much high variability in TA that was close to in situ observations.] to [In contrast, the allochthonous formulation **produced** a much high variability in TA that was close to in situ observations.]

l.439: We modified [Having neglected other allochthonous drivers seems to be justified by the results which yielded a close match to observations and a generally better representation of the other carbonate system variables since DIC, $p$CO$_2$ and pH$_T$ are all closely related to TA (Fig. 4 and Table 3).] by [Having neglected other allochthonous drivers seems to be justified by the results which **showed** a close match to observations and a generally better representation of the other carbonate system variables since DIC, $p$CO$_2$ and pH$_T$ are all closely related to TA (Fig. 4 and Table 3).]

l.531: We changed [The reason for this discrepancy may be related to the fact that our model overestimates seawater $p$CO$_2$ during winter, yielding a sea-air difference close to zero (Fig. 5d). As a result, despite strong winds and low temperatures which would favour CO$_2$ absorption (Middelburg, 2019), the winter CO$_2$ sink is not well represented.] to [The reason for this discrepancy may be related to the fact that our model overestimates seawater $p$CO$_2$ during winter, **resulting in** a sea-air difference close to zero (Fig. 5d). As a result, despite strong winds and low temperatures which would favour CO$_2$ absorption (Middelburg, 2019), the winter CO$_2$ sink is not well represented.]

**Page2 line 25 : add some "-" or () : "model – consistent with observations – predicted…."**

Done.

**P3-L65 : You need to add something here about the reason for 0D.**

We added (l.65):

[Most modelling approaches to investigate carbonate system variables typically employ 3D coupled physical-biogeochemical models and focus on larger coastal areas (e.g., Artioli et al., 2014; Bourgeois et al., 2016). If the focus is on smaller areas this requires higher spatial and temporal resolution to correctly represent the relevant processes (Bourgeois et al., 2016). **However, higher spatial and temporal resolution often result in a significant increase of the calculation time which make more difficult the repetition of numerical experiments, an important step to better understanding the global functioning of the area and its reaction to environmental forcings. A solution to avoid important calculation times is to use a dimensionless model. This type of model allows to conduct large amount of test in short amount of time. As instance,** Lajaunie-Salla et al. (2021) used the dimensionless Eco3M-CarbOx model, which contains a carbonate module performing the resolution of the carbonate system based on total alkalinity (TA) and dissolved inorganic carbon (DIC). Even if the DIC, oceanic partial pressure of CO$_2$ ($p$CO$_2$) and total pH (pH$_T$) representations look reliable, Eco3m-CarbOx tends to minimize the range of TA variations during the year, resulting in a near constant TA (Lajaunie-Salla et al., 2021).]

**P5-table1: You only fill the time resolution information for the wind. Does that mean they're all the same ? It seems from the text that some data are daily.  You should feel them all, or tell in the table description why the other data have no time resolution information.**

All data cited in Table 1 are hourly measurements. To avoid confusion, we fill all the lines of the table.

[Table 1. Data types and their sources used to drive the environmental forcing during the 2017 model run (based on Barré et al., 2023a).

| | Data type | Location | Time resolution |
|---|---|---|---|
| **Sea surface temperature** | Measurements | Planier station | **Hourly** |
| **Salinity** | Measurements | Carry buoy | **Hourly** |
| **Wind** | WRF model results | SOLEMIO station | **Hourly** |
| **Irradiance** | WRF model results | SOLEMIO station | **Hourly** |

| **Atmospheric $p$CO$_2$** | Measurements | Cinq Avenues station | **Hourly** |

]

**P6-L124 to 126. " In addition to …". what you say there sounds obvious, but BGC-model not including mixotroph represent reasonable TA and DIC. Do you have a Reference paper for this statement ?**

We added the reference Mitra et al., (2014): In this article, authors show why it is important to consider a mixotrophs compartment and how not consider it can lead to a failure to capture the true dynamics of the carbon fluxes. (Mitra, A., Flynn, K. J., Burkholder, J. M., Berge, T., Calbet, A., Raven, J. A., Granéli, E., Glibert, P. M., Hansen, P. J., Stoecker, D. K., Thingstad, F., Tillmann, U., Våge, S., Wilken, S., and Zukov M. V.: The role of mixotrophic protists in the biological carbon pump, Biogeosciences, 11, 995-1005, https://doi.org/ 10.5194/bg-11-995-2014, 2014.

**P7-equation1 : You should specify that all terms are define in the appendix A.**

Thank you for pointing this out, we added it l.144: [where i represents the number of organisms. **Processes description can be found in Table A1 (Appendix A) and formulations are available in Barré et al. (2023a).** In this formulation, TA only depends on biogeochemical processes (i.e., TA riverine inputs are excluded).] and l.170: [where i represents the number of organisms. **Processes description can be found in Table A1 (Appendix A) and formulations are available in Barré et al. (2023a).** As an additional modification, we use a more recent version of the gas transfer velocity calculation introduced by Wanninkhof (2014).].

**P7- equ2 and 3 : You can specify the unit at the end of the equation, and remove the following sentence.**

Done.

**I might be wrong but, shouldn't the "photo" terms be more like uptake terms ? Phyto absorbs more DIC than the only ones used for the photosynthesis. Isn't this equation missing the remineralization terms as a source of DIC ?**

Thank you for this interesting comment. We are aware that phytoplankton can absorb more DIC than the one associated with photosynthesis process. However, in Eco3M_MIX-CarbOx, we made the choice to consider only the DIC uptake used through photosynthesis (photosynthetic organisms only use the DIC they need for photosynthesis, surplus is released through respiration process).

The term $Photo_{DIC}^{ORG_{C_i}}$ (with ORG $\in$ [PHYC, MIXC]), when apply to the DIC, represents the DIC uptake linked to photosynthesis, by phytoplankton and mixotrophs. When The term $Photo_{DIC}^{ORG_{C_i}}$ is applied to phytoplankton or mixotrophs, it represents the biomass increase associated with photosynthesis. Both processes are identical, they have the same formulation (photosynthesis based on Geider et al. ,1998) except that when the process is applied to DIC, the biogeochemical flux applied is negative (positive when applied to phytoplankton and mixotrophs). We then used the same notation to be consistent.

In Eco3M_MIX-CarbOx, we consider DIC remineralization through respiration process. Especially, the DIC which comes from bacteria respiration is actually the result of POC and DOC remineralization.

**P8-equ5 and L179-181 : I don't understand why you define Aera being negative when the CO$_2$ flux is toward the sea. In Equ 4, ∂DIC/∂T increases with Aera being positive, what means CO$_2$ flux toward the sea, and pCO$_{2,sw}$ > pCO$_{2,atm}$. There's a discrepancy here you might want to correct.**

Thank you for pointing this out. We corrected the balance equation for DIC (l.167, Eq.4):

$$\frac{\partial DIC}{\partial t} = \sum_{i=1}^{2} \left( Resp_{DIC}^{Phy_{C_i}} \right) + \sum_{i=1}^{2} \left( Resp_{DIC}^{Mix_{C_i}} \right) + Resp_{DIC}^{COP_C} + BR_{DIC}^{BAC_C} + Diss_{DIC}^{CaCO_3} - \sum_{i=1}^{2} \left( Photo_{DIC}^{Phy_{C_i}} \right)$$

$$- \sum_{i=1}^{2} \left( Photo_{DIC}^{Mix_{C_i}} \right) - Prec_{DIC}^{CaCO_3} - \mathbf{Aera_{DIC}}$$

Equation 5 of the manuscript remains unchanged. We still consider that negative aeration values ($pCO_2$,atm > $pCO_{2,sw}$) are associated with $CO_2$ fluxes toward the sea and an increase of DIC, which is now consistent with the DIC balance equation.

**P11-L237 : "the first three terms of Eq.(10)", I think you refer to Eq.11, not 10.**

Thank you for pointing this out, we modified: [The first three terms of the Eq. (10) can be calculated as follow:] by [**The three terms of Eq. (11) can be calculated as follow:**].

**P14-Table3 : just to mention, comparing pH is tricky. Comparing pH change or bias in pH unit can be misleading. Best practice is to compare H$^+$ concentration. See Kwiatkowski and Orr, 2018 (https://www.nature.com/articles/s41558-017-0054-0).**

Thank you for pointing this out. We take into account your suggestion and propose to add in Appendix the statistical indicators calculation for H$^+$ concentration:

[**Appendix C: Statistic indicators calculation for H$^+$ concentration**

**Table C1**: Comparing the different model results to surface observations at SOLEMIO station for H$^+$ concentration. N represents the number of observations. Mean, SD, AE, AAE and RMSD are in the same unit than the considered variable, i.e.: mmol m$^{-3}$ for H$^+$ concentrations. % BIAS is without unit.

|  |  | [H$^+$] |
|---|---|---|
| **N** | Observations | 20 |
| **Mean ± SD** | Observations | $8.08 \times 10^{-9} \pm 5.52 \times 10^{-10}$ |
| **Mean ± SD** | SIMC0 | $8.89 \times 10^{-9} \pm 2.91 \times 10^{-10}$ |
|  | SIMC1 | $8.39 \times 10^{-9} \pm 4.06 \times 10^{-10}$ |
|  | CarbOx | $8.52 \ 10^{-9} \ 2.80 \ 10^{-10}$ |
| **%BIAS** | SIMC0 | -5.33 |
|  | SIMC1 | -3.91 |
|  | CarbOx | -5.47 |
| **AE** | SIMC0 | $-4.30 \times 10^{-10}$ |
|  | SIMC1 | $-3.15 \times 10^{-10}$ |
|  | CarbOx | $-4.42 \ 10^{-10}$ |
| **AAE** | SIMC0 | $6.45 \times 10^{-10}$ |
|  | SIMC1 | $6.05 \times 10^{-10}$ |
|  | CarbOx | $6.36 \ 10^{-10}$ |
| **RMSD** | SIMC0 | $6.98 \times 10^{-10}$ |
|  | SIMC1 | $7.14 \times 10^{-10}$ |
|  | CarbOx | $6.93 \ 10^{-10}$ |

]

Please, note that based on Referee 2 suggestions we modified the statistical indicators calculated. We replaced CF which was not sensitive enough by average error (AE) and average absolute error (AAE):

$$AE = \frac{\sum_{i=1}^{N}(O_i - M_i)}{n}$$

(Eq. S4)

$$AAE = \frac{\sum_{i=1}^{N}(|O_i - M_i|)}{n}$$

(Eq. S5)

Where O represents the observations and M the model results, calculations are based on Stow et al. (2009).

And we modified the text accordingly (l.301):

[**For statistical indicators, %BIAS values are systematically lower than 10 %, with the highest values obtained for pCO$_2$ with ~6 % while the remaining variables had values < 1 %. Similarly, pCO$_2$ had the highest RMSD, AAE and AE which suggests that this parameter is not as well represented in the model as the other variables. Furthermore, SIMC1 produced the best TA representation resulting in the lowest values for %BIAS, AE, AAE and RMSD (Table 3). Moreover, SIMC1 produced an annual mean-TA that was closest to the observations. While the SIMC0 and Eco3m-CarbOx results are fairly similar. SIMC0 produced a slightly better representation of TA compared to Eco3m-CarbOx (%BIAS, AE, AAE and RMSD slightly lower). For pH$_T$, SIMC1 outperformed SIMC0 based on %BIAS (Table 3), however, AE, AAE and RMSD values are similar for the three simulations. We then performed the calculation of statistical indicators on H$^+$ concentration as, according to some authors (Kwiatkowski & Orr, 2018), comparing H$^+$ concentrations is a better practice than comparing pH. Results are available in Appendix C. Based on Table C1, SIMC1 also outperformed SIMC0 based on AE and AAE. For studying DIC and pCO$_2$, the situation is less clear as the simulations performed differently for different indicators, making it difficult to pick a clear winner. Still SIMC1 shows the best AAE and RMSD values for DIC, and the best %BIAS, AE, and AAE for pCO$_2$. In conclusion, SIMC1 shows the best overall indicator values for the examined variables (more specifically, it outperformed the other simulations in 13 of 20 indicator comparisons when including H$^+$ concentrations comparison).**]

**P15 -Fig 5 : the e,f,g and h panels are not useful. There is no additional information, and it's not even zoomed-in. Instead, I would remove them, make the picture slightly bigger, and highlight the SUP like you do in Fig. 6 with a shading or something similar.**

Done. We modified the caption and the text accordingly.

[Figure]

**Figure 5**: Time series of (a) in situ daily average sea surface temperature (black line) and salinity (grey line) (b) SIMC1 daily average wind speed (c) the difference between SIMC1 daily average seawater $pCO_2$ and in situ daily average atmospheric $pCO_2$ (d) SIMC1 daily average air-sea $CO_2$ fluxes (aeration process). The summer upwelling period (from 1 May to 1 October) is highlighted in yellow.

**P16-Fig 6 : The panel d is quite difficult to look out, it can be quite difficult to differentiate the different blue lines (especially nTA and nDIC have very similar colours). Plus, most of the time the curves in this panel are between -100 to +100 µatm, while the y-axe goes from -600 to +600 µatm. Apart from the big events, it's quite difficult to see what's happening there. Maybe take the whole page for this picture ?**

We agree. We changed the colour used for nTA representation in figure 6d. If necessary, we also propose to add an appendix which includes the enlargement of figure 6d.

[Figure]

**Figure 6:** Time series for 2017 of daily average (a) in situ temperature and salinity (b) modelled nDIC and nTA (c) modelled seawater and in situ atmospheric $p$CO$_2$ (d) $p$CO$_2$ anomalies generated by DIC, TA, Fw+S and temperature based on the approach in Lovenduski et al. (2007) (Note: the dark blue line is sometimes obscured by the black line, especially in March. **An enlargement of the panel d is available in Appendix X.**) (e, j) $p$CO$_2$ anomalies generated by aeration, solubility, and biological processes based on the approach in Turi et al. (2014). LSE and an upwelling event have been highlighted. The summer upwelling period (SUP) is indicated by yellow shading.

[**Appendix X: Time series of daily average pCO2 anomalies generated by DIC, TA, Fw+S and temperature based on the approach described in Lovenduski et al. (2007), for 2017. Enlargement of the panel d of the figure 6.**

[Figure]

**Figure C1: Time series for 2017 of daily average (a) $p$CO$_2$ anomalies generated by DIC, TA, Fw+S and temperature based on the approach in Lovenduski et al. (2007) (Note: the dark blue line is sometimes obscured by the black line, especially in March), (b) Enlargement of the panel a between -250 and 250 µatm. LSE and an upwelling event have been highlighted. The summer upwelling period (SUP) is**

**indicated by yellow shading.**

**]**

**P17-L365 to 370 : You forgot to refer to Fig. 6e somewhere in this section.**

We added it (l.365-370). [The four LSE are also visible in the solubility-generated anomalies generating strong decreases **(Fig. 6e)**. However, only two LSE are easily identifiable (15 March with a drop from -41 µatm to -163 µatm and 6 May with a drop from 8 µatm to -75 µatm) while the other two appear to be obscured by temperature-related counter-movements. Since aeration- and solubility-generated anomalies show opposite seasonality, they partly cancel each other out. While aeration seems to dominate from November to May, (apart from LSE), solubility appears to dominate from May to November and during LSE. Biological processes are never the dominant driver of $p$CO$_2$ variations as they are systematically smaller (by a factor of 2 to 3) than aeration and solubility-generated anomalies **(Fig. 6e)**. Biology-induced anomalies are always negative, providing evidence that biological processes always decrease $p$CO$_2$.]

**P19-L445 : "we could (not cloud) provide".**

Thank you for pointing this out, we corrected it.

**P20 – L477-8 : "While we only considered TA inputs" (only in the allochthonous formulation, I guess), "Rhône River intrusion can also bring nutrient". This is never explained till now. I already said it in the first part of the review, but you have to be clear about this. The reader cannot fully understand your results otherwise. The model biology only feels the environment changes/variations through the physical forcing only (T, S and light). The biology reacts to the Rhône water only because it is fresher, or to the upwelling because it is colder, but not because of the associated nutrient changes (that do not occur). It is important to tell it because the biology can react in the opposite way than otherwise expected and explain that because it is a 0D model you probably don't have much choice (as I understand it). Knowing that, I am surprised by the DIC variations Fig. 3, that are surprisingly good.**

**P21-L482-4 : same remark than just above.**

We hope that adding 2.2.1 and the figure 2 to the manuscript clarifies this point. However, we also modified this paragraph to better emphasize that only changes in temperature, salinity and AT are considered by the model during these events (l.477 to 482):

[**In all four LSE, biological processes did not have any significant impact on $p$CO$_2$ variations (Fig. 6e). To interpret this result, it is important to consider the assumptions used by Eco3M_MIX-CarbOx (section 2.2). Rhône River intrusion can significantly modify the biogeochemistry of the bay as they are typically associated with temperature and salinity changes and TA, DIC and nutrients inputs (Gatti et al., 2006; Fraysse et al., 2014; Lajaunie-Salla et al., 2021). Due to its 0D configuration, Eco3M_MIX-CarbOx only represents temperature and salinity changes and TA inputs (only if the allochthonous formulation is used for the latter).** Lajaunie-Salla et al. (2021) showed that nutrient inputs associated to Rhône River intrusion in the BoM led to an increase in chlorophyll concentration. This phytoplankton growth leads to further decrease in $p$CO$_2$, which means that by neglecting nutrient inputs we possibly underestimated the importance of biological processes, and especially of autotrophic processes during Rhône River intrusions. Moreover, the high DIC concentrations observed in Rhône River waters (2995 ± 575 µM on average, Sempere et al., 2000) could also affect $p$CO$_2$ variations by increasing the nDIC contribution during intrusion events which counteract the overall of $p$CO$_2$ that is typically observed during these events.]


The switch to a salinity-TA relationship is possible thanks to the fact that in the bay of Marseille, TA variations are mainly the results of rivers contributions, particularly the Rhône River one. It was demonstrated by the lack of variations observed when we modelled TA only based on biogeochemical processes which take place in the box. For DIC, a different reasoning must be adopted, mainly because the processes which impact DIC dynamics are very different than the one which impact TA dynamic. As we consider a surface layer, DIC dynamics is mostly the results of temperature and salinity changes (which are considered by the model) and biogeochemical processes (especially air-sea $CO_2$ exchanges) (Hassoun et al., 2015). The Rhône River can bring DIC to the BoM, these inputs are diluted (far from 2877 µmol $kg^{-1}$, the value observed in the Rhône River, Table S1) and, due to the action of other processes (solubility effects and biogeochemical processes) on DIC dynamics, which is more pronounced in this case, have a less significant impact than on TA dynamics.

Moreover, it is important to note that, in Eco3M_MIX-CarbOx, TA is the main driver of carbonate system. In other words, a change in TA results in significant changes in DIC, $pH_T$ and $pCO_2$ as demonstrated by the Figure 4 of the manuscript. By representing the contribution of the Rhône River on TA we then indirectly apply it to the three other variables of the carbonate system. We are aware that the contribution of the Rhône River to the DIC considered in that way (through TA) is only an indirect effect. It is important to highlight that in the case of our 0D configuration, it is not always possible to consider allochthonous contributions. So far, we experimented two ways to consider external contributions in Eco3M_MIX-CarbOx, which gave satisfactory results:

- By using a VAR=f(S) relation (done in this paper for TA),
- By forcing the variable with a file which include an interpolation of the measurements performed at SOLEMIO for this variable (done in our first paper for nutrient concentrations)

Using a DIC=f(S) relation did not produce the expected results. In fact, it is not recommended to use such a formulation to represent DIC dynamics, especially in surface water due to the numerous processes from which its dynamic results (Hassoun et al., 2015). Using an interpolation of SOLEMIO DIC measurements force us to represent the variable dynamics as a forcing, which had no real interest here given that the representation of DIC obtained was already rather correct when based on biogeochemical processes.

**Table S1**: Salinity-DIC couples for LSE events measured at SOLEMIO between 6 June 2016 and 26 June 2019 (last data available).

| Salinity | DIC ($\mu mol.kg^{-1}$) |
|---|---|
| 37.11 | 2321.3 |
| 37.78 | 2280.1 |
| 37.30 | 2259.3 |
| 36.82 | 2323.8 |
| 37.62 | 2288.7 |
| 37.18 | 2260.3 |
| 37.66 | 2269.8 |
| 37.32 | 2249.0 |

**and nutrients as well?**

[Figure]

**Figure S1:** Time series of surface (a) salinity (CARRY measurements), and interpolated (b) $NO_3^-$ concentration, (c) $NH_4^+$ concentration and (d) $PO_4^{3-}$ concentration at SOLEMIO station. SOLEMIO data are represented by blue markers and the four LSE are indicated by the red dotted lines.

Rhône River intrusion events are associated with an increase of nutrient concentrations in the area, especially nitrate and phosphate (Fraysse et al., 2014). However, in our case, the four low salinity events are not systematically associated with a nutrient increase at the station. In fact, only the first and last events (15 March and 5 September respectively) have an impact on nutrient concentrations at SOLEMIO with the first event being the most significant (Fig. S1). This pattern can be explained by the salinity data used by the model. The measurements are performed at CARRY station (near the Côte Bleue, see figure 1 of the manuscript for location) which is more significantly impacted by the Rhône River plume as it is closer to the river mouth than SOLEMIO station. In consequence, decreases of salinity measured at CARRY are not systematically observed or can be less significant at SOLEMIO. Salinity measurements are also performed at SOLEMIO, however their temporal resolution is low (fortnightly measurements) compared to the CARRY one (hourly measurements). In fact, Rhône River intrusion events duration is variable and can be less than 15 days (ex: short-lived intrusions, Fraysse et al., 2014), therefor it is important to consider the highest temporal resolution possible to better catch them with measurements which is why we chose to work with CARRY measurements instead of SOLEMIO measurements (Fig. S1a).

**That this may lead to biases is discussed in lines 477 to 484; but given the extremely high DIC concentration in Rhone water quoted on line 482, I wonder whether this inconsistency may not invalidate the main results.**

We understand your concerns, however, as indicated in the previous point, the strong DIC value (2877 µmol.kg-1) observed in the Rhône River never reaches SOLEMIO as the Rhône River plume is quickly diluted and the DIC values which really reaches SOLEMIO station are rarely higher than 2300 µmol.kg-1 (mean value of 2281.5 µmol.kg-1, Table S1).

Moreover, as indicated in the previous point, it is important to note that in Eco3M_MIX-CarbOx, TA is the main driver of carbonate system. In other words, a change in TA results in significant changes in DIC, pH and pCO2 as demonstrated by the Figure 4 of the manuscript. By representing the contribution of the Rhône River on TA we then indirectly apply it to the three other variables of the carbonate system.

**3) This leads me to a more conceptual difficulty with the approach. The model concept is that of an arbitrary one cubic metre volume at the surface of the bay, and that the model just represents fluxes within this volume. Spatial fluxes are excluded (except for CO2 flux, more on that below). This only allows either to model a variable as purely forced from what is happening inside the box, or to prescribe it, e.g. as a function of salinity. For a proper modelling of how external fluxes (e.g. in mol/s) change concentrations (mol/m^3/s) inside the modelled region, one would have to define the volume that is affected by these fluxes. A reasonable choice might be to model a column of water within the mixed layer, as was done in many zero-dimensional models, e.g. Fasham et al, 1990 or Hurtt and Armstrong 1999. That would allow a consistent treatment of the effects of mixing on TA, DIC, nutrients.**

**This difficulty becomes especially clear when the authors discuss the possible reasons for their low net annual air-sea flux of CO2, which is in contrast to observation-based estimates. Here they state that "aeration is is simulated by applying Eq (5) to 1 m^3 of surface water at the SOLEMIO station, which tends to overestimate the effect of aeration processes on DIC..." (line 534 ff). Indeed: if the control volume is that shallow, it will be lead to a too fast approach of DIC towards equilibrium, and hence an underestimate of fluxes.**

We understand your concern. We think that it is important to note that, in this study, we relied on Eco3M-CarbOx (Lajaunie-Salla et al., 2021) for the calculation of carbonate variables. So, Eco3M-CarbOx was our starting point to implement carbonate system variables in Eco3M_MIX-CarbOx. With our study, we aim to bring answers to the concerns raised by this previous study, we then use the same concept and try to improve the representation of carbonate cycle variables (by adding mixotrophs organisms processes to the state equations and switching TA formulation by a newly implemented allochthonous formulation). In this way, we were able to compare both models, then knowing how our modifications impact the carbonate system variables representation. Even if we did not manage to obtain a realistic representation of air-sea $CO_2$ fluxes, we provide some improvements to the initial concept and give some examples of suitable and unsuitable use of it (first part and second part of this study respectively), then confirming that the only way to obtain realistic fluxes is to consider a larger layer. We could have done it the way you propose but, considering that it required a complete review of our 0D configuration, we decided to focus directly on the coupling of Eco3M_MIX-CarbOx, in 3D (which is still in test phase).

**4) And finally, while the diagnostic TA leads to an improvement in model results, as evidenced by decreases in %BIAS, RMSD and a 'cost function' presented in Table 3, the improvements are fairly modest. Indeed I would be interested in knowing whether a model with TA prescribed constant at the average of observations would not have fared at least similarly good as the two presented model cases.**

As suggested, we run a simulation with a constant TA (mean of SOLEMIO measurements for 2017 = 2591.2 µmol kg$^{-1}$). In Table S2, we presented the calculation of the statistical indicators presented in the manuscript. As we decided to modify them, based on your suggestions, we also provided average absolute error (AAE), and average error (AE) calculated as in Stow et al. (2009) except that, to be consistent with calculations of statistical indicators used previously (Allen et al., 2007) the difference is applied between observations and model which means that for %BIAS and AE, if a positive value is obtained the model underestimates the observations.

**Table S2**: Statistical indicators calculation for the simulation with a constant TA (TA = 2591.2 µmol kg$^{-1}$). Mean, SD, AE, AAE and RMSD are in the same unit than the considered variable, i.e.: µmol kg$^{-1}$ for TA and DIC and µatm for $p$CO$_2$. CF and %BIAS are without unit.

|  |  | TA | DIC | $p$CO$_2$ | pH$_T$ |
|---|---|---|---|---|---|
| **N** | Observation | 20 | 20 | 20 | 20 |
| **Mean ± SD** | Observation | 2591.2 ± 19.4 | 2294.9 ± 24.0 | 391.0 ± 31.0 | 8.09 ± 0.03 |
| **Mean ± SD** | Model | 2591.2 ± 0.22 | 2305.7 ± 26.1 | 418.0 ± 28.9 | 8.07 ± 0.03 |
| **CF** | Model | 0.85 | 0.82 | 1.14 | 1.14 |
| **%BIAS** | Model | -0.002 | -0.50 | -5.79 | 0.26 |
| **RMSD** | Model | 18.90 | 26.14 | 38.45 | 0.03 |
| **AAE** | Model | 16.5 | 19.7 | 35.5 | 0.03 |
| **AE** | Model | -0.06 | -11.5 | -22.6 | 0.02 |

We also represented the daily mean values of TA, DIC, pH$_T$ and $p$CO$_2$ for the simulations SIMC0, SIMC1 and constant TA (Fig. S2) to compare the three simulations carbonate system variables representation.

[Figure]

**Figure S2**: Comparison of model outputs from SIMC0 (autochthonous formulation, Table 2 of the manuscript), SIMC1 (allochthonous formulation, Table 2 of the manuscript), and constant TA simulation, model runs showing daily average (a) TA, (b) DIC, (c) $pCO_2$ and, (d) $pH_T$ for 2017. SOLEMIO data are represented by blue markers.

We are aware that improvements seem fairly modest, especially when studying the statistical indicators (Table S2). However, values are generally slightly better for the simulation SIMC1, especially for the three other carbonate system variables. In fact, the major improvement bring by the switch to an allochthonous formulation for TA, is that TA variations are represented which seems more realistic and tend to improve the representation of the other three carbonate system variables.

Major comments :

**Figure 1 is identical to the one on the companion paper and definitively isn't needed should this paper be published in GMD.**

Done. We replaced the map by the following sentence: **[A map of the study area showing the location of stations where measurements were carried, and places of interest can be found in Barré et al. (2023a).]** and added at the end of the 2.1.

**State equation for TA, Eq. (1): The terms in the equation are not properly defined. The definition of the terms is given in the Appendix (Table A1), but the table is not referenced here.**

Thank you for pointing this out, we added it l.144: [where i represents the number of organisms. **Processes description can be found in Table A1 (Appendix A) and formulations are available in Barré et al. (2023a).** In this formulation, TA only depends on biogeochemical processes (i.e., TA riverine inputs are excluded).] and l.170: [where i represents the number of organisms. **Processes description can be found in Table A1 (Appendix A) and formulations are available in Barré et al. (2023a).** As an additional modification, we use a more recent version of the gas transfer velocity calculation introduced by Wanninkhof (2014).].

**The two linear S-TA relations, presented on page 7, which are valid below and above a salinity threshold of 37.8 are discontinuous at S=37.8. This should lead to sudden jumps in the TA value if this threshold is crossed. Are there any effects of this discontinuity visible in the results?**

Thank you for this interesting comment. Indeed, using the allochthonous formulation leads to sudden jump in TA when the thresholds is crossed. These sudden jumps are also observed in $pH_T$ and $pCO_2$.

When we implemented the TA allochthonous formulation, we used two points: a first one which represents the Rhône River water at the river mouth (S = 0, TA = 2885 µmol $kg^{-1}$) and a second one which represents the Rhône River water which reach SOLEMIO during a LSE (S = 36.82, TA = 2600.6 µmol $kg^{-1}$). We chose the second point as it was the most significative LSE on the period covered by the SOLEMIO measurements (2017 and 26 June 2019). We then consider it as representative of the Rhône River water which reach the BoM. Even though TA values associated with LSE are variable and highly depend on the period of the year (Fig. S3). TA values equal or above 2600 µmol $kg^{-1}$ do not seem the most representative (Table S3) and LSE seem associated with TA highly dependent on Rhône River seasonality (mean value = 2575 µmol $kg^{-1}$).

[Figure]

**Figure S3**: TA measurements in the Rhône River (data: Naïades, https://naiades.eaufrance.fr, first data available: January 2018).

**Table S3:** Salinity-TA couples for LSE events measured at SOLEMIO between 6 June 2016 and 26 June 2019 (last data available).

| Salinity | TA (µmol kg-1) |
|----------|----------------|
| 37.11 | 2603.0 |
| 37.78 | 2579.6 |
| 37.30 | 2585.5 |
| 36.82 | 2600.6 |
| 37.62 | 2585.8 |
| 37.18 | 2560.8 |
| 37.66 | 2568.4 |
| 37.32 | 2520.7 |

We think that it might be interesting to improve our allochthonous formulation to better manage the threshold crossing case. To avoid (or at least reduce these instabilities) it could be interesting to take into account Rhône River seasonality in the allochthonous formulation and we plan to do this in a future work.

**Page 8, line 175: In principle the model equations would not change had you chosen to assume the effected layer to be deeper, except that then the flux would then be distributed over a larger volume. Why not take at least H as the annual average mixed layer depth> Taking it as 1m is equivalent to speeding up the gas exchange by a factor H_real, the real affected layer.**

As suggested, we ran a simulation with a modified function for aeration process (Eq. S1, SIMR1 in the following). We considered a mean annual value of 30.5 m for mixed layer depth (mean of winter value

= 41 m and summer value = 20 m (Wimart-Rousseau et al., 2020)). Daily average of modelled carbonate system variables and air-sea CO₂ fluxes are represented in figure S4.

[Figure]

**Figure S4:** Comparison of model outputs from SIMC1 (aeration process apply on a 1 m layer, Table 2 of the manuscript) and SIMR1 (aeration process apply on 30.5 m layer, model runs showing daily average (a) TA, (b) DIC, (c) $pCO_2$, (d) $pH_T$, and Air-sea CO₂ fluxes for 2017. SOLEMIO data are represented by blue markers.

We obtained a mean annual value for air-sea CO2 fluxes of -113.6 mmol m$^{-2}$ yr$^{-1}$ which is better than the one obtained previously (-0.21 mmol m$^{-2}$ yr$^{-1}$) but still lower than the value suggested by Wimart-Rousseau et al. (2020) (-803 mmol m$^{-2}$ yr$^{-1}$). These results are interesting as we can see that, by considering a larger layer, we better represent the seasonality of air-sea CO₂ fluxes (sink in winter and source in summer) (Fig. 4e) which is mainly explained by the fact that we are able to represent the undersaturation observed for $pCO_2$ in winter, especially at the end of the year (Fig. S4c). In this simulation, we used a constant MLD, however, we believe that using a variable MLD (deeper in winter than in summer) could emphasize this result. These results also show that by considering a larger layer to apply the aeration fluxes, we significantly modify DIC representation (Fig. S4b). Thus, the seasonality well modelled previously, is no longer visible. DIC values are also much less variable, then far from the dynamics described by observations.

To conclude, we believe that more than modifying the thickness of the layer impacted by aeration process, we need to move to a 3D configuration to better represent air-sea CO₂ fluxes without impacting the representation of DIC which is already, rather correct.

**Page 8 and Appendix B, pH and pCO2 calculation: It is good to see that the pH scale differences are taken properly into account, and fugacity has been calculated correctly. But much of this is fairly standard, e.g. the iterative calculation of pH, described in Figure B1. This could be left away.**

We considered your comment, however, as we provide some corrections to Lajaunie-Salla et al. (2021) Appendix A, we think that it is necessary to keep Appendix B in the manuscript.

**Page 9, Figure 3: The quality of the Figure is awful. But also it does not convey much information, I would leave it away.**

We considered your suggestion. However, we think that figure 3 allows to better visualize the model calculation steps and how these can be modified by the choice of TA formulation. We decided to modified figure 3 to increase its quality:

[Figure]

**Figure 3**: Flow diagram illustrating the steps needed to calculate pHT and $pCO_2$ (a) using the autochthonous formulation (Eq. 1) and (b) with the allochthonous formulation (Eq. 2 and 3). Physical forcings include temperature (T), salinity (S), solar irradiance (IRR), wind speed (Wind) and atmospheric $pCO_2$ ($pCO_{2,ATM}$).

We hope it is clearer this way.

**Page 10, lines 218-220: It is not clear to me how the salinity-normalized nTA and nDIC are exactly defined, by a linear correlation with salinity with zweo intercept? If so, why do that if the observed S-TA relation in the oceanographic region is different?**

Salinity-normalised changes in nTA and nDIC were calculated by dividing by in situ salinity and multiplying by mean salinity.

To clarify, we added (l.219): [Though, we isolate the changes of TA and DIC due to variations in freshwater inputs using the salinity normalised TA (nTA) and DIC (nDIC) **which are obtained by dividing the considered variable by in situ salinity and by multiplying the result by mean in situ salinity,** and adding another term to regroup them.]

**page 11, definition of the statistical indicators: while the definition of RSMD and %BIAS is rather clear, that of the cost function is less clear: Typically, a cost function aggregates model-data-disaggreement for different variables, possibly with different units, into a single scalar variable (Stow et al, 2009). But what exactly the variables are that enter the CF, and how the different variables are nondimenionalized and aggregated into one CF should be properly defined.**

We based our CF calculation on Allen et al. (2007). In this work, CF is defined as: [The cost function gives a non-dimensional value which is indicative of the "goodness of fit" between two sets of data; it quantifies the difference between model results and measurement data (see OSPAR Commission, 1998). It is a measure of ratio of the model data misfit to a measure of the variance of the data; the closer the value is to zero the better the model.], and is calculated as follow:

$$CF = \frac{1}{N} \sum_{i=1}^{N} \left( \frac{|O_i - M_i|}{\sigma_O} \right)$$

Where O represents the observations, M the model results and σO is the standard deviation of the observations.

**page 11, interpretation of statistical indicators: Whether a CF<1 is considered very good, would probably depend on the definition of CF, and cannot be stated as generally as on line 255-256. If the individual cost function terms e.g. consist of the squared model-data difference scaled by the variance in the individual variables, and are then added together, the expected height of the CF would depend on how many different variables are finally added together. Also, I don't think one can generally say (line 252-253) that a %BIAS<10% is excellent; I would think that depends on the ratio of natural variability to the mean of the variable in question. For TA, with a high background value, a 10% BIAS is rather large.**

To interpret CF and %BIAS, we used the interpretation of Radach & Moll (2006) and Marechal (2004) respectively. We understand that these interpretations seem not strict enough. Moreover, CF indicator seems rather insensitive as almost all variables for the tested simulations show CF value lower than 1. To improve our statistical analysis of model results, we proposed to consider two other indicators: AE and AAE, to replace CF. We based their calculations on Stow et al. (2009) except that, to be consistent with the calculation of RMSD used previously (Allen et al., 2007), the difference is applied between observations and model which means that for AE, if a positive value is obtained the model underestimates the observations. Moreover, we added the formulation of each statistical indicators used to avoid confusion.

We modified the 2.4 accordingly:

[revised manuscript text omitted]

**Table 3, Page 14: If the variance of the observed TA and DIC values is on the order of 20 micromol/kg (note, units should be given in the table), then I'd say a RMSD of about the same order of magnitude is not an excellent agreement. It is not terrible either, though. A similar remark holds for %BIAS.**

We hope that modifications mentioned above allow to clarify this point. We indicated units in the caption of Table 3.

**Figure 4, page 12: The time-series of the difference between the model runs (right panel) does not convey much new information, I would remove them.**

We represented the differences between simulations as we think that it allows to better visualize them and then emphasize the fact that modifying TA formulation yields different model outputs for DIC, $p$CO$_2$ and pH$_T$. Considering that, we decided to keep the figure 4 as is.

**Also, I have a question to the data (crosses in Figure 4): to me it is not clear whether all four carbon system variables were measured independently, or whether e.g. DIC and TA were measured, and pH and pCO2 calculated from them. If they were measured independently, how consistent are they with respect top each other, given the used set of carbon system equations?**

DIC and TA are measured, and we calculate pH$_T$ and $p$CO$_2$ by using CO2SYSv3 (Sharp et al., 2020, originally developed by Lewis and Wallas (1998)) on MATLAB. The set of constants used is the same than the one used to perform the calculation of pH$_T$ and $p$CO$_2$ in the model.

**Figure 5, page 15: The subpanels on the right are simply a cutout of the panels on the left for the summer period. What is the purpose of this duplicated information?**

[Figure]

**Figure 5: Time series of (a) in situ daily average sea surface temperature (black line) and salinity (grey line) (b) SIMC1 daily average wind speed (c) the difference between SIMC1 daily average seawater**

*p*CO₂ and in situ daily average atmospheric *p*CO₂ (d) SIMC1 daily average air-sea CO₂ fluxes (aeration process). The summer upwelling period (from 1 May to 1 October) is highlighted in yellow.

We wanted to highlight the SUP, we agree that these panels do not bring more information and that left panel are pretty clear so we modified Figure 5 by deleting the right panels and highlighted the SUP in yellow on left panels.

**Page 21, Lines 506 ff: Would including the DIC and nutrient input from upwelling improve the model-data agreement, or the converse?**

As indicated in point 2), in our configuration, we can consider these inputs in two ways: by using a relation with salinity or by using an interpolation of SOLEMIO measurements which is then read by the model. For DIC, both methods do not give satisfactory results. For nutrients, we used the second methods in Barré et al. (2023a). In both cases, it is difficult to directly test (through a simulation which take them into account) their effect on model-data agreement. However, to answer your question, we propose the followings hypotheses based on DIC and nutrients measurements study. We represent a linear interpolation of SOLEMIO measurements for these variables on figure S5:

[Figure]

**Figure S5:** Time series of surface (a) temperature (PLANIER measurements), and interpolated (b) DIC, (c) $NO_3^-$ concentration, (d) $NH_4^+$ concentration and (e) $PO_4^{3-}$ concentration at SOLEMIO station. SOLEMIO data are represented by blue markers and the SUP is shaded in yellow.

For DIC, during the SUP, measurements show values around 2283 µmol kg⁻¹ (mean of DIC measurements during the SUP). All upwellings of the period do not necessarily impact DIC. Only two events are noticeable: at the beginning of July and mid-September. These two events do not seem to be correlated with LSE, or Cortiou water inputs (generally associated with high $NH_4^+$ concentrations). The first event is not reproduced by the model, we then assume that this DIC increase could be associated with an upwelling event. However, the second one is well reproduced by the model which means that it is not resulting from an upwelling input (as, for now, we do not take these inputs into account in the model) (Fig. 4 of the manuscript). Considering these results, we believe that adding DIC inputs from upwelling could improve the realism of our representation, and consequently the data-model agreement.

For nutrients, it clearly appears that during the SUP, their dynamics are only slightly affected by upwelling events as nutrients concentrations remain close to 0 for most of the time. Only two nutrient inputs are noticeable during the SUP: in July and September. However, these events do not correspond to upwellings as the first one is associated with Cortiou water which reaches SOLEMIO (high $NH_4^+$ concentration) and the second one is, as showed in the manuscript associated with a Rhône River intrusion. This low impact can be explained by the fact that, when the upwelling takes place, nutrients which are upwelled are quickly consumed by the phytoplankton present in the area, then not reaching the station. Considering these results, we suppose that taking into account nutrients inputs associated with upwelling events could improve the model data agreement as it might bring some more realism to our representation, but not enough to consider them here, as their impact at the station is quite limited.

To conclude, we think that considering upwelling inputs could be a great addition to improve the realism of our representation and then the model-data agreement, especially for DIC, however, considering that, in the present configuration, these contributions can hardly be taken into account, we believe that switching to a 3D configuration will be the most appropriate way to confirm this.

**Page 22, Line 530ff: Can one give a conjecture why the model overestimates pCO2 during winter?**

We already explained why the model overestimates $p$CO$_2$ during winter l.535: [Seawater $p$CO$_2$, air-sea CO$_2$ fluxes and DIC are closely connected (Appendix B, Fig. 3). In Eco3M_MIX-CarbOx, aeration is simulated by applying Eq. (5) to 1 m$^3$ of surface water at SOLEMIO station which tends to overestimate the impact of aeration process on DIC and, due to the close link between DIC and $p$CO$_2$, also on $p$CO$_2$].

As seawater $p$CO$_2$ calculation is closely linked to air-sea CO$_2$ fluxes and DIC, we can assume that, when we tend to overestimate the impact of aeration process on DIC, we then impact $p$CO$_2$ and also overestimate it in winter.

Technical comments:

Thank you for this, we take them into account.

---

## Referee Report (RR1)

Review of Barré et al. 2023B-authors response : Implementation and assessment of a model including mixotrophs and carbonate cycle (Eco3M_MIX_CarbOx v1.0) – Part 2. –  Julien Palmiéri.

In my review of Barré et al's study, I've made quite a lot of comments, not all easy to assess, and I would like to thank Lucille Barré and her co-authors for the efforts they've demonstrate in their answer. I must have to highlight that the paper is now much clearer. The addition of the new figure 1 and the additional explanations in the text help the reader to understand what is really happening in the model, and to appreciate what is calculated from what is imposed to the model.
All my questions and corrections have been successfully addressed, and I am then happy with the revised version of the paper. I didn't realize it takes 45 minutes to run 3y in 0D, it's quite something! That does help explaining the need of the 0D. Still, to me, this choice will stay a weakness in the paper. But we won't change that now, and we'll see the real potential of the model in the bay of Marseille in your next paper (where ECO3M shouldn't need the allochthonous formulation to get the alkalinity right).

There are few typo mistakes I've spotted, but they are minor and will be easily fixed (based on the track-changes author response pdf) :
L145 : "...the only exchanges allow**ed** between…"
L150 :  "contrary to C"  I would change for something like "unlike what is done for the C pool". At your convenience, but contrary to C doesn't sound right.
L152 : "In the following, …" I would say "In the following section,…"
L753-4 : "… would have been more complex to conduct in 3D (i.e., longer simulations and isolation of pCO2 variation drivers' contributions more difficult as the model is more complex)." Could be improved without the repetition of "complex". Maybe something like " … would have been a tedious task to realize in 3D (i.e., longer simulations and isolation of pCO2 variation drivers' contributions more difficult due to the complexity of the model)." ?

Last question. While filling the journal review page. I am asked about the reproducibility of the experiment, if someone else would like to test your code. Although you explain all your work in great details, I realize I wouldn't know where to get your code, apart from asking for it by mail… Is there I link to a git or svn repository ? As I think that for GMD the model should be freely accessible. If there is such a thing for ECO3M, and for your version of the code, could you add them in the paper ?

Again, thanks for your effort!

Best regards,

Julien Palmiéri.

---

## Author Response (AR4)

**Authors' response [07/02/2024]**

**REFEREE 1:**

First, we would like to thank Julien Palmiéri for his careful evaluation of our manuscript and his interesting comments which we believe will help us to improve it. Please, find hereafter our response to these comments.

*The changes made in the manuscript are indicated in* **blue** *and the changes made in the supplementary material are indicated in* **green**.

Major comments

**1) First, the reason of the choice is not given. I understand it is greatly needed for developments like adding the mixotrophs in Eco3M. It's a huge task that require such lightweight configuration to test, verify that nothing is broken, and make sure the fluxes between each element of the model are reasonable. So, this choice is easily explained for the part 1, but less for this one.**

We agree with this point. The use of a 0D configuration may seem more justified for the first part of our study as adding mixotrophs required many tests.

We chose to also use a dimensionless configuration for the part II as we wanted to provide a reliable representation of the carbonate system, which consider mixotrophs organisms, in the simplest way possible. In other words, we wanted to provide a tool which is easy to use, easy to adapt to other coastal area (by modifying the environmental forcings and the AT-S correlation) and give reliable results in a short amount of time.

Another reason to use a dimensionless configuration rather than a 3D configuration was the possibility to compare Eco3M_MIX-CarbOx to Eco3M-CarbOx in the same type of implementation. By comparing the models, we can answer several questions and especially: How mixotrophs affect carbonate variables ? Does adding them provide a more accurate representation of these variables ? and then build further relevant simulation strategies for 3D.

Considering these reasons, this dimensionless configuration can be seen as a laboratory to test and then build further relevant simulation strategies for 3D.

To justify the use of a 0D configuration, we first clarified the aim of the study at the end of the introduction (l.73):

[Here we try to provide a more realistic representation of carbonate system variables in the BoM. As a starting point, we used the concept of the dimensionless Eco3M-CarbOx model (Lajaunie-Salla et al., 2021), which aims to represent a small volume of surface water (i.e., 1 m$^3$ ) in the BoM. We developed a planktonic ecosystem model which contains, among others, mixotrophic organisms, modified the carbonate module described by Lajaunie-Salla et al. (2021) and added it to our newly developed planktonic ecosystem model to obtain the Eco3M_MIX-CarbOx model (v1.0). We implemented two types of TA formulation and compared the simulation results to in situ observations to identify which formulation was capable to deliver the more realistic results: (i) a formulation that only considers biological processes (referred to as autochthonous formulation) and (ii) a new TA formulation that depends only on salinity (referred to as allochthonous formulation). Furthermore, we simulate air-sea CO$_2$ fluxes to determine whether the BoM act as a sink or a source of CO$_2$ and provide a detailed analysis of drivers of seawater $p$CO$_2$ variations for two specific hydrodynamic processes typical for the BoM: (i) Rhône River intrusion and (ii) summer upwelling events. **With this study, we aim to provide a new tool**

**which allow to obtain a reliable representation of the carbonate system in the simplest way as possible: by using a dimensionless configuration which is easy to use, adapt and give results in a short amount of time.]**

And we gave a justification of this choice in the Section 2.2 (l.119):

**[**In this study, we used the Eco3M_MIX-CarbOx model (v1.0) which was developed to represent the dynamics of the seawater carbonate system and mixotrophs in the BoM and was implemented using the Eco3M (Ecological Mechanistic and Molecular Modelling) platform (Baklouti et al., 2006a, b). **Eco3M_MIX-CarbOx is a dimensionless model (0D): we consider a volume of 1 m$^3$ of surface water at SOLEMIO station, in this volume the state variables only vary over time as the model is not coupled with a hydrodynamic model. We chose to use a 0D configuration as this configuration has several advantages, especially, calculation times are low (around 45 minutes in our case). It allows us to make several test simulations to better understand the biogeochemical functioning of the BoM and its possible reactions to environmental forcings.]**

**In the introduction, it's made mention of the need for high resolution model for coastal regional study, and the next sentence announce the use of a dimensionless (0D hereafter) configuration.**

We understand that these sentences can be confusing. We add a sentence to make a better link between 3D introduction and use of 0D (l.62).

**[**Most modelling approaches to investigate carbonate system variables typically employ 3D coupled physical-biogeochemical models and focus on larger coastal areas (e.g., Artioli et al., 2014; Bourgeois et al., 2016). If the focus is on smaller areas this requires higher spatial and temporal resolution to correctly represent the relevant processes (Bourgeois et al., 2016). **However, higher spatial and temporal resolution often result in a significant increase of the calculation time which make more difficult the repetition of numerical experiments, an important step to better understanding the global functioning of the area and its reaction to environmental forcings. A solution to avoid important calculation times is to use a dimensionless model. This type of model allows to conduct large amount of test in short amount of time. For instance,** Lajaunie-Salla et al. (2021) used the dimensionless Eco3M-CarbOx model, which contains a carbonate module performing the resolution of the carbonate system based on total alkalinity (TA) and dissolved inorganic carbon (DIC). Even if the DIC, oceanic partial pressure of $CO_2$ ($pCO_2$) and total pH ($pH_T$) representations look reliable, Eco3m-CarbOx tends to minimize the range of TA variations during the year, resulting in a near constant TA (Lajaunie-Salla et al., 2021).**]**

**2) A second point is that this 0D brings more questions than answers. Because it is a surface box, the model does not represent advection and mixing. The physics variable/forcing come from observations and hence include annual cycle and external forcing, including specific phenomena like summer upwellings or the Rhône waters passing by. But what about the nutrients ?**

We agree that hydrodynamic processes, especially upwelling and Rhône River intrusions in the BoM are associated with nutrients inputs. In Eco3M_MIX-CarbOx, upwellings are only represented by strong variations of temperature and Rhône River intrusions are only represented by strong variations of salinity and TA inputs when allochthonous TA formulation is used. In other words, we do not consider the possible inputs of nutrients associated with these events and assumed that nutrients are fully the result of autochthonous biological processes (due to 0D configuration) which means that they are modelled based on the following state equations :

$$\frac{\partial NO_3}{\partial t} = Nitrif_{NO_3}^{NH_4} - \sum_{i=1}^{2} Upt_{NO_3}^{Phy_{N_i}} - Upt_{NO_3}^{CM_{N_i}}$$

$$\frac{\partial NH_4}{\partial t} = Excr_{NH_4}^{COP_N} + Excr_{NH_4}^{NCM_N} + Remin_{NH_4}^{BAC_N} - \sum_{i=1}^{2} \left(Upt_{NH_4}^{Phy_{N_i}}\right) - Upt_{NH_4}^{CM_N} - Upt_{NH_4}^{BAC_N} - Nitrif_{NH_4}^{NO_3}$$

$$\frac{\partial PO_4}{\partial t} = Excr_{PO_4}^{COP_P} + Excr_{PO_4}^{NCM_P} + Remin_{PO_4}^{BAC_P} - \sum_{i=1}^{2} \left(Upt_{PO_4}^{PHY_{P_i}}\right) - Upt_{PO_4}^{CM_P} - Upt_{PO_4}^{BAC_P}$$

(Eq. I)

Nitrate concentration results from nitrification and phytoplankton and CM uptakes. Ammonium concentration results from copepods and NCM excretion, bacterial remineralisation, heterotrophic bacteria, phytoplankton, and CM uptakes and losses from nitrification. Finally, phosphate concentration results from copepods and NCM excretion, bacterial remineralisation and heterotrophic bacteria, phytoplankton, and CM uptakes.

**What are the external forces driving the biology of the model ?**

Biology is impacted by temperature and/or irradiance. Depending on the biogeochemical process considered, both or only one of them can have an impact and this impact could be direct or indirect. Nitrification which is performed by nitrifying bacteria (organisms not considered explicitly in the model) depends on temperature and modelled dissolved oxygen concentration:

$$Nitrif_{NH_4}^{NO_3} = tx_{NITRIF} * NH_4 * f_{Q_{10},nitrif}^{T} * \frac{O_2}{O_2 + K_{O_2}}$$

$$f_{Q_{10},nitrif}^{T} = Q_{10,nitrif}^{\frac{T-10}{10}}$$

(Eq. II)

where $tx_{NITRIF}$ represents the fraction of $NH_4^+$ used for nitrification, $K_{O2}$ is the dissolved oxygen half saturation constant and $Q_{10,nitrif}$ is the temperature coefficient for nitrification.

We detailed the modelling of planktonic organisms in the companion paper (Barré et al., 2023a). To sum up, heterotrophic bacteria processes are directly impacted by temperature. Phytoplankton and CM processes are directly impacted by temperature (all processes) and irradiance (photosynthesis and grazing). Copepods and NCM processes are indirectly impacted by temperature and irradiance through the consumed preys.

**This is not explained until the discussion, what is extremely frustrating, as we don't really understand what the model sees and feels or not, until the very end, when the author reveals some of the experiment limitations.**

We provided all the balance equations, detailed formulations of biogeochemical processes and parameters values in the companion paper (Barré et al., 2023a). For organisms, we did not find relevant to provide them again. However, we understand the necessity of adding explanations about the nutrients in the manuscript. We decided to add a subsection in Section 2.2, in which we detail the modelling of nutrients and organic matter.

**[2.2.1 Nutrients and organic matter representation in the model**

**As we use a dimensionless configuration, we assume that nutrients are fully the result of autochthonous biological processes. In other terms, we do not consider allochthonous inputs of**

nutrients (i.e., from rivers or atmosphere as instance). For all the simulations, nutrients dynamics are represented by the following state equations:

$$\frac{\partial NO_3}{\partial t} = Nitrif_{NO_3}^{NH_4} - \sum_{i=1}^{2} Upt_{NO_3}^{Phy_{N_i}} - Upt_{NO_3}^{CM_{N_i}}$$

$$\frac{\partial NH_4}{\partial t} = Excr_{NH_4}^{COP_N} + Excr_{NH_4}^{NCM_N} + Remin_{NH_4}^{BAC_N} - \sum_{i=1}^{2} \left(Upt_{NH_4}^{Phy_{N_i}}\right) - Upt_{NH_4}^{CM_N} - Upt_{NH_4}^{BAC_N} - Nitrif_{NH_4}^{NO_3}$$

$$\frac{\partial PO_4}{\partial t} = Excr_{PO_4}^{COP_P} + Excr_{PO_4}^{NCM_P} + Remin_{PO_4}^{BAC_P} - \sum_{i=1}^{2} \left(Upt_{PO_4}^{PHY_{P_i}}\right) - Upt_{PO_4}^{CM_P} - Upt_{PO_4}^{BAC_P}$$

(1)

The concentration of NO$_3^-$ results from nitrification and phytoplankton and CM uptakes. Ammonium concentration results from copepods and NCM excretion, bacterial remineralisation, heterotrophic bacteria, phytoplankton, and CM uptakes and losses from nitrification. Phosphate concentration results from copepods and NCM excretion, bacterial remineralisation and heterotrophic bacteria, phytoplankton, and CM uptakes.

Such as nutrients dynamics, organic matter (dissolved and particulate) dynamic is only the result of autochthonous biological processes (Eq. 2 and 3).

$$\frac{\partial DOC}{\partial t} = \sum_{i=1}^{2} \left(Exu_{DOC}^{PHY_{C_i}}\right) + \sum_{i=1}^{2} \left(Exu_{DOC}^{MIX_{C_i}}\right) + Excr_{DOC}^{COP_C} + Mort_{DOC}^{BAC_C} - BP_{DOC}^{BAC_C}$$

$$\frac{\partial DON}{\partial t} = \sum_{i=1}^{2} \left(Exu_{DON}^{PHY_{N_i}}\right) + \sum_{i=1}^{2} \left(Exu_{DON}^{MIX_{N_i}}\right) + Mort_{DON}^{BAC_N} - Upt_{DON}^{CM_N} - Upt_{DON}^{PICO_N} - Upt_{DON}^{BAC_N}$$

$$\frac{\partial DOP}{\partial t} = \sum_{i=1}^{2} \left(Exu_{DOP}^{PHY_{P_i}}\right) + \sum_{i=1}^{2} \left(Exu_{DOP}^{MIX_{P_i}}\right) + Mort_{DOP}^{BAC_P} - Upt_{DOP}^{CM_P} - Upt_{DOP}^{PICO_P} - Upt_{DOP}^{BAC_P}$$

(2)

The concentration of dissolved organic carbon (DOC), nitrogen (DON) and phosphorus (DOP) depends on phytoplankton and mixotrophs exudation, copepods excretion, heterotrophic bacteria mortality (natural mortality) and CM, PICO and heterotrophic bacteria uptake.

$$\frac{\partial POC}{\partial t} = E_{POC}^{COP_C} + Predation_{POC}^{COP_C} - BP_{POC}^{BAC_C}$$

$$\frac{\partial PON}{\partial t} = E_{PON}^{COP_N} + Predation_{PON}^{COP_N} - Upt_{PON}^{BAC_N}$$

$$\frac{\partial POP}{\partial t} = E_{POP}^{COP_P} + Predation_{POP}^{COP_P} - Upt_{POP}^{BAC_P}$$

(3)

The concentration of particulate organic carbon (POC), nitrogen (PON) and phosphorus (POP) depends on copepods egestion, predation by higher trophic levels on copepods (closure term of the model) and heterotrophic bacteria production and uptake. POM particles are large enough to sink, however, we do not consider a term to represent their removal from the surface box by sinking. In our case, the POM, such as the DOM, stay in the box and is constantly recycling.

**A detailed description and formulations of processes can be found in Barré et al. (2023a). Processes notation description can be found in Table A1 (Appendix A).]**

We also modified Appendix A to add the definition of processes introduced by the equations of this new subsection:

[Table A1: Description of state equation processes.

| Notation | Process |
|---|---|
| **Copepods** | |
| $Excr_{NutX}^{COP_X}$
 $NutX \in [NH_4^+, PO_4^{3-}]$
 $X \in [N, P]$ | Excretion of nutrient X by copepods |
| $Excr_{DOC}^{COP_C}$ | DOC excretion by copepods |
| $Resp_{DIC}^{COP_C}$ | Copepods respiration |
| $E_{POX}^{COP_X}$
 $X \in [C, N, P]$ | Copepods egestion |
| $Predation_{POX}^{COP_X}$
 $X \in [C, N, P]$ | Predation by higher trophic levels on copepods |
| **Mixotrophs (Mix $\in$ [NCM, CM])** | |
| $Exu_{DOX}^{Mix_{X_i}}$
 $X \in [C, N, P]$ | DOX exudation by mixotrophs |
| $Resp_{DIC}^{Mix_C}$ | Mixotrophs respiration |
| $Photo_{DIC}^{Mix_C}$ | Mixotrophs photosynthesis |
| $Excr_{NutX}^{NCM_X}$
 $NutX \in [NH_4^+, PO_4^{3-}]$
 $X \in [N, P]$ | Excretion of nutrient X by NCM |
| $Upt_{NutX}^{CM_X}$
 $X \in [N, P]$
 $NutX \in [NO_3^-, NH_4^+, PO_4^{3-}]$ | Uptake of nutrient X by constitutive mixotrophs |
| $Upt_{DOX}^{CM_X}$
 $X \in [N, P]$ | Uptake of DOX by constitutive mixotrophs |
| **Phytoplankton (Phy $\in$ [NMPHYTO, PICO])** | |
| $Resp_{DIC}^{Phy_C}$ | Phytoplankton respiration |
| $Photo_{DIC}^{Phy_C}$ | Phytoplankton photosynthesis |
| $Upt_{NutX}^{Phy_X}$
 $NutX \in [NO_3^-, NH_4^+, PO_4^{3-}]$ | Uptake of nutrient X by phytoplankton |
| $Exu_{DOX}^{Phy_X}$
 $X \in [C, N, P]$ | DOX exudation by phytoplankton |
| $Upt_{DOX}^{PICO_X}$
 $X \in [N, P]$ | Uptake of DOX by picophytoplankton |
| **Heterotrophic bacteria** | |
| $BP_X^{BAC_C}$
 $X \in [DOC, POC]$ | Bacterial production |
| $BR_{DIC}^{BAC_C}$ | Bacterial respiration |
| $Upt_{POX}^{BAC_X}$ | POX uptake by heterotrophic bacteria |

]

**3) Still about the 0D, what happens to the POM ? Do they sink ? Are they removed from the surface box ? Or do they float there and are slowly remineralized (as if the bay is mixed enough to keep the particles around) ? This is important as it has an impact on TA and DIC and all other nutrients concentration.**

In the model, the POC, PON and POP dynamics result from copepods egestion, higher trophic level predation on copepods (closure terms of the model) and heterotrophic bacteria uptake (Eq. III). They do not sink and are constantly recycling in the surface box using the processes indicated above.

$$\frac{\partial POC}{\partial t} = E_{POC}^{COP_C} + Predation_{POX}^{COP_X} - BP_{POC}^{BAC_C}$$

$$\frac{\partial PON}{\partial t} = E_{PON}^{COP_N} + Predation_{PON}^{COP_N} - Upt_{PON}^{BAC_N}$$

$$\frac{\partial POP}{\partial t} = E_{POP}^{COP_P} + Predation_{POP}^{COP_P} - Upt_{POP}^{BAC_P}$$

(Eq. III)

Using these state equations, the POM compartment is balanced (Fig. I) which is why we do not consider a term to represent their removal from the surface box by sinking.

[Figure]

**Figure I**. Time-series of daily averaged (a) POC, (b) PON and (c) POP for the three years of simulation (repetition of 2017 three times) for the reference simulation (SIMC0, Table 2 of the manuscript).

We hope that adding the part 2.2.1 (see point 2) which contains a description of dissolved and particulate organic matter representation in the model will clarify this point.

**4) Somehow it looks like (and I am sorry to say that, but I am sure you agree with me) the work you've done here (changing from autochthonous TA formulation – what is what you ideally want to use – to the abiotic, allochthonous formulation) is a way to fix a problem due to the configuration choice, that is not done for this kind of study. Your conclusion (you need to switch from 0D to 3 or at least 1D) should have been one of Lajaunie-Salla et al. 2021's study.**

We agree, by changing the formulation of TA from autochthonous to allochthonous, our aim was to correct a bias due to the configuration choice.

By implementing an allochthonous formulation of TA in Eco3M_MIX-CarbOx, we first wanted to explain the result obtained by Lajaunie-Salla et al. (2021) and make sure that this result was not due to a poor representation of a biological process that could affect TA dynamics. Then, with the results obtained in our study, we could confirm that the lack of variation in the TA representation of Lajaunie-Salla et al. (2021) is explained by the fact that the 0D configuration does not allow (at first sight) the consideration of allochthonous contributions and especially of the Rhône River which is the main source of TA variations in the area.

Nevertheless, the 0D configuration has several advantages, including its short calculation time which allowed us to provide a detailed analysis of drivers of seawater $p$CO$_2$ variations, particularly during specific hydrodynamic processes typical for the BoM. This type of study is still uncommon in the area, as few of them investigated the carbonate system dynamics, especially the $p$CO$_2$ variations drivers (reference study: Wimart-Rousseau et al., 2020) and would have been more difficult to conduct in 3D. That is why we chose to keep working on the 0D configuration (therefore, looking for a way to better represent the TA on which $p$CO$_2$ calculation depends) and present the results obtained with this configuration as we think that, even if some points deserve to be reworked in 3D, the 0D already allows to obtain interesting results.

We understand that our choice to work with a 0D configuration may raise questions from readers. We propose to modify the last sentence of the discussion part (l.676) to better explain our choice :

[Nevertheless, dimensionless model also offers some advantages including short simulation time and easy adaptability **which allowed us to provide a detailed analysis of drivers of seawater $pCO_2$ variations, particularly during specific hydrodynamic processes typical for the BoM. This type of study is still uncommon in the area, as few of them investigated the carbonate system dynamics, especially the $pCO_2$ variations drivers and would have been more complex to conduct in 3D (i.e., longer simulations and isolation of pCO_2 variation drivers' contributions more difficult as the model is more complex).**]

**5) Before publication I would require the author to better explain the choice and implications of the 0D in the method section, so that the reader can really understand the experiments and the results.**

**- How are the nutrients managed (initialized with annual average value)?**
**- What happens to all sinking materials ? Even if they stay in the box, we need to know.**

We detailed how nutrients and sinking materials are managed by the model in point 2 and 3 of this response, respectively. Nutrients dynamics are the results of biological processes which take place in the box only and particles which are large enough to sink (i.e., POM, organisms larger than CM and $CaCO_3$ in our case) are not remove from the surface box by sinking, they stay in the box. Both, nutrients and sinking materials, are constantly recycled in the box.

**- Are total N, P, SI, Fe, Alk supposed to be conserved within the box ? Or are they allowed to fluctuate with some external sources and sinks from/to outside the box, apart from the air-sea $CO_2$ flux?**

We do not consider Si and Fe in our model. Total N, and P are supposed to be conserved within the box as we do not consider any external source or sink from/to the water column (Fig. II).

[Figure]

**Figure II**. Time-series of daily averaged total nitrogen ($N_{TOT}$) and phosphorus ($P_{TOT}$) for the three years of simulation (repetition of 2017 three times) for the reference simulation (SIMC0, Table 2 of the manuscript).

When TA calculation is based on autochthonous formulation, TA is supposed to be conserved in the box. However, when we repeat the year 2017 three times, we observe a decreasing trend for TA (-16µmol kg$^{-1}$ in three years) (Fig. III). This decrease is explained by the prevailing of precipitation compared to dissolution. For the bay of Marseille, this result seems consistent as other studies (Bensoussan & Gattuso, 2007 ; Wimart-Rousseau et al., 2020) suggest a net calcifying system, however, in our case, it means that TA is not conserved in the box.

When TA calculation is based on allochthonous formulation, the TA result is the balance between Rhône River TA sources and sinks due to the net calcifying system.

[Figure]

**Figure III.** Time series of daily averaged TA for the three years of simulation (repetition of 2017 three times) for the reference simulation (SIMC0, Table 2 of the manuscript).

**I ask for major revision, just to be sure this part is improved.**

To improve this part, in addition to add subsection 2.2.1 Nutrients and organic matter representation in the model**,** we modified the subsection 2.2 Model description to better explain the 0D concept and the implications for the present study:

l.119: **[**In this study, we used the Eco3M_MIX-CarbOx model (v1.0) which was developed to represent the dynamics of the seawater carbonate system and mixotrophs in the BoM and was implemented using the Eco3M (Ecological Mechanistic and Molecular Modelling) platform (Baklouti et al., 2006a, b). Eco3M_MIX-CarbOx is a dimensionless model (0D): we consider a volume of 1 $m^3$ of surface water at SOLEMIO station, in this volume the state variables only vary over time as the model is not coupled with a hydrodynamic model. We chose to use a 0D configuration as this configuration has several advantages namely, calculation times are low (around 45 minutes in our case). It allows to make several test simulations to better understand the biogeochemical functioning of the BoM and its possible reactions to environmental forcings. **]**

l.137: **[**

[Figure]

**Figure 1. Schematic representation of 0D concept used in this study with Eco3M_MIX-CarbOx. T: temperature, S: Salinity and OM: Organic matter.**

By using the dimensionless model Eco3M_MIX-CarbOx, we aim to represent a small volume of surface water (1 m³) at the SOLEMIO station (Fig. 1). This small volume is closed which means that: (i) it does not exchange matter (i.e., nutrients, organic matter, organisms) with the water column, (ii) in our case, as we implemented a carbonate module which allows the representation of air-sea $CO_2$ fluxes, the only exchanges allow between the volume and the atmosphere are the air-sea $CO_2$ fluxes, (iii) within the volume the matter is continuously recycled. As a result, when the water column is impacted by an hydrodynamic event which modifies its properties (i.e., which bring nutrients, organic matter, impact salinity or temperature for example), the event impacts only temperature and salinity of the volume (Note: in the volume, TA may be impacted by a specific event : Rhône river intrusion in the BoM, we detailed this particular case in subsection 2.2.2 ; Fig. 1), and total N and P are supposed to be conserved within the volume as, contrary to C, we do not consider any external source or sink from/to the water column or the atmosphere (see Fig. S1 of supplementary material for total N and P conservation verification).

In the following, we provide a detailed description of the carbonate system module. We also give a brief description of nutrients and organic matter representation. A detailed description of other compartments, especially of mixotrophs compartment can be found in Barré et al. (2023a). Equations and parameters used by the model are also explained in this previous study.]

We also modified the new subsection 2.2.1 Nutrients and organic matter representation, and the subsection 2.2.2 TA formulation, to refer to the new figure 1:

l.156: [As we use a dimensionless configuration, we assume that nutrients are fully the result of autochthonous biological processes. In other terms, we do not consider allochthonous inputs of nutrients (i.e., from rivers or atmosphere as instance, **Fig. 1**)]

l.183: [POM particles are large enough to sink, however, we do not consider a term to represent their removal from the surface box by sinking. In our case, the POM, such as the DOM, stay in the box and is constantly recycling **(Fig. 1)**]

l.194: [To remedy this shortcoming, we decided to express TA in two ways. In the first one, we considered only autochthonous TA variations **(i.e., variations of TA are only the result of processes which take place in the volume, Fig. 1)**. In the second one, we considered allochthonous TA variations **(i.e., in the volume, TA dynamics is impacted by external contributions, Fig. 1).**]

and we propose to add figure II to supplementary material:

**[S1.1 Total N and P conservation in Eco3M_MIX-CarbOx**

**By using the dimensionless model Eco3M_MIX-CarbOx, we aim to represent a small volume of surface water (1 m$^3$) at the SOLEMIO station. This small volume is closed which means that: (i) it does not exchange matter with the water column or the atmosphere, except for air-sea $CO_2$ exchanges, and (ii) within the volume, the matter is continuously recycled. Accordingly, total N and P are supposed to be conserved in this small volume. To check that, we sum the variables in N (P) for the reference simulation (SIMC0, Table 2 of the manuscript) for the three years of simulation (repetition of the year 2017 three times) (Fig. S1).**

[Figure]

**Figure S1. Time-series of daily averaged total nitrogen (NTOT) and phosphorus (PTOT) for the three years of simulation (repetition of 2017 three times) for the reference simulation (SIMC0, Table 2 of the manuscript).**]

**For the rest I cannot ask you to re-do everything in 1 or 3D, this will most probably be your next paper anyway.**

Indeed, we plan to study carbonate system variables dynamics in the BoM using a 3D coupled model and to present the results of this study in another manuscript.

Modifications in the text:

**The English might need some rewording. I am not an english native, so I cannot help much for that, but I would recommend a second read. For example, you make an extensive use of the word "yielding". It is a nice word, but you should replace some of them with relevant synonyms.**

As we are not native speakers, we have sent our manuscript to a native speaker before submission. So, the version provided to you was already corrected and rephrased by a native speaker, however, we took into account your suggestion and minimize the use of "yielding".

l.200: We rephrased [As a last term we included the mixotrophic uptake of nutrients which yields the following state equation for TA] to [As a last term we included the mixotrophic uptake of nutrients. **TA is calculated as follows:**]

l.369: We modified [Both simulations yield a strong decrease of $p$CO$_2$ on March 15th , in response to a Rhône River intrusion in the BoM.] to [**For both simulations a strong decrease of $p$CO$_2$ is modelled on March 15th , in response to a Rhône River intrusion in the BoM.**]

l.301 (old): We deleted [Regarding the coast function, simulations yielded CF < 2 for all variables which is considered very good (CF < 1) or good (1 ≤ CF < 2) (Table 3).]**.**

l.388: We modified [Furthermore, SIMC1 produced the best TA representation yielding the lowest values for CF, %BIAS and RMSD (Table 3).] to [Furthermore, SIMC1 produced the best TA representation **resulting in** the lowest values for CF, %BIAS and RMSD (Table 3).]

l.519: We changed [In contrast, the allochthonous formulation yielded a much high variability in TA that was close to in situ observations.] to [In contrast, the allochthonous formulation **produced** a much high variability in TA that was close to in situ observations.]

l.527: We modified [Having neglected other allochthonous drivers seems to be justified by the results which yielded a close match to observations and a generally better representation of the other carbonate system variables since DIC, $p$CO$_2$ and pH$_T$ are all closely related to TA (Fig. 4 and Table 3).] by [Having neglected other allochthonous drivers seems to be justified by the results which **showed** a close match to observations and a generally better representation of the other carbonate system variables since DIC, $p$CO$_2$ and pH$_T$ are all closely related to TA (Fig. 4 and Table 3).]

l.623: We changed [The reason for this discrepancy may be related to the fact that our model overestimates seawater $p$CO$_2$ during winter, yielding a sea-air difference close to zero (Fig. 5d). As a result, despite strong winds and low temperatures which would favour CO$_2$ absorption (Middelburg, 2019), the winter CO$_2$ sink is not well represented.] to [The reason for this discrepancy may be related to the fact that our model overestimates seawater $p$CO$_2$ during winter, **resulting in** a sea-air difference close to zero (Fig. 5d). As a result, despite strong winds and low temperatures which would favour CO$_2$ absorption (Middelburg, 2019), the winter CO$_2$ sink is not well represented.]

**Page2 line 25 : add some "-" or () : "model – consistent with observations – predicted…."**

Done.

**P3-L65 : You need to add something here about the reason for 0D.**

We added (l.65):

[Most modelling approaches to investigate carbonate system variables typically employ 3D coupled physical-biogeochemical models and focus on larger coastal areas (e.g., Artioli et al., 2014; Bourgeois et al., 2016). If the focus is on smaller areas this requires higher spatial and temporal resolution to correctly represent the relevant processes (Bourgeois et al., 2016). **However, higher spatial and temporal resolution often result in a significant increase of the calculation time which make more**

difficult the repetition of numerical experiments, an important step to better understanding the global functioning of the area and its reaction to environmental forcings. A solution to avoid important calculation times is to use a dimensionless model. This type of model allows to conduct large amount of test in short amount of time. As instance, Lajaunie-Salla et al. (2021) used the dimensionless Eco3M-CarbOx model, which contains a carbonate module performing the resolution of the carbonate system based on total alkalinity (TA) and dissolved inorganic carbon (DIC). Even if the DIC, oceanic partial pressure of $CO_2$ ($pCO_2$) and total pH ($pH_T$) representations look reliable, Eco3m-CarbOx tends to minimize the range of TA variations during the year, resulting in a near constant TA (Lajaunie-Salla et al., 2021).]

**P5-table1: You only fill the time resolution information for the wind. Does that mean they're all the same ? It seems from the text that some data are daily. You should feel them all, or tell in the table description why the other data have no time resolution information.**

All data cited in Table 1 are hourly measurements. To avoid confusion, we fill all the lines of the table.

[**Table 1.** Data types and their sources used to drive the environmental forcing during the 2017 model run (based on Barré et al., 2023a).

|  | Data type | Location | Time resolution |
|---|---|---|---|
| **Sea surface temperature** | Measurements | Planier station | **Hourly** |
| **Salinity** | Measurements | Carry buoy | **Hourly** |
| **Wind** | WRF model results | SOLEMIO station | **Hourly** |
| **Irradiance** | WRF model results | SOLEMIO station | **Hourly** |
| **Atmospheric $pCO_2$** | Measurements | Cinq Avenues station | **Hourly** |

]

**P6-L124 to 126. " In addition to …". what you say there sounds obvious, but BGC-model not including mixotroph represent reasonable TA and DIC. Do you have a Reference paper for this statement ?**

We added the reference Mitra et al., (2014): In this article, authors show why it is important to consider a mixotrophs compartment and how not consider it can lead to a failure to capture the true dynamics of the carbon fluxes. (Mitra, A., Flynn, K. J., Burkholder, J. M., Berge, T., Calbet, A., Raven, J. A., Granéli, E., Glibert, P. M., Hansen, P. J., Stoecker, D. K., Thingstad, F., Tillmann, U., Våge, S., Wilken, S., and Zukov M. V.: The role of mixotrophic protists in the biological carbon pump, Biogeosciences, 11, 995-1005, https://doi.org/ 10.5194/bg-11-995-2014, 2014.

**P7-equation1 : You should specify that all terms are define in the appendix A.**

Thank you for pointing this out, we added it l.206: [where i represents the number of organisms. **Processes description can be found in Table A1 (Appendix A) and formulations are available in Barré et al. (2023a).** In this formulation, TA only depends on biogeochemical processes (i.e., TA riverine inputs are excluded).] and l.235: [where i represents the number of organisms. **Processes description can be found in Table A1 (Appendix A) and formulations are available in Barré et al. (2023a).** As an additional modification, we use a more recent version of the gas transfer velocity calculation introduced by Wanninkhof (2014).].

**P7- equ2 and 3 : You can specify the unit at the end of the equation, and remove the following sentence.**

Done.

**I might be wrong but, shouldn't the "photo" terms be more like uptake terms ? Phyto absorbs more DIC than the only ones used for the photosynthesis. Isn't this equation missing the remineralization terms as a source of DIC ?**

Thank you for this interesting comment. We are aware that phytoplankton can absorb more DIC than the one associated with photosynthesis process. However, in Eco3M_MIX-CarbOx, we made the choice to consider only the DIC uptake used through photosynthesis (photosynthetic organisms only use the DIC they need for photosynthesis, surplus is released through respiration process).

The term $Photo_{DIC}^{ORG_{C_i}}$ (with ORG $\epsilon$ [PHYC, MIXC]), when apply to the DIC, represents the DIC uptake linked to photosynthesis, by phytoplankton and mixotrophs. When The term $Photo_{DIC}^{ORG_{C_i}}$ is applied to phytoplankton or mixotrophs, it represents the biomass increase associated with photosynthesis. Both processes are identical, they have the same formulation (photosynthesis based on Geider et al. ,1998) except that when the process is applied to DIC, the biogeochemical flux applied is negative (positive when applied to phytoplankton and mixotrophs). We then used the same notation to be consistent.

In Eco3M_MIX-CarbOx, we consider DIC remineralization through respiration process. Especially, the DIC which comes from bacteria respiration is actually the result of POC and DOC remineralization.

**P8-equ5 and L179-181 : I don't understand why you define Aera being negative when the CO₂ flux is toward the sea. In Equ 4, ∂DIC/∂T increases with Aera being positive, what means CO₂ flux toward the sea, and pCO₂,sw > pCO₂,atm. There's a discrepancy here you might want to correct.**

Thank you for pointing this out. We corrected the balance equation for DIC (l.167, Eq.4):

$$\frac{\partial \text{DIC}}{\partial t} = \sum_{i=1}^{2}\left(\text{Resp}_{\text{DIC}}^{\text{Phy}_{C_i}}\right) + \sum_{i=1}^{2}\left(\text{Resp}_{\text{DIC}}^{\text{Mix}_{C_i}}\right) + \text{Resp}_{\text{DIC}}^{\text{COP}_C} + \text{BR}_{\text{DIC}}^{\text{BAC}_C} + \text{Diss}_{\text{DIC}}^{\text{CaCO}_3} - \sum_{i=1}^{2}\left(\text{Photo}_{\text{DIC}}^{\text{Phy}_{C_i}}\right)$$
$$- \sum_{i=1}^{2}\left(\text{Photo}_{\text{DIC}}^{\text{Mix}_{C_i}}\right) - \text{Prec}_{\text{DIC}}^{\text{CaCO}_3} - \textbf{Aera}_{\textbf{DIC}}$$

Equation 5 of the manuscript remains unchanged. We still consider that negative aeration values (pCO₂,atm > pCO₂,sw) are associated with CO₂ fluxes toward the sea and an increase of DIC, which is now consistent with the DIC balance equation.

**P11-L237 : "the first three terms of Eq.(10)", I think you refer to Eq.11, not 10.**

Thank you for this, we modified: [The first three terms of the Eq. (10) can be calculated as follow:] by [The three terms of Eq. (11) can be calculated as follow:].

**P14-Table3 : just to mention, comparing pH is tricky. Comparing pH change or bias in pH unit can be misleading. Best practice is to compare H⁺ concentration. See Kwiatkowski and Orr, 2018 (https://www.nature.com/articles/s41558-017-0054-0).**

We take into account your suggestion and propose to add in Appendix the statistical indicators calculation for H⁺ concentration:

[Appendix C. Statistic indicators calculation for H⁺ concentration

Table C1. Comparing the different model results to surface observations at SOLEMIO station for H⁺ concentration. N represents the number of observations. Mean, SD, AE, AAE and RMSD are in the same unit than the considered variable, i.e.: mmol m⁻³ for H⁺ concentrations. % BIAS is without unit.

|  |  | [H$^+$] |
|---|---|---|
| N | Observations | 20 |
| Mean ± SD | Observations | 8.08 × 10$^{-9}$ ± 5.52 × 10$^{-10}$ |
| Mean ± SD | SIMC0 | 8.89 × 10$^{-9}$ ± 2.91 × 10$^{-10}$ |
|  | SIMC1 | 8.39 × 10$^{-9}$ ± 4.06 × 10$^{-10}$ |
|  | CarbOx | 8.52 × 10$^{-9}$ ± 2.80 × 10$^{-10}$ |
| %BIAS | SIMC0 | -5.33 |
|  | SIMC1 | -3.91 |
|  | CarbOx | -5.47 |
| AE | SIMC0 | -4.30 × 10$^{-10}$ |
|  | SIMC1 | -3.15 × 10$^{-10}$ |
|  | CarbOx | -4.42 × 10$^{-10}$ |
| AAE | SIMC0 | 6.45 × 10$^{-10}$ |
|  | SIMC1 | 6.05 × 10$^{-10}$ |
|  | CarbOx | 6.36 × 10$^{-10}$ |
| RMSD | SIMC0 | 6.98 × 10$^{-10}$ |
|  | SIMC1 | 7.14 × 10$^{-10}$ |
|  | CarbOx | 6.93 × 10$^{-10}$ |

]

Please, note that based on Referee 2 suggestions we modified the statistical indicators calculated. We replaced CF which was not sensitive enough by average error (AE) and average absolute error (AAE):

$$AE = \frac{\sum_{i=1}^{N}(O_i - M_i)}{n}$$

(Eq. IV)

$$AAE = \frac{\sum_{i=1}^{N}(|O_i - M_i|)}{n}$$

(Eq. V)

Where O represents the observations and M the model results, calculations are based on Stow et al. (2009).

And we modified the text accordingly (l.392):

[For statistical indicators, %BIAS values are systematically lower than 10 %, with the highest values obtained for pCO$_2$ with ~6 % while the remaining variables had values < 1 %. Similarly, pCO$_2$ had the highest RMSD, AAE and AE which suggests that this parameter is not as well represented in the model as the other variables. Furthermore, SIMC1 produced the best TA representation resulting in the lowest values for %BIAS, AE, AAE and RMSD (Table 3). Moreover, SIMC1 produced an annual mean-TA that was closest to the observations. While the SIMC0 and Eco3m-CarbOx results are fairly similar. SIMC0 produced a slightly better representation of TA compared to Eco3m-CarbOx (%BIAS, AE, AAE and RMSD slightly lower). For pH$_T$, SIMC1 outperformed SIMC0 based on %BIAS (Table 3), however, AE, AAE and RMSD values are similar for the three simulations. We then performed the calculation of statistical indicators on H$^+$ concentration as, according to some authors (Kwiatkowski & Orr, 2018), comparing H$^+$ concentrations is a better practice than comparing pH. Results are available in Appendix C. Based on Table C1, SIMC1 also outperformed SIMC0 based on AE and AAE. For studying DIC and pCO$_2$, the situation is less clear as the simulations performed differently for different indicators, making it difficult to pick a clear winner. Still SIMC1 shows the best AAE and RMSD values

for DIC, and the best %BIAS, AE, and AAE for $p$CO$_2$. In conclusion, SIMC1 shows the best overall indicator values for the examined variables (more specifically, it outperformed the other simulations in 13 of 20 indicator comparisons when including H$^+$ concentrations comparison).]

P15 -Fig 5 : the e,f,g and h panels are not useful. There is no additional information, and it's not even zoomed-in. Instead, I would remove them, make the picture slightly bigger, and highlight the SUP like you do in Fig. 6 with a shading or something similar.

Done. We modified the caption and the text accordingly.

[

[Figure]

Figure 5. Time series of (a) in situ daily average sea surface temperature (black line) and salinity (grey line) (b) SIMC1 daily average wind speed (c) the difference between SIMC1 daily average seawater $p$CO$_2$ and in situ daily average atmospheric $p$CO$_2$ (d) SIMC1 daily average air-sea CO$_2$ fluxes (aeration process). The summer upwelling period (from 1 May to 1 October) is highlighted in yellow.]

**P16-Fig 6 : The panel d is quite difficult to look out, it can be quite difficult to differentiate the different blue lines (especially nTA and nDIC have very similar colours). Plus, most of the time the curves in this panel are between -100 to +100 µatm, while the y-axe goes from -600 to +600 µatm. Apart from the big events, it's quite difficult to see what's happening there. Maybe take the whole page for this picture ?**

We agree. We changed the colour used for nTA representation in figure 6d.

[Figure]

**Figure 6.** Time series for 2017 of daily average (a) in situ temperature and salinity (b) modelled nDIC and nTA (c) modelled seawater and in situ atmospheric $pCO_2$ (d) $pCO_2$ anomalies generated by DIC, TA, S+Fw and temperature based on the approach in Lovenduski et al. (2007) (Note: the dark blue line is sometimes obscured by the black line, especially in March. **An enlargement of the panel d is available in Appendix D.**) (e, j) $pCO_2$ anomalies generated by aeration, solubility, and biological processes based

on the approach in Turi et al. (2014). LSE and an upwelling event have been highlighted. The summer upwelling period (SUP) is indicated by yellow shading.]

If necessary, we also propose to add an appendix which includes the enlargement of figure 6d.

[**Appendix D: Time series of daily average *p*CO₂ anomalies generated by DIC, TA, S+Fw and temperature based on the approach described in Lovenduski et al. (2007), for 2017. Enlargement of the panel d of the figure 6.**

[Figure]

**Figure D1: Time series for 2017 of daily average (a) *p*CO₂ anomalies generated by DIC, TA, S+Fw and temperature based on the approach in Lovenduski et al. (2007) (Note: the dark blue line is sometimes obscured by the black line, especially in March), (b) Enlargement of the panel a between -250 and 250 µatm. LSE and an upwelling event have been highlighted. The summer upwelling period (SUP) is indicated by yellow shading.]**

*NB: We made corrections in the ΔpCO₂ decomposition formulation based on Lovenduski et al. (2007) (subsection 2.3.1). These corrections slightly modified the S+Fw term contribution to pCO₂ variations. Accordingly, we corrected the panel (d) of figure 6.*

**P17-L365 to 370 : You forgot to refer to Fig. 6e somewhere in this section.**

We added it (new lines: l.453-460): [The four LSE are also visible in the solubility-generated anomalies generating strong decreases **(Fig. 6e)**. However, only two LSE are easily identifiable (15 March with a drop from -41 µatm to -163 µatm and 6 May with a drop from 8 µatm to -75 µatm) while the other two appear to be obscured by temperature-related counter-movements. Since aeration- and solubility-generated anomalies show opposite seasonality, they partly cancel each other out. While aeration seems to dominate from November to May, (apart from LSE), solubility appears to dominate from May to November and during LSE. Biological processes are never the dominant driver of $p$CO$_2$ variations as they are systematically smaller (by a factor of 2 to 3) than aeration and solubility-generated anomalies **(Fig. 6e)**. Biology-induced anomalies are always negative, providing evidence that biological processes always decrease $p$CO$_2$.]

**P19-L445 : "we could (not cloud) provide".**

Thank you, we corrected it.

**P20 – L477-8 : "While we only considered TA inputs" (only in the allochthonous formulation, I guess), "Rhône River intrusion can also bring nutrient". This is never explained till now. I already said it in the first part of the review, but you have to be clear about this. The reader cannot fully understand your results otherwise. The model biology only feels the environment changes/variations through the physical forcing only (T, S and light). The biology reacts to the Rhône water only because it is fresher, or to the upwelling because it is colder, but not because of the associated nutrient changes (that do not occur). It is important to tell it because the biology can react in the opposite way than otherwise expected and explain that because it is a 0D model you probably don't have much choice (as I understand it). Knowing that, I am surprised by the DIC variations Fig. 3, that are surprisingly good.**

**P21-L482-4 : same remark than just above.**

We hope that adding 2.2.1 and the figure 2 to the manuscript clarifies this point. However, we also modified this paragraph to better emphasize that only changes in temperature, salinity and TA are considered by the model during these events (l.574):

[In all four LSE, biological processes did not have any significant impact on $p$CO$_2$ variations (Fig. 6e). To interpret this result, it is important to consider the assumptions used by Eco3M_MIX-CarbOx (section 2.2). Rhône River intrusion can significantly modify the biogeochemistry of the bay as they are typically associated with temperature and salinity changes and TA, DIC and nutrients inputs (Gatti et al., 2006; Fraysse et al., 2014; Lajaunie-Salla et al., 2021). Due to its 0D configuration, Eco3M_MIX-CarbOx only represents temperature and salinity changes and TA inputs (only if the allochthonous formulation is used for the latter, Fig. 1). Lajaunie-Salla et al. (2021) showed that nutrient inputs associated to Rhône River intrusion in the BoM led to an increase in chlorophyll concentration. This phytoplankton growth leads to further decrease in $p$CO$_2$, which means that by neglecting nutrient inputs we possibly underestimated the importance of biological processes, and especially of autotrophic processes during Rhône River intrusions.]

And l.613:

[Although upwelling events also bring nutrients and DIC to the surface**. In Eco3M_MIX-CarbOx, these effects are not considered, and upwelling events are only represented through temperature decrease in the volume.**]

References:

Barré, L., Diaz, F., Wagener, T., Van Wambeke, F., Mazoyer, C., Yohia, C. and Pinazo, C.: Implementation and assessment of a model including mixotrophs and the carbonate cycle (Eco3M_MIX-CarbOx v1.0) in a highly dynamic Mediterranean coastal environment (Bay of Marseille, France) (Part I): Evolution of ecosystem composition under limited light and nutrient conditions, https://doi.org/10.5194/gmd-16-6701-2023, 2023.

Lajaunie-Salla, K., Diaz, F., Wimart-Rousseau, C., Wagener, T., Lefevre, D., Yohia, C., Xueref-Remy, I., Nathan, B., Armengaud, A., and Pinazo, C.: Implementation and assessment of a carbonate system model (Eco3m-CarbOx v1.1) in a highly dynamic Mediterranean coastal site (Bay of Marseille, France), Geoscience Model Developpment, 14, 295–321, https://doi.org/10.5194/gmd-14-295-2021, 2021.

Mitra, A., Flynn, K. J., Burkholder, J. M., Berge, T., Calbet, A., Raven, J. A., Granéli, E., Glibert, P. M., Hansen, P. J., Stoecker, D. K., Thingstad, F., Tillmann, U., Våge, S., Wilken, S., and Zukov M. V.: The role of mixotrophic protists in the biological carbon pump, Biogeosciences, 11, 995-1005, https://doi.org/10.5194/bg-11-995-2014, 2014.

Stow, C. A., Jolliff, J., McGillicuddy Jr, D. J., Doney, S. C., Allen, J. I., Friedrichs, M. A., Rose, K. A. and Wallhead, P.: Skill assessment for coupled biological/physical models of marine systems, Journal of Marine Systems, 76(1-2), 4-15, 2009.

First, we would like to thank referee 2 for his/her careful evaluation of our manuscript and his/her interesting comments which we believe will help us to improve it. Please, find hereafter our response to these comments.

*The changes made in the manuscript are indicated in blue and the changes made in the supplementary material are indicated in green.*

**1) Firstly, I would say that it does not fit the scope of GMD, which is there to present new developments in models. While the companion paper, with its presentation of a mixotroph compartment, meets this criterium, the main new thing in this manuscript is a diagnostic relation between TA and salinity, which may improve the results, but conceptually is a fairly small step and has been used in many different models so far. From this side I would rather recommend publication in a different journal, where the focus is more on the considered system itself, i.e. a model for the BoM.**

We understand your concern, however we believe that this manuscript has its place in GMD. Eco3M_MIX-CarbOx is a new model which has been developed to consider both mixotrophs and carbonate system. We decided to present Eco3M_MIX-CarbOx in two parts to show both sides of the model distinctly. It allowed us to propose clear studies, easier to read than one which would have been longer, and at the same time to highlight the two main developments which, together, constitute the originality of our model. However, it is important to keep in mind that both parts of the study aim to present this new model. We believe that the study, as a whole (both parts), fits well the scope of GMD and especially, meet the criteria of 'model description papers'.

Moreover, both studies are strongly linked. In the first part, we focused on the planktonic ecosystem description, especially on mixotrophs. We detailed their implementation in the model and study their dynamics in the area. In the second part, we focused on the carbonate system which is barely mentioned in the first part and detailed its representation in the model. This second study is based on the first one as we present a representation of carbonate system by considering the impact of mixotrophs (photosynthesis, respiration...) on these variables. We think that the strong connection between both studies also justify their publication in the same journal.

**2) The switch to a salinity-TA relationship is motivated by the desire to represent the episodic intrusion of freshwater from the nearby Rhone into the BoM, and also the influence of evaporation and precipitation. My first question here is: If these freshwater fluxes affect the TA balance so strongly, should they not also influence DIC?**

First, we would like to stress the fact that "an excess of alkalinity" which likely reflects alkalinity inputs to coastal areas has been described for the entire Mediterranean Sea (Schneider et al., 2007). This study, at the global scale, has forged our conviction that, in a coastal area close to the Rhone River, alkalinity inputs from Rhone River needed to be considered even in this 0D configuration.

The switch to a salinity-TA relationship is possible thanks to the fact that in the bay of Marseille, TA variations are mainly the results of rivers contributions, particularly the Rhône River one. It was demonstrated by the lack of variations observed when we modelled TA only based on biogeochemical processes which take place in the box. For DIC, a different reasoning must be adopted, mainly because the processes which impact DIC dynamics are very different than the one which impact TA dynamic. As we consider a surface layer, DIC dynamics is mostly the results of temperature and salinity changes (which are considered by the model) and biogeochemical processes (especially air-sea $CO_2$ exchanges) (Hassoun et al., 2015). The Rhône River can bring DIC to the BoM, these inputs are diluted (far from

2877 µmol kg⁻¹, the value observed in the Rhône River, Table I) and, due to the action of other processes (solubility effects and biogeochemical processes) on DIC dynamics, which is more pronounced in this case, have a less significant impact than on TA dynamics.

**Table I**. Salinity-DIC couples for LSE events measured at SOLEMIO between 6 June 2016 and 26 June 2019 (last data available).

|  | Salinity | DIC ($\mu mol.kg^{-1}$) |
| --- | --- | --- |
| 6 June 2016 | 37.11 | 2321.3 |
| 4 July 2016 | 37.78 | 2280.1 |
| 2 November 2016 | 37.30 | 2259.3 |
| 15 March 2017 | 36.82 | 2323.8 |
| 5 September 2017 | 37.18 | 2260.3 |
| 31 May 2018 | 37.66 | 2269.8 |
| 26 June 2019 | 37.32 | 2249.0 |

Moreover, it is important to note that, in Eco3M_MIX-CarbOx, TA is the main driver of carbonate system. In other words, a change in TA results in significant changes in DIC, $pH_T$ and $pCO_2$ as demonstrated by the Figure 4 of the manuscript. By representing the contribution of the Rhône River on TA we then indirectly apply it to the three other variables of the carbonate system.

**and nutrients as well?**

[Figure]

**Figure I.** Time series of surface (a) salinity (CARRY measurements), and interpolated (b) NO₃⁻ concentration, (c) NH₄⁺ concentration and (d) PO₄³⁻ concentration at SOLEMIO station. SOLEMIO data are represented by blue markers and the four LSE are indicated by the red dotted lines.

Rhône River intrusion events are associated with an increase of nutrient concentrations in the area, especially nitrate and phosphate (Fraysse et al., 2014). However, in our case, the four low salinity events are not systematically associated with a nutrient increase at the station. In fact, only the first and last events (15 March and 5 September respectively) have an impact on nutrient concentrations at SOLEMIO with the first event being the most significant (Fig. I). This pattern can be explained by the salinity data used by the model. The measurements are performed at CARRY station (near the Côte Bleue, see figure 1 of the manuscript for location) which is more significantly impacted by the Rhône River plume as it is closer to the river mouth than SOLEMIO station. In consequence, decreases of salinity measured at CARRY are not systematically observed or can be less significant at SOLEMIO.

Salinity measurements are also performed at SOLEMIO, however their temporal resolution is low (fortnightly measurements) compared to the CARRY one (hourly measurements). In fact, Rhône River intrusion events duration is variable and can be less than 15 days (ex: short-lived intrusions, Fraysse et al., 2014), therefor it is important to consider the highest temporal resolution possible to better catch them with measurements which is why we chose to work with CARRY measurements instead of SOLEMIO measurements (Fig. Ia).

**That this may lead to biases is discussed in lines 477 to 484; but given the extremely high DIC concentration in Rhone water quoted on line 482, I wonder whether this inconsistency may not invalidate the main results.**

We understand your concerns, however, as indicated in the previous point, the strong DIC value (2877 µmol kg$^{-1}$) observed in the Rhône River never reaches SOLEMIO as the Rhône River plume is quickly diluted and the DIC values which really reaches SOLEMIO station are rarely higher than 2300 µmol kg$^{-1}$ (mean value of 2281.5 µmol kg$^{-1}$, Table I).

Moreover, as indicated in the previous point, it is important to note that in Eco3M_MIX-CarbOx, TA is the main driver of carbonate system. In other words, a change in TA results in significant changes in DIC, pH$_T$ and $p$CO$_2$ as demonstrated by the Figure 4 of the manuscript. By representing the contribution of the Rhône River on TA we then indirectly apply it to the three other variables of the carbonate system.
* * *
To clarify this point, we propose to modify the subsection 4.2.1 by adding:

l.574: [**In all four LSE, biological processes did not have any significant impact on $p$CO$_2$ variations (Fig. 6e). To interpret this result, it is important to consider the assumptions used by Eco3M_MIX-CarbOx (section 2.2). Rhône River intrusion can significantly modify the biogeochemistry of the bay as they are typically associated with temperature and salinity changes and TA, DIC and nutrients inputs (Gatti et al., 2006; Fraysse et al., 2014; Lajaunie-Salla et al., 2021). Due to its 0D configuration, Eco3M_MIX-CarbOx only represents temperature and salinity changes and TA inputs (only if the allochthonous formulation is used for the latter, Fig. 1). For the studied events, linking measured surface salinity to measured DIC (Appendix E) showed that the four events are not systematically associated to a DIC increase at SOLEMIO even though the Rhône River mouth DIC value (2877 µmol kg$^{-1}$, value calculated by using TA and pH from Schneider et al. (2007) and Aucour et al. (1999) respectively) is much higher than the mean value at the station (2294.9 µmol kg$^{-1}$) which means that these values are significantly diluted before reaching SOLEMIO. However, for more realism and as these inputs could affect $p$CO$_2$ variations by increasing the nDIC contribution, considering them could be an interesting addition to the present configuration. Moreover, linking measured surface salinity to measured nutrients concentrations (Appendix E) showed that only the first and last events (15 March and 5 September respectively) have an impact on nutrient concentrations at SOLEMIO with the first event being the most significant.** Lajaunie-Salla et al. (2021) showed that these nutrient inputs led to an increase in chlorophyll concentration. This phytoplankton growth leads to further decrease in $p$CO$_2$, which means that by neglecting these nutrient inputs we possibly underestimated the importance of biological processes, and especially of autotrophic processes during these Rhône River intrusions.]

and added a new Appendix :

[**Appendix E. DIC and nutrients SOLEMIO data interpolation**

**As we represent a closed volume, we do not consider nutrients and DIC inputs which could be associated with LSE or upwelling events (Gatti et al., 2006, Fraysse et al., 2013, 2014, Lajaunie-Salla et al., 2021). To assess if these inputs impact SOLEMIO, we interpolated DIC and nutrients**

measurements performed at the station, then studying the trend observed during the events studied in the present study (Fig. E1).

[Figure]

**Figure E1.** Time series of surface (a) temperature (PLANIER measurements) and salinity (CARRY measurements) and interpolated (b) DIC, (c) $NO_3^-$, (d) $NH_4^+$ and (e) $PO_4^{3-}$ concentration at SOLEMIO station. SOLEMIO data are represented by blue markers. Rhone River intrusions are indicated by the red dotted lines and the SUP is shaded in yellow.

**Table E1.** Surface DIC and nutrients concentration measurements at SOLEMIO station during LSE for the year 2017.

| Date | Event | DIC ($\mu$mol.kg$^{-1}$) | $NO_3^-$ (mmol.m$^{-3}$) | $NH_4^+$ (mmol.m$^{-3}$) | $PO_4^{3-}$ (mmol.m$^{-3}$) |
|---|---|---|---|---|---|
| 15 March | LSE | 2323.8 | 5.5 | 0.03 | 0.07 |
| 6 May | LSE | No measurement available | | | |
| 15 June | LSE | No measurement available | | | |
| 5 September | LSE | 2260.3 | 0.9 | 0.04 | 0.05 |

]

3) This leads me to a more conceptual difficulty with the approach. The model concept is that of an arbitrary one cubic metre volume at the surface of the bay, and that the model just represents fluxes within this volume. Spatial fluxes are excluded (except for $CO_2$ flux, more on that below). This only allows either to model a variable as purely forced from what is happening inside the box, or to prescribe it, e.g. as a function of salinity. For a proper modelling of how external fluxes (e.g. in mol/s) change concentrations (mol/m^3/s) inside the modelled region, one would have to define the volume that is affected by these fluxes. A reasonable choice might be to model a column of water within the mixed layer, as was done in many zero-dimensional models, e.g. Fasham et al, 1990 or Hurtt and Armstrong 1999. That would allow a consistent treatment of the effects of mixing on TA, DIC, nutrients.

**This difficulty becomes especially clear when the authors discuss the possible reasons for their low net annual air-sea flux of CO₂, which is in contrast to observation-based estimates. Here they state that "aeration is is simulated by applying Eq (5) to 1 m^3 of surface water at the SOLEMIO station, which tends to overestimate the effect of aeration processes on DIC..." (line 534 ff). Indeed: if the control volume is that shallow, it will be lead to a too fast approach of DIC towards equilibrium, and hence an underestimate of fluxes.**

We understand your concern. We think that it is important to note that, in this study, we relied on Eco3M-CarbOx (Lajaunie-Salla et al., 2021) for the calculation of carbonate variables. So, Eco3M-CarbOx was our starting point to implement carbonate system variables in Eco3M_MIX-CarbOx. With our study, we aim to bring answers to the concerns raised by this previous study, we then use the same concept and try to improve the representation of carbonate system variables (by adding mixotrophs organisms processes to the state equations and switching TA formulation to a newly implemented allochthonous formulation). In this way, we were able to compare both models, then knowing how our modifications impact the carbonate system variables representation.

Even if we did not manage to obtain a realistic representation of air-sea CO₂ fluxes (mainly for the representation of seasonality and annual mean value), we provide some improvements to the initial concept and give some examples of suitable and unsuitable use of it (first part and second part of this study respectively), then confirming that the only way to obtain realistic fluxes is to consider a larger layer and the processes which impact it. We could have done it the way you propose but, considering that it required a complete review of our 0D configuration, we decided to focus directly on the coupling of Eco3M_MIX-CarbOx, in 3D (which is still in test phase).

Before switching to a 3D configuration, we performed several tests to obtain a better representation of air-sea CO₂ fluxes (sensitivity to $K_{600}$ parameter, $K_{ex}$ formulation and wind formulation). In a more conceptual way, as you suggest it in one of your next comments, we also ran a simulation in which we apply the aeration process to a larger thickness of water (SIMR1 in the following). We considered an annual mean value of 30.5 m for mixed layer depth (mean of winter value = 41 m and summer value = 20 m (Wimart-Rousseau et al., 2020)). Daily average of modelled carbonate system variables and air-sea CO₂ fluxes are represented in figure II for this simulation and compared to the SIMC1 (aeration process apply on a 1 m layer, Table 2 of the manuscript).

[Figure]

**Figure II.** Comparison of model outputs from SIMC1 (aeration process apply on a 1 m layer, Table 2 of the manuscript) and SIMR1 (aeration process apply on 30.5 m layer, model runs showing daily average (a) TA, (b) DIC, (c) $p$CO$_2$, (d) pH$_T$, and air-sea CO$_2$ fluxes for 2017. SOLEMIO data are represented by blue markers.

We obtained a mean annual value for air-sea CO$_2$ fluxes of -113.6 mmol m$^{-2}$ yr$^{-1}$ which is better than the one obtained previously (-0.21 mmol m$^{-2}$ yr$^{-1}$) but still lower than the value suggested by Wimart-Rousseau et al. (2020) (-803 mmol m$^{-2}$ yr$^{-1}$). These results are interesting as we can see that, by considering a larger layer, we better represent the seasonality of air-sea CO$_2$ fluxes (sink in winter and source in summer) (Fig. IIe) which is mainly explained by the fact that we are able to represent an undersaturation for $p$CO$_2$ in winter, especially at the end of the year (Fig. IIc). In this simulation, we used a constant MLD, however, we believe that using a variable MLD (deeper in winter than in summer) could emphasize this result. These results also show that by considering a larger layer to apply the aeration fluxes, we significantly modify DIC representation (Fig. IIb). Thus, the seasonality well modelled previously, is no longer visible. DIC values are also much less variable, then far from the dynamics described by observations.

To conclude, we believe that more than modifying the thickness of the layer impacted by aeration process, we need to switch to a 3D configuration to represent the entire water column and the processes which impact it, to better represent air-sea CO$_2$ fluxes seasonality, annual mean value and by extension, DIC dynamics.
* * *
As these results allow us to support the fact that the switch to a 3D configuration is inevitable, we propose to include them in supplementary material. We also add a table (below) which aims to compare annual mean value and daily range value of SIMC1, SIMR1 and Wimart-Rousseau et al. (2020) study:

**[S1.3 Simulation with modified aeration process**

**By considering a small volume of 1 m$^3$ at the surface, Eco3M_MIX-CarbOx fail to represent seasonality and annual mean value of air-sea CO$_2$ fluxes. To better understand this feature, we ran a simulation with a modified version of aeration process (Eq. S1).**

$$\mathbf{Aera} = \frac{\mathbf{K_{ex}}}{\mathbf{30.5}} * \mathbf{\alpha} * \left(\mathbf{pCO_{2,sw}} - \mathbf{pCO_{2,atm}}\right)$$

**(S1)**

**where Aera is in mmol m$^{-3}$ s$^{-1}$. Kex represents the gas transfer velocity (Wanninkhof, 2014) in cm h$^{-1}$, α the CO$_2$ solubility coefficient (Weiss, 1974) in mol L$^{-1}$ atm$^{-1}$, $p$CO$_{2,sw}$ the seawater $p$CO$_2$ modelled at the previous time step in μatm, $p$CO$_{2,atm}$ the atmospheric $p$CO$_2$ from CAV in μatm. The process is now applied to a larger thickness of water which represents the mean value of mixed layer depth in the area (H = 30.5 m, Wimart-Rousseau et al., 2020).**

**Table S2. Comparison of annual mean value and daily value range obtained for the SIMC1 (H = 1 m), SIMR1 (H = 30.5 m) and in Wimart-Rousseau et al. (2020) study.**

| | Annual mean value (mmol m$^{-2}$ yr$^{-1}$) | Daily value range (mmol m$^{-2}$ d$^{-1}$) |
|---|---|---|
| SIMC1 | -0.21 | [-13, 15] |
| SIMR1 | -113.6 | [-33, 34] |
| Wimart-Rousseau et al. (2020) | -803 | [-15, 10] |

**This new simulation (SIMR1) is compared to the simulation in which allochthonous formulation of TA is used (SIMC1, Table 2 of the manuscript). The representation of the variables of carbonate**

**system and air-sea CO₂ fluxes for both simulations are presented in figure S3. A comparison of annual mean values of air-sea CO₂ fluxes for both simulations and Wimart-Rousseau et al. (2020) study is available in Table S2.**

[Figure]

**Figure S3. Comparison of model outputs from SIMC1 (aeration process apply on a 1 m layer, Table 2 of the manuscript) and SIMR1 (aeration process apply on 30.5 m layer, model runs showing daily average (a) TA, (b) DIC, (c) pCO₂, (d) pH_T, and air-sea CO₂ fluxes for 2017. SOLEMIO data are represented by blue markers.]**

To clarify this point, we modified the subsection 4.3 of the manuscript and refer to the supplementary material:

l.649: **[**Seawater $pCO_2$, air-sea $CO_2$ fluxes and DIC are closely connected (Appendix B, Fig. 3). In Eco3M_MIX-CarbOx, aeration is simulated by applying Eq. (5) to 1 m³ of surface water at SOLEMIO station which tends to overestimate the impact of aeration process on DIC and, due to the close link between DIC and pCO₂, also on pCO₂. **To overcome this problem, we need to consider a larger layer of water on which aeration process is applied. Consequently, we ran a simulation in which we considered a larger thickness of water (H = 30.5, annual mean value of the mixed layer depth in the area ; Eq.8) to apply the aeration process. This simulation and its results are described in supplementary material. By increasing the volume on which aeration process is applied, the annual mean value of air-sea CO₂ fluxes is more realistic (-113.6 mmol m⁻² yr⁻¹), but still, much lower than the one obtained by Wimart-Rousseau et al. (2020) in the area. In fact, to represent the air-sea CO₂ fluxes, especially their annual mean value in a more realistic way, we must consider, on the one hand, a realistic volume of water on which the aeration process is applied and on the other hand, all the processes that take place in the water column and impact this flux. Consequently, overcoming this problem requires the switch to a 3D configuration, which is planned for our future work.]**

and l.673: **[**3D models typically allow more realistic representations of the water column, they would allow us to: (i) consider a more realistic water **column (volume and processes which impact it)** to perform our air-sea CO₂ fluxes calculation, (ii) consider autochthonous and allochthonous contributions to TA variations, (iii) consider the effects of nutrients and DIC inputs from the Rhône River intrusions and local upwellings.]**

**4) And finally, while the diagnostic TA leads to an improvement in model results, as evidenced by decreases in %BIAS, RMSD and a 'cost function' presented in Table 3, the improvements are fairly**

**modest. Indeed I would be interested in knowing whether a model with TA prescribed constant at the average of observations would not have fared at least similarly good as the two presented model cases.**

As suggested, we ran a simulation with a constant TA (mean of SOLEMIO measurements for 2017 = 2591.2 µmol kg⁻¹). In Table II, we presented the calculation of the statistical indicators presented in the manuscript. As we decided to modify them, based on your suggestions, we also provided average absolute error (AAE), and average error (AE) calculated as in Stow et al. (2009) except that, to be consistent with calculations of statistical indicators used previously (Allen et al., 2007) the difference is applied between observations and model which means that for %BIAS and AE, if a positive value is obtained the model underestimates the observations.

**Table II.** Statistical indicators calculation for the simulation with a constant TA (TA = 2591.2 µmol kg⁻¹). Mean, SD, AE, AAE and RMSD are in the same unit than the considered variable, i.e.: µmol kg⁻¹ for TA and DIC and µatm for $pCO_2$. CF and %BIAS are without unit.

|  |  | TA | DIC | $pCO_2$ | pH$_T$ |
|---|---|---|---|---|---|
| **N** | Observation | 20 | 20 | 20 | 20 |
| **Mean ± SD** | Observation | 2591.2 ± 19.4 | 2294.9 ± 24.0 | 391.0 ± 31.0 | 8.09 ± 0.03 |
| **Mean ± SD** | Model | 2591.2 ± 0.22 | 2305.7 ± 26.1 | 418.0 ± 28.9 | 8.07 ± 0.03 |
| **CF** | Model | 0.85 | 0.82 | 1.14 | 1.14 |
| **%BIAS** | Model | -0.002 | -0.50 | -5.79 | 0.26 |
| **RMSD** | Model | 18.90 | 26.14 | 38.45 | 0.03 |
| **AAE** | Model | 16.5 | 19.7 | 35.5 | 0.03 |
| **AE** | Model | -0.06 | -11.5 | -22.6 | 0.02 |

We also represented the daily mean values of TA, DIC, pH$_T$ and $pCO_2$ for the simulations SIMC0, SIMC1 and constant TA (Fig. III) to compare the three simulations carbonate system variables representation.

[Figure]

**Figure III.** Comparison of model outputs from SIMC0 (autochthonous formulation, Table 2 of the manuscript), SIMC1 (allochthonous formulation, Table 2 of the manuscript), and constant TA simulation, model runs showing daily average (a) TA, (b) DIC, (c) $pCO_2$ and, (d) pH$_T$ for 2017. SOLEMIO data are represented by blue markers.

We are aware that improvements seem fairly modest, especially when studying the statistical indicators (Table II of this response and Table 3 of the manuscript). However, values are generally slightly better for the simulation SIMC1, especially for the three other carbonate system variables. In fact, the major improvement brought by the switch to an allochthonous formulation for TA, is that TA variations are represented which seems more realistic and tend to improve the representation of the other three carbonate system variables.

We propose to add these results to supplementary material:

[**S1.2 Simulation with constant TA**

**In addition to simulations with autochthonous and allochthonous TA formulations, we ran a simulation in which TA is set to a constant (mean of surface SOLEMIO measurements for the year 2017: 2591.2 µmol kg$^{-1}$). Statistical indicators (%BIAS, AAE, AE and RMSD) for this simulation are presented in Table S1. We compare the representation of the carbonate system variables obtained for the three types of TA: autochthonous, allochthonous, and constant in figure S2.**

[Figure]

**Figure S2. Comparison of model outputs from SIMC0 (autochthonous formulation, Table 2 of the manuscript), SIMC1 (allochthonous formulation, Table 2 of the manuscript), and constant TA simulation, model runs showing daily average (a) TA, (b) DIC, (c) $p$CO$_2$ and, (d) pH$_T$ for 2017. SOLEMIO data are represented by blue markers.**

**Table S1. Statistical indicators calculation for the simulation with a constant TA (TA = 2591.2 µmol kg$^{-1}$). Mean, SD, AE, AAE and RMSD are in the same unit than the considered variable, i.e.: µmol kg$^{-1}$ for TA and DIC, µatm for $p$CO$_2$ and mmol m$^{-3}$ for [H$^+$]. %BIAS is without unit.**

|  |  | TA | DIC | $p$CO$_2$ | pH$_T$ | [H$^+$] |
|---|---|---|---|---|---|---|
| **N** | **Observation** | 20 | 20 | 20 | 20 | 20 |
| **Mean ± SD** | **Observation** | 2591.2 | 2294.9 | 391.0 | 8.09 | $8.08 \times 10^{-9}$ |
|  |  | ± 19.4 | ± 24.0 | ± 31.0 | ± 0.03 | $\pm 5.52 \times 10^{-10}$ |
| **Mean ± SD** | **Model** | 2591.2 | 2305.7 | 418.0 | 8.07 | $8.48 \times 10^{-9}$ |
|  |  | ± 0.22 | ± 26.1 | ± 28.9 | ± 0.03 | $\pm 2.64 \times 10^{-10}$ |
| **%BIAS** | **Model** | -0.002 | -0.50 | -5.79 | 0.26 | -4.95 |

| | | | | | | |
|---|---|---|---|---|---|---|
| AAE | Model | 16.5 | 19.7 | 35.5 | 0.03 | $6.26 \times 10^{-10}$ |
| AE | Model | -0.06 | -11.5 | -22.6 | 0.02 | $-4.00 \times 10^{-10}$ |
| RMSD | Model | 18.90 | 26.14 | 38.45 | 0.03 | $6.78 \times 10^{-10}$ |

]

We modified the manuscript accordingly. We added (l.277): [Simulations were conducted using both formulations (autochthonous and allochthonous) for the year 2017 (Table 2, SIMC0 and SIMC1). **In addition, we ran a simulation in which TA is set to a constant (TA = 2591.2 µmol kg$^{-1}$, Table 2, SIMCSTE). This simulation and its results are detailed in supplementary material.**]

and modified Table 2 and its caption:

[Table 2. Summary of simulation properties. **Simulation with constant TA is detailed in supplementary material.**

| Simulation name | Total Alkalinity | Temperature | Salinity | Air-sea CO₂ fluxes | Biology |
|---|---|---|---|---|---|
| **SIMCSTE-Constant TA** | **Constant: TA = 2591.2 µmol kg$^{-1}$** | **Temperature file** | **Salinity file** | **Allowed** | **Yes** |
| **SIMC0-Modelled TA (autochthonous formulation)** | Modelled (Eq. 4) | Temperature file | Salinity file | Allowed | Yes |
| **SIMC1-Calculated TA (allochthonous formulation)** | Calculated: TA = f(S) | Temperature file | Salinity file | Allowed | Yes |
| **SIMC2-Aeration effect** | Calculated: TA = f(S) | Temperature file | Salinity file | Not allowed | Yes |
| **SIMC3-Biology effect** | Calculated: TA = f(S) | Temperature file | Salinity file | Not allowed | No |
| **SIMC4-Solubility effect** | Calculated: TA = f(S) | Constant: T= 16.4°C | Constant: S = 38.1 | Not allowed | No |

]

Major comments :

**Figure 1 is identical to the one on the companion paper and definitively isn't needed should this paper be published in GMD.**

Done. We replaced the map by the following sentence: [**A map of the study area showing the location of stations where measurements were carried, and places of interest can be found in Barré et al. (2023a).**] and added it at the end of the 2.1 (l.116).

**State equation for TA, Eq. (1): The terms in the equation are not properly defined. The definition of the terms is given in the Appendix (Table A1), but the table is not referenced here.**

Thank you for pointing this out, we added it l.206: [where i represents the number of organisms. **Processes description can be found in Table A1 (Appendix A) and formulations are available in Barré et al. (2023a).** In this formulation, TA only depends on biogeochemical processes (i.e., TA riverine inputs are excluded).] and l.235: [where i represents the number of organisms. **Processes description can be found in Table A1 (Appendix A) and formulations are available in Barré et al. (2023a).** As an additional

modification, we use a more recent version of the gas transfer velocity calculation introduced by Wanninkhof (2014).].

**The two linear S-TA relations, presented on page 7, which are valid below and above a salinity threshold of 37.8 are discontinuous at S=37.8. This should lead to sudden jumps in the TA value if this threshold is crossed. Are there any effects of this discontinuity visible in the results?**

Thank you for this interesting comment. Indeed, using the allochthonous formulation leads to sudden jump in TA when the threshold is crossed. These sudden jumps are also observed in $pH_T$ and $pCO_2$.

When we implemented the TA allochthonous formulation, we used two points: a first one which represents the Rhône River water at the river mouth (S = 0, TA = 2885 µmol kg$^{-1}$) and a second one which represents the Rhône River water which reaches SOLEMIO during a LSE (S = 36.82, TA = 2600.6 µmol kg$^{-1}$). We chose the second point as it was the most significant LSE on the period covered by the SOLEMIO measurements (2017 and 26 June 2019). We then consider it as representative of the Rhône River water which reach the BoM. Even though TA values associated with LSE are variable and highly depend on the period of the year (Fig. IV). TA values equal or above 2600 µmol kg$^{-1}$ do not seem the most representative (Table III) and LSE are associated with TA highly dependent on Rhône River seasonality (mean value = 2575 µmol kg$^{-1}$).

[Figure]

**Figure IV.** TA measurements in the Rhône River (data: Naïades, https://naiades.eaufrance.fr, first data available: January 2018).

**Table III.** Salinity-TA couples for LSE events measured at SOLEMIO between 6 June 2016 and 26 June 2019 (last data available).

|  | Salinity | TA (µmol kg-1) |
|---|---|---|
| 6 June 2016 | 37.11 | 2603.0 |
| 4 July 2016 | 37.78 | 2579.6 |
| 2 November 2016 | 37.30 | 2585.5 |
| 15 March 2017 | 36.82 | 2600.6 |
| 5 September 2017 | 37.18 | 2560.8 |
| 31 May 2018 | 37.66 | 2568.4 |
| 26 June 2019 | 37.32 | 2520.7 |

We think that it might be interesting to improve our allochthonous formulation to better manage the threshold crossing case. To avoid (or at least reduce these instabilities) it could be interesting to take into account Rhône River seasonality in the allochthonous formulation.

As we think it can be an interesting way to improve our allochthonous formulation, we mentioned it in the manuscript:

l.539: [While our results seem to provide a realistic representation of TA dynamics in the BoM, we could have included other factors such as sediments, which have been shown to be important for TA dynamics, particularly in coastal areas (Brenner et al., 2016; Gustafsson et al., 2014). We plan to add TA supplies by sediments in our future work. **Moreover, from a more conceptual perspective, the use of the present TA allochthonous formulation allowed to manage two cases of salinity, namely S ≤ 37.8 and S > 37.8 with two different equations (Eq.5 and 6), however the switch from one to another, in other words crossing the threshold value, may lead to instabilities in TA representation. A solution to better manage the threshold crossing case is to represent the Rhone River inputs more realistically. Here, we used two S-TA couples (S and TA at the mouth of the Rhône River and S and TA measured at SOLEMIO during the most significant Rhone River intrusion event of 2017) to obtain the dilution formulation. With this method, we do not take into account the seasonality of TA in the Rhône River which can bring significant variations (Figure S4 and Table S3 of supplementary material).**]

We also added Figure IV and Table III to supplementary material.

**Page 8, line 175: In principle the model equations would not change had you chosen to assume the effected layer to be deeper, except that then the flux would then be distributed over a larger volume. Why not take at least H as the annual average mixed layer depth> Taking it as 1m is equivalent to speeding up the gas exchange by a factor H_real, the real affected layer.**

We hope that the simulation and modifications mentioned in point 3 of this response clarify this point.

**Page 8 and Appendix B, pH and *p*CO₂ calculation: It is good to see that the pH scale differences are taken properly into account, and fugacity has been calculated correctly. But much of this is fairly standard, e.g. the iterative calculation of pH, described in Figure B1. This could be left away.**

We considered your comment, however, as we provide some corrections to Lajaunie-Salla et al. (2021) Appendix A, we think that it is necessary to keep Appendix B in the manuscript.

**Page 9, Figure 3: The quality of the Figure is awful. But also it does not convey much information, I would leave it away.**

We considered your suggestion. However, we think that figure 3 allows to better visualize the model calculation steps and how these can be modified by the choice of TA formulation. We decided to modified figure 3 to increase its quality:

[

[Figure]

**Figure 3**. Flow diagram illustrating the steps needed to calculate pHT and $p$CO$_2$ (a) using the autochthonous formulation (Eq. 1) and (b) with the allochthonous formulation (Eq. 2 and 3). Physical

forcings include temperature (T), salinity (S), solar irradiance (IRR), wind speed (Wind) and atmospheric $p$CO$_2$ ($p$CO$_{2,ATM}$).]

We hope it is clearer this way.

**Page 10, lines 218-220: It is not clear to me how the salinity-normalized nTA and nDIC are exactly defined, by a linear correlation with salinity with zweo intercept? If so, why do that if the observed S-TA relation in the oceanographic region is different?**

Salinity-normalised changes in nTA and nDIC were calculated as follow:

$$nTA = \frac{\overline{S}}{S} * TA$$

$$nDIC = \frac{\overline{S}}{S} * DIC$$

(Eq. I)

where $\overline{S}$ represents the mean salinity for the year 2017.

We agree that the subsection 2.3.1 of the manuscript needs to be clarify. Also, we made some corrections in this part (l. 284):

[**Following the reasoning presented in Lovenduski et al. (2007), $p$CO$_2$ variations can be expressed as the sum of variations generated by changes in TA, DIC, temperature and salinity as follow:**

$$\Delta pCO_2 = \Delta pCO_2^{TA} + \Delta pCO_2^{DIC} + \Delta pCO_2^{T} + \Delta pCO_2^{S}$$

$$\Delta pCO_2 = \frac{\partial pCO_2}{\partial TA} * (TA - \overline{TA}) + \frac{\partial pCO_2}{\partial DIC} * (DIC - \overline{DIC}) + \frac{\partial pCO_2}{\partial T} * (T - \overline{T}) + \frac{\partial pCO_2}{\partial S} * (S - \overline{S})$$

**(12)**

**Where $\Delta pCO_2$ is in μatm. The overbar in $\overline{TA}, \overline{DIC}, \overline{T},$ and $\overline{S}$ denotes the annual mean. Freshwater inputs can induce changes in TA and DIC. Though, we isolate the changes of TA and DIC due to variations in freshwater inputs using the salinity-normalised TA (nTA = $\overline{S}$/S × TA) and DIC (nDIC = $\overline{S}$/S × DIC) and adding another term to regroup them. For simplicity, we only use one term to designate salinity and freshwater inputs (i.e., S+Fw term). Eq. (12) can thus be rewritten as:**

$$\Delta pCO_2 = \Delta pCO_2^{nTA} + \Delta pCO_2^{nDIC} + \Delta pCO_2^{S+Fw} + \Delta pCO_2^{T}$$

$$\Delta pCO_2 = rS * \frac{\partial pCO_2}{\partial TA} * (nTA - \overline{nTA}) + rS * \frac{\partial pCO_2}{\partial DIC} * (nDIC - \overline{nDIC}) + \frac{\partial pCO_2}{\partial S} * (S - \overline{S}) +$$

$$\left[rSTA * \frac{\partial pCO_2}{\partial TA} * (S - \overline{S}) + rSDIC * \frac{\partial pCO_2}{\partial DIC} * (S - \overline{S})\right] + \frac{\partial pCO_2}{\partial T} * (T - \overline{T})$$

$$rS = \frac{S}{\overline{S}} \,|\, rSTA = \frac{\overline{TA}}{\overline{S}} \,|\, rSDIC = \frac{\overline{DIC}}{\overline{S}}$$

**(13)**

**See Appendix A in Lovenduski et al. (2007) for more details about the computation. Derivatives are obtained using the approach suggested by Sarmiento and Gruber, (2006).]**

By correcting the $\Delta p$CO$_2$ decomposition formulation, we slightly modified the contribution of S+Fw term. Accordingly, we corrected the panel d of figure 6:

[

[Figure]

**Figure 6.** Time series for 2017 of daily average (a) in situ temperature and salinity (b) modelled nDIC and nTA (c) modelled seawater and in situ atmospheric $p$CO$_2$ (d) $p$CO$_2$ anomalies generated by DIC, TA, S+Fw and temperature based on the approach in Lovenduski et al. (2007) (Note: the dark blue line is sometimes obscured by the black line, especially in March. An enlargement of the panel d is available in Appendix D) (e, j) $p$CO$_2$ anomalies generated by aeration, solubility, and biological processes based on the approach in Turi et al. (2014). LSE and an upwelling event have been highlighted. The summer upwelling period (SUP) is indicated by yellow shading.]

**page 11, definition of the statistical indicators: while the definition of RSMD and %BIAS is rather clear, that of the cost function is less clear: Typically, a cost function aggregates model-data-disaggreement for different variables, possibly with different units, into a single scalar variable (Stow**

**et al, 2009). But what exactly the variables are that enter the CF, and how the different variables are nondimenionalized and aggregated into one CF should be properly defined.**

We based our CF calculation on Allen et al. (2007). In this work, CF is defined as: "The cost function gives a non-dimensional value which is indicative of the "goodness of fit" between two sets of data; it quantifies the difference between model results and measurement data (see OSPAR Commission, 1998). It is a measure of ratio of the model data misfit to a measure of the variance of the data; the closer the value is to zero the better the model", and is calculated as follow:

$$CF = \frac{1}{N} \sum_{i=1}^{N} \left( \frac{|O_i - M_i|}{\sigma_O} \right)$$

(Eq. II)

Where O represents the observations, M the model results and σO is the standard deviation of the observations.

**page 11, interpretation of statistical indicators: Whether a CF<1 is considered very good, would probably depend on the definition of CF, and cannot be stated as generally as on line 255-256. If the individual cost function terms e.g. consist of the squared model-data difference scaled by the variance in the individual variables, and are then added together, the expected height of the CF would depend on how many different variables are finally added together. Also, I don't think one can generally say (line 252-253) that a %BIAS<10% is excellent; I would think that depends on the ratio of natural variability to the mean of the variable in question. For TA, with a high background value, a 10% BIAS is rather large.**

To interpret CF and %BIAS, we used the interpretation of Radach & Moll (2006) and Marechal (2004) respectively. We understand that these interpretations seem not strict enough. Moreover, CF indicator seems rather insensitive as almost all variables for the tested simulations show CF value lower than 1. To improve our statistical analysis of model results, we proposed to consider two other indicators: AE and AAE, to replace CF. We based their calculations on Stow et al. (2009) except that, to be consistent with the calculation of RMSD used previously (Allen et al., 2007), the difference is applied between observations and model which means that for AE, if a positive value is obtained the model underestimates the observations. Moreover, we added the formulation of each statistical indicators used to avoid confusion.

We modified the 2.4 accordingly:

[revised manuscript text omitted]

**Table 3, Page 14: If the variance of the observed TA and DIC values is on the order of 20 micromol/kg (note, units should be given in the table), then I'd say a RMSD of about the same order of magnitude is not an excellent agreement. It is not terrible either, though. A similar remark holds for %BIAS.**

We hope that modifications mentioned above allow to clarify this point. We indicated units in the caption of Table 3:

l.380: [Table 3. Comparing the different model results to surface observations at SOLEMIO station for TA, DIC, seawater $p$CO$_2$, and pH$_T$. N represents the number of observations. **Mean, SD, AE, AAE and RMSD are in the same unit than the considered variable, i.e.: µmol kg$^{-1}$ for TA and DIC and µatm for $p$CO$_2$. %BIAS is without unit.**]

**Figure 4, page 12: The time-series of the difference between the model runs (right panel) does not convey much new information, I would remove them.**

We represented the differences between simulations as we think that it allows to better visualize them and then emphasize the fact that modifying TA formulation yields different model outputs for DIC, $p$CO$_2$ and pH$_T$. Considering that, we decided to keep the figure 4 as is.

**Also, I have a question to the data (crosses in Figure 4): to me it is not clear whether all four carbon system variables were measured independently, or whether e.g. DIC and TA were measured, and pH and $p$CO$_2$ calculated from them. If they were measured independently, how consistent are they with respect top each other, given the used set of carbon system equations?**

DIC and TA are measured, and we calculate pH$_T$ and $p$CO$_2$ by using CO2SYSv3 (Sharp et al., 2020, originally developed by Lewis and Wallas (1998)) on MATLAB. The set of constants used is the same than the one used to perform the calculation of pH$_T$ and $p$CO$_2$ in the model.

To clarify, we added:

l.113: [To evaluate our representation of carbonate system variables, we compared our model results to in situ measurements by using a carbonate parameters data set which includes TA, DIC and salinity data (https://www.seanoe.org, last access: 14 February 2023). Measurements are performed fortnightly at SOLEMIO station. **pH$_T$ and $p$CO$_2$ are calculated based on measured TA and DIC, by using CO2SYSv3 (Sharp et al., 2020, originally developed by Lewis and Wallas (1998)) on MATLAB.**]

**Figure 5, page 15: The subpanels on the right are simply a cutout of the panels on the left for the summer period. What is the purpose of this duplicated information?**

We wanted to highlight the SUP, we agree that these panels do not bring more information and that left panel are pretty clear, so we modified Figure 5 by deleting the right panels and highlighted the SUP in yellow on left panels:

[

[Figure]

**Figure 5: Time series of (a) in situ daily average sea surface temperature (black line) and salinity (grey line) (b) SIMC1 daily average wind speed (c) the difference between SIMC1 daily average seawater $pCO_2$ and in situ daily average atmospheric $pCO_2$ (d) SIMC1 daily average air-sea $CO_2$ fluxes (aeration process). The summer upwelling period (from 1 May to 1 October) is highlighted in yellow.**]

**Page 21, Lines 506 ff: Would including the DIC and nutrient input from upwelling improve the model-data agreement, or the converse?**

In our configuration, we tested two ways to consider allochthonous contribution: by using a relation with salinity or by using an interpolation of SOLEMIO measurements which is then read by the model. For DIC, both methods do not give satisfactory results. For nutrients, we used the second methods in Barré et al. (2023a). In both cases, it is difficult to directly test (through a simulation which take them into account) their effect on model-data agreement. However, to answer your question, we propose the followings hypotheses based on DIC and nutrients measurements study. We represent a linear interpolation of SOLEMIO measurements for these variables on figure V.

For DIC, during the SUP, measurements show values around 2283 µmol kg$^{-1}$ (mean of DIC measurements during the SUP). All upwellings of the period do not necessarily impact DIC. Only two events are noticeable: at the beginning of July and mid-September. These two events do not seem to be correlated with LSE, or Cortiou water inputs (generally associated with high $NH_4^+$ concentrations). The first event is not reproduced by the model, we then assume that this DIC increase could be associated with an upwelling event. However, the second one is well reproduced by the model which means that it is not resulting from an upwelling input (as, for now, we do not take these inputs into account in the model) (Fig. 4 of the manuscript). Considering these results, we believe that adding DIC inputs from upwelling could improve the realism of our representation, and consequently the data-model agreement.

For nutrients, it clearly appears that during the SUP, their dynamics are only slightly affected by upwelling events as nutrients concentrations remain close to 0 for most of the time. Only two nutrient inputs are noticeable during the SUP: in July and September. However, these events do not correspond to upwellings as the first one is associated with Cortiou water which reaches SOLEMIO (high $NH_4^+$ concentration) and the second one is, as showed in the manuscript associated with a Rhône River intrusion. This low impact can be explained by the fact that, when the upwelling takes place, nutrients which are upwelled are quickly consumed by the phytoplankton present in the area, then not reaching

the station. Considering these results, we suppose that taking into account nutrients inputs associated with upwelling events could improve the model data agreement as it might bring some more realism to our representation, but not enough to consider them here, as their impact at the station is quite limited.

To conclude, we think that considering upwelling inputs could be a great addition to improve the realism of our representation and then the model-data agreement, especially for DIC, however, considering that, in the present configuration, these contributions can hardly be taken into account, we believe that switching to a 3D configuration will be the most appropriate way to confirm this.

[Figure]

**Figure V.** Time series of surface (a) temperature (PLANIER measurements), and interpolated (b) DIC, (c) $NO_3^-$ concentration, (d) $NH_4^+$ concentration and (e) $PO_4^{3-}$ concentration at SOLEMIO station. SOLEMIO data are represented by blue markers and the SUP is shaded in yellow.

To clarify this point, we propose to modify the subsection 4.2.1 of the manuscript:

l.613: **[**Although upwelling events also bring nutrients and DIC to the surface. **In Eco3M_MIX-CarbOx, these effects are not considered, and upwelling events are only represented through temperature decrease in the volume. During the SUP, by linking surface temperature measurements and surface DIC and nutrients concentration measurements at SOLEMIO (Appendix E), we showed that: (i) among the upwelling events, only two (at the beginning of July and mid-September) are linked to a noticeable DIC variation, and (ii) surface nutrients concentration dynamics seems only slightly affected by upwelling events (nutrients concentrations remain close to 0 for most of the time) explained by the fact that, when the upwelling takes place, nutrients which are upwelled are quickly consumed by the phytoplankton present in the area, then not systematically reaching the station. Even though the effect of upwelling events on DIC and nutrients concentration seems limited at SOLEMIO station, it may be interesting to consider them for more realism as, the temporal coverage of SOLEMIO measurements remains low (15 days) and we cannot exclude the fact that an impact can be observed but not caught by measurements. Indeed, even if low, a nutrient input can promote primary production (Fraysse et al., 2013), then increase the contribution of biological processes (especially of autotrophic processes) resulting in a stronger decrease in $p$CO$_2$ while DIC inputs would**

**increase the importance of nDIC thereby reducing the decrease of $p$CO$_2$ associated with these events.**

**]**

We also completed the Table E1 of the new appendix E (see point 2) with nutrients and DIC measurements during the SUP:

**[Table E1.** Surface DIC and nutrients concentration measurements at SOLEMIO station during LSE **and SUP** for the year 2017.

| Date | Event | DIC ($\mu$mol.kg$^{-1}$) | NO$_3^-$ (mmol.m$^{-3}$) | NH$_4^+$ (mmol.m$^{-3}$) | PO$_4^{3-}$ (mmol.m$^{-3}$) |
|---|---|---|---|---|---|
| 15 March | LSE | 2323.8 | 5.5 | 0.03 | 0.07 |
| 6 May | LSE | No measurement available | | | |
| **10 May** | **SUP** | **2279.1** | **0.1** | **0.01** | **0.01** |
| **24 May** | **SUP** | **2288.7** | **0.06** | **0.02** | **0.02** |
| **8 June** | **SUP** | **2281.0** | **0.05** | **0.02** | **0.03** |
| 15 June | LSE | No measurement available | | | |
| **22 June** | **SUP** | **2299.0** | **0.09** | **0.01** | **0.09** |
| **4 July** | **SUP** | **2316.9** | **0.03** | **0.04** | **0.03** |
| **19 July** | **SUP** | **2277.6** | **0.4** | **1.05** | **0.07** |
| **30 August** | **SUP** | **2262.4** | **0.02** | **0.12** | **0.02** |
| 5 September | LSE **and SUP** | 2260.3 | 0.9 | 0.04 | 0.05 |
| **18 September** | **SUP** | **2305.4** | **0.04** | **0.02** | **0.02** |

**]**

**Page 22, Line 530ff: Can one give a conjecture why the model overestimates pCO2 during winter?**

As seawater $p$CO$_2$ calculation is closely linked to air-sea CO$_2$ fluxes and DIC, we can assume that, when we tend to overestimate the impact of aeration process on DIC, we then impact $p$CO$_2$ and also overestimate it in winter.

We already explained it l.649: **[**Seawater $p$CO$_2$, air-sea CO$_2$ fluxes and DIC are closely connected (Appendix B, Fig. 3). In Eco3M_MIX-CarbOx, aeration is simulated by applying Eq. (5) to 1 m$^3$ of surface water at SOLEMIO station which tends to overestimate the impact of aeration process on DIC and, due to the close link between DIC and $p$CO$_2$, also on $p$CO$_2$**]**.

To make this sentence clearer, we added: **[**Seawater $p$CO$_2$, air-sea CO$_2$ fluxes and DIC are closely connected (Appendix B, Fig. 3). In Eco3M_MIX-CarbOx, aeration is simulated by applying Eq. (5) to 1 m$^3$ of surface water at SOLEMIO station which tends to overestimate the impact of aeration process on DIC and, due to the close link between DIC and $p$CO$_2$, also on $p$CO$_2$. **Indeed, when the aeration process is applied to this small volume, the balance between atmosphere and the volume is quickly reached, which then impact the representation of $p$CO$_2$.]**.

Technical comments:

Thank you for this, we take them into account.

Point #2 : My second comment was on the lack of consistency in the treatment of Alkalinity and DIC (and as later mentioned also nutrients), when effect of Rhone water intrusions on alkalinity is modelled with a salinity-alkalinity relationship. I don't buy the argument that the inputs of DIC and nutrients from Rhone water are 'diluted', as stated in the author's reply, because dilution acts equally on DIC, Alkalinity and nutrients. But I see the point that, while alkalinity is mainly affected by freshwater fluxes, DIC and nutrients are much stronger affected by biological uptake, and DIC additionally by air-sea gas exchange, so that the riverine signal can be lost before the intrusion reaches the modelled site. That this is

**probably case is shown by the authors by referring to data. The authors should nevertheless probably replace the erroneous explanation by 'dilution' from their modified manuscript in this part ("which means that these values are significantly diluted before reaching SOLEMIO"), as it cannot be the physical process of dilution, which would equally dilute the alkalinity signal.**

We agree that the word diluted was inappropriate in our revised manuscript. As suggested, we replaced (l.578) : [For the studied events, linking measured surface salinity to measured DIC (Appendix E) showed that the four events are not systematically associated to a DIC increase at SOLEMIO even though the Rhône River mouth DIC value (2877 µmol kg$^{-1}$, value calculated by using TA and pH from Schneider et al. (2007) and Aucour et al. (1999) respectively) is much higher than the mean value at the station (2294.9 µmol kg$^{-1}$) which means that these values are significantly diluted before reaching SOLEMIO.] by : [For the studied events, linking measured surface salinity to measured DIC (Appendix E) showed that the four events are not systematically associated to a DIC increase at SOLEMIO even though the DIC value obtained at the Rhône River mouth (2877 µmol kg$^{-1}$, value calculated by using TA and pH from Schneider et al. (2007) and Aucour et al. (1999) respectively) is much higher than the mean value at the station (2294.9 µmol kg$^{-1}$). Based on this observation, we can assume that, for DIC, the riverine signal is quickly lost when moving away from the Rhone River mouth and is not reaching SOLEMIO station. Contrary to TA which is mainly affected by Rhone River inputs in the area, DIC is impacted by air-sea CO$_2$ exchanges and biological processes which can explain this pattern.]

**Point #3: Concerning my comments on choosing an arbitrary volume of 1 cubic metre as modelling domain (which was also a major criticism by referee 1), I am happy to see that the authors have made an additional run using a fixed water column of the average depth of the mixed layer at the site. While this is still a step away from using a seasonally varying mixed layer, it at least removes a systematic bias in the relation between the air-sea flux and the change of DIC concentration. I must, however, say that I probably would then have simply replaced all model runs with that choice, rather than treating it as a further sensitivity study, buried in the appendix, given that it leads to a somewhat better annual air-sea flux in the model.**

We understand that treating this run as a further sensitivity study and, therefore, placing it in supplementary material can be a bit frustrating for the reader. We considered other ways to integrate it to the manuscript and at the end, we chose to do as it because, to us, it is the easiest and most suitable way to integrate this run within the manuscript. We considered to replace our reference simulation by this one, as you suggested it above and, in your recommendation, however this replacement would result in several changes. To be consistent, it would require to re-do all the other runs to modify the considered layer thickness and consequently, re-do the entire study to consider these new simulations (statistical indicators calculation, Δ$p$CO$_2$ decompositions, simulations comparisons). We think that it represents a lot of work for a result which is still quite far from what is observed in the literature since the main problem of 0D comes from the fact that, as you specify it in the following, we do not consider all the processes that impact the fluxes in the water column.

**This however, brings me to a weak point of the study that I had somehow overlooked in my last review: I had not realized that the model is indeed completely closed in its nutrient inventory in the model box, because it neglects sinking out of organic matter; everything produced is remineralized there. I am pretty sure that this is the reason that even with**

**assuming a deeper box the model fails to represent the overall annual air-sea flux of carbon at the station: What happens in reality is very likely that over summer a negative pCO2 difference to the atmosphere is maintained because the biomass that was build up is at least partially exported before being remineralized. During winter mixing then, higher DIC and nutrient concentrations are entrained back into the surface layer. Without these vertical fluxes it is no wonder that pCO2 can always stay close to equilibrium. The simplest way how that effect could be implemented into the model would be to assume the mixed layer to be homogeneous but allow mixed layer depth to vary; deepening of the mixed layer then leads to entrainment of water (with DIN and DIC) from below, while shallowing leads to no concentration change. And then also to allow sinking, i.e. loss of organic matter. Such a model setup has been done e.g. for the BATS station, in a paper, I think by Scott Doney in the early nineties, I didn't find it quickly, though.**

We understand your concerns but, as we specified it in our previous response, in this study we relied on Eco3M-CarbOx (Lajaunie-Salla et al., 2021) for the calculation of carbonate system variables. We used this model as a starting point to implement them in Eco3M_MIX-CarbOx and then, bring some answers to the concerns raised by the previous study. By using the same concept in both studies (a closed volume of water of 1m3 at the surface), we were able to compare the results of both models consistently. Even if we did not manage to obtain a realistic annual mean value and seasonality of air-sea $CO_2$ fluxes, the model still manages to provide a good representation of the carbonate system variables.

Correct this pattern by using the method you proposed above requires a complete review of our 0D configuration. As specified in our previous response, we decided to focus directly on the coupling of Eco3M_MIX-CarbOx, in 3D to obtain a better representation of air-sea $CO_2$ fluxes by considering all types of processes which occur in the water column, especially vertical mixing and matter transfer to the bottom of the water column.

**I do not argue that the authors have to do that; but if they don't, they should probably acknowledge that it is impossible with their model to represent the annual cycle of air-sea CO2 flux, although the model might still get the carbonate system in the water approximately right.**

We agree with that. Even if we somewhat approach this subject, it must be stated in a clearer way.

Consequently, we added in the discussion section (l.657): [In fact, to represent the air-sea $CO_2$ fluxes, especially their annual mean value in a more realistic way, we must consider, on the one hand, a realistic volume of water on which the aeration process is applied and on the other hand, all the processes that take place in the water column and impact this flux, *especially vertical mixing and matter transfer to the bottom of the water column. Consequently, in the present state, Eco3M_MIX-CarbOx is unable to represent the annual cycle of air-sea $CO_2$ fluxes.* Overcoming this problem requires the switch to a 3D configuration, which is planned for our future work.]

We also specified it in the conclusion by changing the sentence (l.697): [However, in winter, the model was unable to reproduce the undersaturation seen in seawater $p$CO₂ measurements at SOLEMIO station and rather overestimate it. As a result, the commonly observed seasonality of air-sea $CO_2$ fluxes in the north-western Mediterranean was not reproduced by our model which directly impacted our estimates of the overall yearly air-sea $CO_2$ flux. While correctly identifying the BoM as an overall sink

of $CO_2$, our model significantly underestimated the magnitude (our model : -0.21 mmol m$^{-2}$ per year, Wimart-Rousseau et al., (2020): -803 mmol m$^{-2}$ per year).**]** to: **[**However, in winter, the model was unable to reproduce the undersaturation seen in seawater $p$CO$_2$ measurements at SOLEMIO station and rather overestimate it. As a result, the present configuration of Eco3M_MIX-CarbOx is unable to reproduce the commonly observed seasonality of air-sea $CO_2$ fluxes in the north-western Mediterranean. This pattern directly impacts our estimates of the overall yearly air-sea $CO_2$ flux, as, even if the model clearly identifies the bay as a $CO_2$ sink, it does not allow to reproduce the observed mean annual value of air-sea $CO_2$ fluxes (our model : -0.21 mmol m$^{-2}$ per year, Wimart-Rousseau et al., (2020): -803 mmol m$^{-2}$ per year).**]**

And in the abstract by changing (l. 23): **[**While our model was able to correctly represent the daily range of air-sea $CO_2$ fluxes, we were unable to correctly estimate the yearly total air-sea $CO_2$ flux. Although the model - consistent with observations - predicted the BoM to be a sink of $CO_2$ on a yearly basis, the magnitude of this $CO_2$ sink was underestimated which may be an indication of the limitations inherent in dimensionless models for representing air-sea $CO_2$ fluxes.**]** to: **[**While we were able to correctly represent the daily range of air-sea $CO_2$ fluxes, the present configuration of Eco3M_MIX-CarbOx does not allow to correctly reproduce the annual cycle of air-sea $CO_2$ fluxes observed in the area. This pattern directly impacts our estimates of the overall yearly air-sea $CO_2$ flux, as, even if the model clearly identifies the bay as a $CO_2$ sink, its magnitude was underestimated which may be an indication of the limitations inherent in dimensionless models for representing air-sea $CO_2$ fluxes.**]**

**4) The referee's wider point (#4 and Recommendation) about manuscript length is also worth considering:**

I think it is good that the authors have tested how much their assumed diagnostic relation of TA with salinity gives an improved carbonate system, by comparing with a run with constant alkalinity. It is shown that the prescribed variable alkalinity gives a somewhat better fit to data than the constant alkalinity. I am unsure, though, whether the details of that run really then need to be shown in the paper; I would probably just add the statistics of that run to a table with statistical indicators, without adding new figures.

As you raised this point in your previous report, we thought that other readers might also have this question when reading our manuscript, which is why we saw this run and the associated results as an interesting addition to the manuscript. However, we agree that values of statistical indicators are enough to show that simply using a mean value to represent TA in the model does not allow to represent it in the best way. Consequently, we delete the figure S2 from the supplementary material.

[…] One example in question is the new supplement S1.1 and Figure S1. If all recycling is done within the box, and assuming that the model is written correctly, then the fact that the model conserves total nitrogen and phosphorous does not need to be shown. A short statement in the main text would be enough.

We agree with that. We already specified it in the manuscript (l.141): **[**As a result, when the water column is impacted by an hydrodynamic event which modifies its properties (i.e., which bring nutrients, organic matter, impact salinity or temperature for example), the event impacts only temperature and salinity of the volume, **[...]**, and total N, and P are supposed to be conserved within the volume as,

contrary to C, we do not consider any external source or sink from/to the water column or the atmosphere.**]**

We delete the corresponding part from the supplementary material and the sentence which refers to it in the manuscript (l.146).

**And maybe it would make sense, instead of adding another appendix with the model run with modified mixed layer depth, to make that the new standard, and rather re-do the other runs. That would again save some discussion in the main text and the appendix. These are just suggestions, but I think that a bit more conciseness would help this manuscript to become noticed.**

As specified in point #3, we think that this change represents a lot of work for a result which is still quite far from what is observed in the literature since the main problem of 0D comes from the fact that, as you specify it in the following, we do not consider all the processes that impact the fluxes in the water column. Moreover, we think that such a modification may raise questions from the reader. In the first part of the study (Barré et al., 2023a) we considered an arbitrary volume of $1m^3$ then defining it as a feature of our configuration. To be consistent with this first part it makes sense to continue the study with the same water volume. We then decided to keep this part as it.

However, we agree with the fact that some parts of the manuscript, especially in the materials and methods section can be summarized or move to appendices. We detail the changes made below.

**For instance, if material can be summarised straightforwardly and / or be moved to an appendix, that will help with the readability of your manuscript.**

To help with the readability of our manuscript we propose:

- to reduce the section study area, as a description of the study area, and especially of forcings can be found in the first part of the study:

We delete the Table 1 and refer the reader to the first part of the study instead (l.115): **[A detailed description of forcings used by the model and a map of the study area showing the location of stations where measurements were carried, and places of interests can be found in Section 2.1 of Barré et al. (2023) (Table 1 and Fig. 1 respectively).]**

- to move section 2.2.4 to Appendix B:

To do so, we bring together sections 2.2.2 and 2.2.3, then create a new section "2.2.2 Carbonate system variables calculation" (l.187, in the following we cut the parts which have not been modified for more readability):

**[2.2.2 Carbonate system variables calculation**

**In Eco3M_MIX-CarbOx, we consider the four main carbonate system variables: TA, DIC, $pH_T$ and $pCO_2$. We describe their calculation by the model in this section.**

In Eco3m-CarbOx,TA representation lacks variations during the year. Eco3m-CarbOx did not account for TA inputs by rivers, especially by the Rhône River which has an average alkalinity of 2885 µmol kg⁻¹ (Schneider et al., 2007). **[...]** We implemented both TA-S formulations in our Eco3M_MIX-CarbOx model, and the formulation to be used was chosen based on the salinity : if salinity value used by the

model for the time step considered ≤ 37.8, the TA-S dilution (Eq.6) was applied; else for salinity value > 37.8 the TA-S correlation was applied (Eq. 5, Figs. 2c,d). With this method, TA only depends on salinity (i.e., biological processes are neglected).

The DIC formulation used in our Eco3M_MIX-CarbOx model is very similar to the formulation used in Eco3M-CarbOx except that we added the mixotroph organisms' processes to our equation. As a results, DIC depends on phytoplankton, mixotrophs, zooplankton and bacterial respiration, air-sea $CO_2$ fluxes (aeration process), dissolution of CaCO3, phytoplankton and mixotrophs photosynthesis and precipitation of $CaCO_3$ (Eq.7). **[...]** By convention, we will consider negative aeration values (i.e., $pCO_{2,atm}$ > $pCO_{2,sw}$) to represent fluxes from the atmosphere into the ocean and vice versa. Furthermore, we will express air-sea $CO_2$ fluxes in the more frequently used units of mmol m$^{-2}$ per unit time.

**pH$_T$ and $p$CO$_2$ are then obtained using the value of TA and DIC. Their calculation is detailed in appendix B.** Simulations were conducted using both TA formulations (autochthonous and allochthonous) for the year 2017 (Table 1, SIMC0 and SIMC1). In addition, we ran a simulation in which TA is set to a constant (TA = 2591.2 µmol kg$^{-1}$, Table 1, SIMCSTE). This simulation and its results are detailed in supplementary material.**]**

and we reorganized Appendix B (l.714, in the following we cut the parts which have not been modified for more readability):

**[**Appendix B: pH$_T$ and $p$CO$_2$ calculation

[revised manuscript text omitted]

In addition to TA, DIC, $pH_T$ and $pCO_2$, statistical indicators were calculated for $H^+$ concentrations.]
* * *
We hope that our explanations and the changes that we propose will clarify your latest concerns and improve the readability of our manuscript. Again, we would like to thank you and both referees for your evaluation of our revised manuscript and helpful comments and suggestions.